# Measurement report: Long-term measurements of ozone concentrations in semi-natural African ecosystems

Hagninou Elagnon Venance Donnou[1], Aristide Barthélémy Akpo[1], Money Ossohou[2,3], Claire Delon[4], Véronique Yoboué[3], Dungall Laouali[5], Marie Ouafo-Leumbe[6], Pieter Gideon Van Zyl[7], Ousmane Ndiaye[8], Eric Gardrat[4], Maria Dias-Alves[4], and Corinne Galy-Lacaux[4]

[1]Laboratoire de Physique du Rayonnement, Faculté des Sciences et Techniques, Université d'Abomey-Calavi, Cotonou, 01 B.P. 526, Benin

[2]Department of Physics, University of Man, Man, Côte d'Ivoire

[3]Laboratoire des Sciences de la Matière, de l'Environnement et de l'Energie Solaire, Université Félix Houphouët-Boigny, Abidjan, Côte d'Ivoire

[4]Laboratoire d'Aérologie, Université Toulouse III Paul Sabatier, CNRS, Toulouse, 31400, France

[5]Laboratoire de Climat-Environnement et Matériaux-Rayonnement, Université Abdou Moumouni, Faculté des Sciences et Techniques, Niamey, BP 10662, Niger.

[6]Department of Earth Sciences, Faculty of Sciences, University of Douala, Douala, P.O. Box 2701, Cameroun

[7]Atmospheric Chemistry Research Group, Chemical Resource Beneficiation, North-West University, Potchefstroom, 2520, South Africa

[8]Centre de Recherches Zootechniques de Dahra, Institut Sénégalais de Recherches Agricoles, Dahra, Senegal

Correspondence to: H. E. Venance Donnou (donhelv@yahoo.fr) and Corinne Galy-Lacaux (corinne.galy-lacaux@aero.obs-mip.fr)

**Abstract.** In the framework of the International Network to study Deposition and Atmospheric chemistry in Africa (INDAAF) program, we present the seasonal variability of atmospheric ozone concentrations at the regional scale. The correlations of local atmospheric chemistry and meteorological parameters to ozone photochemistry are investigated, as are long-term trends in ozone concentrations. Fourteen measurement sites were identified for this study, representative of the main African ecosystems: dry savannas (Banizoumbou, Niger; Katibougou and Agoufou, Mali; Bambey and Dahra, Senegal), wet savannas (Lamto, Côte d'Ivoire; Djougou, Benin), forests (Zoétélé, Cameroon; Bomassa, Republic of Congo) and semi-agricultural/arid savanna (Mbita, Kenya; Louis Trichardt, Amersfoort, Skukuza and Cape Point, South Africa). As part of several study programmes, validation and intercomparison tests of passive samplers at remote sites have been carried out to ensure controlled-quality measurements and to provide reliable long-term gases concentrations. Over the period 1995-2020, monthly ozone concentrations were measured at these sites using passive samplers. Monthly averages of surface ozone range from $4.7\pm1.4$ ppb (Bomassa) to $31.0\pm10.5$ ppb (Louis Trichardt). Ozone levels in the wet season (in dry savanna) are higher and in the same order of magnitude to concentrations in the dry season (in wet savanna). In East Africa, ozone levels show no marked seasonality. We established a positive gradient of mean annual $O_3$ concentrations from West Central Africa to South Africa. In the dry savanna, under the influence of temperature, ozone concentrations are closely linked to Biogenic Volatile Organic Carbon (BVOC) emissions ($0.51 < r < 0.95$). It is also sensitive to nitrogen monoxide (NO) emissions in the presence of high precipitation and humidity. Biogenic VOC emissions, anthropogenic NOx, temperature and radiation exhibit a good correlation ($0.49 < r < 0.92$) with $O_3$ in wet savannas and forests. At the southern African sites, the photochemistry of $O_3$ is influenced by humidity, rainfall, temperature, NOx emissions (anthropogenic and biogenic) and VOC. At the annual scale, from 2000 to 2020, Katibougou and Banizoumbou sites (dry savanna) experienced a decrease in ozone concentrations respectively around

-2.4 ppb decade$^{-1}$ (with a very high certainty) and -0.8 ppb decade$^{-1}$ (with a medium certainty) at 95% confidence interval. Seasonal Kendall statistical tests revealed with a high certainty decreasing trends of -0.7 ppb decade$^{-1}$ in Banizoumbou and -2.4 ppb decade$^{-1}$ in Katibougou. These decreasing trends are consistent with those observed for nitrogen dioxide (NO$_2$) and biogenic VOCs. An increasing trend is observed in Zoétélé (2001-2020), with the Sen slope estimated at 0.7 ppb decade$^{-1}$ and at Skukuza (2000-2015; Sen slope = 3.4 ppb decade$^{-1}$). The increasing trends are consistent with the increase in biogenic emissions at Zoétélé and NO$_2$ levels at Skukuza. Very few surface O$_3$ measurements exist in Africa, and long-term results presented in this study are the most extensive for the studied ecosystems.

## 1 Introduction

Ozone (O$_3$) is a greenhouse gas, difficult to observe and quantify on a global scale due to its acute spatial variability resulting from its variable photochemical lifetime (between 20 and 25 days) (Cooper at al., 2020; Young et al., 2013). It has phytotoxic effects altering key plant physiological processes that can significantly reduce the productivity of agricultural crops and ecosystems (Dufour at al., 2021; Lelieveld et al., 2015; Mills et al., 2018; Monks et al., 2015). At the local scale, its presence in high concentrations in the lower troposphere is harmful to human health, notably through irritation of the upper airways (Camredon and Aumont, 2007; Schultz et al., 2017). Ozone is a secondary air pollutant, meaning that it is not emitted directly but formed in the troposphere as a result of oxidative chemical reactions of precursor gases such as nitrogen oxides (NOx), carbon monoxide (CO) and volatile organic compounds (VOCs) (Lu et al., 2019; Schultz et al., 2017). It is chemically lost by photodissociation or by surface deposition and uptake by plant stomata (Silva and Heald, 2018), or by heterogeneous reactions involving aerosols. Stomatal uptake of O$_3$ and subsequent damage to plants can lead to changes in biosphere-climate interactions (Sadiq et al., 2017). Mitigating its negative impacts on health requires reducing both pollutant concentrations and population exposure (Petetin et al., 2022). These changes are compounded by the variation of O$_3$ precursors, which in recent decades have shifted from high and mid-latitudes to low latitudes, where O$_3$ production efficiency is greater (Zhang et al., 2016). These variations are particularly significant in tropical regions, where seasonal cycles linked to natural and anthropogenic sources of gas and particle emissions are well marked (Adon et al., 2010). Zhang et al. (2016) indicate that both modeling and observational studies about ozone trends are not uniform regionally or seasonally, i.e. even in the tropics where a number of sites with ozonesonde profiles exhibit no trend (Thompson et al., 2021). A study with sondes over equatorial southeast Asia by Stauffer et al. (2024), shows no definite ozone trend annually but a 6-8% decade$^{-1}$ increase limited to 3 months.yr$^{-1}$. Air quality forecasts could therefore be used to warn the population of the potential occurrence of a pollution episode (Petetin et al., 2022). Its long-term importance for atmospheric chemistry has been investigated by several studies on air quality (Monks and Leigh 2009) and atmosphere-biosphere interactions (Fowler et al., 2009).

International Global Atmospheric Chemistry Project (IGAC) has produced the Tropospheric Ozone Assessment Report (TOAR) on the global measures for Climate Change, Human Health and Crop/Ecosystem Research (www.igacproject.org/TOAR). This report stated that free tropospheric O$_3$ has increased during the industrial era and in recent decades (Gaudel et al., 2018; Tarasick et al., 2019). Despite these years of regional and global surface O$_3$ research and monitoring, many regions of the world such as Africa, South America, the Middle East and India, remain under sampled, leading to incomplete knowledge of the horizontal, vertical and temporal distribution of O$_3$ (Cooper et al., 2014; Lin et al., 2015; Mills et al., 2018; Oltmans et al., 2013; Sofen et al., 2016). Although Africa is considered as one of the most sensitive continents to air pollution and climate change, it is one of the least studied (Laakso et al., 2012; Swartz et al., 2020a).

From this perspective, long term measurement programs play a vital role in studies of air pollution and the various changes in the chemical composition of the atmosphere. These long-term assessments are crucial for posing the most topical research questions on atmospheric chemistry (Vet et al., 2014), in order to provide the right answers for relevant decision-making at local and global scale. In situ, satellite, O$_3$-sonde and aircraft observations (IAGOS research infrastructure) provide a substantial amount of information on the current distribution of tropospheric O$_3$, its variability and trends (Gaudel et al., 2018; Tarasick et al., 2019). They are well suited to improve our understanding of emissions, transport, chemical reactions,

deposition processes and the impacts of atmospheric species on human health, vegetation and climate change (Lefohn et al., 2018). Numerous projects and programs long-term have therefore sprung up in several places around the world, for decades. In Africa, the International Network to Study Deposition and Atmospheric Composition in AFrica (INDAAF; https://indaaf.obs-mip.fr), operational since 1995, is dedicated to study the evolution of the chemical composition of the atmosphere and deposition fluxes. INDAAF is a national observatory (Service National d'Observation, SNO) of the Institut National des Sciences de l'Univers (INSU) of the Centre National de Recherche Scientifique (CNRS) and of the Institut de Recherche pour le Developpement (IRD), and a labelled component of the European research infrastructure Aerosols, Clouds and Trace gases Research Infrastructure (ACTRIS). The INDAAF long term monitoring network is also labelled by Global Atmosphere Watch (GAW) program of the World Meteorological Organization (WMO) as a contributing network and is a component of the DEBITS (Deposition of Biogeochemically Important Trace Species) activity of IGAC (International Global Atmospheric Chemistry).

Previous studies have considered surface ozone levels in Africa. Indeed, many large African field international campaigns (EXPRESSO, SAFARI/TRACE-A, ORACLES, SAFARI-2000, MOZAIC, AMMA) have been performed over the past 30 years on African air quality and environment. The links to dynamical factors affecting ozone seasonality (Diab et al., 2003, 2004), the interannual variability in ozone related to ENSO and NOx (Balashov et al., 2014) over the South African Highveld, regional convective influence, and the ENSO transition (Thompson et al., 2003b) and widespread impact of biomass burning and domestic fires in southern Africa occurring several months each year are well established (Thompson et al., 2003b). The mean ozone profile in the lower troposphere over the coast of Gulf of Guinea (December-February) and over Congo (June-August) in the burning season is characterized by systematically high ozone (Sauvage et al., 2005). The combination of high NOx emissions from soil north of 13°N and northward advection by the monsoon flux of VOC-enriched air masses contributes to the ozone maximum simulated at higher latitudes (Saunois et al., 2009). Adon et al. (2010) characterized the ozone concentration levels (together with several atmospheric pollutants), from 2002 to 2007, at seven remote sites in West and Central Africa, while Martins et al. (2007) investigated $O_3$ concentrations in Southern Africa over a period of 9 to 11 years (1995-2005). The high ozone values recorded in southern Africa by Martins et al. (2007) are linked to the anthropogenic effect on the chemical species recorded in the atmosphere of the region. Biogenic emissions are the main contributor to ozone production, through the emission of NOx as precursors during the wet season in the dry savanna region (Adon et al., 2010). This result is consistent with the observations made by Saunois et al. (2009) during the AMMA programme. In the dry season (wet savanna), biomass burning is the dominant factor as mentioned by Sauvage et al. (2005). As for the tropical forests of Central Africa, they appear to be a major $O_3$ sink. In South Africa, Swartz et al. (2020a) assessed long-term seasonal and interannual trends of $O_3$ based on a 21-years (1995-2015) dataset at the Cape Point station. This work was continued at 3 historic IDAF-DEBITS sites (Amersfoort, Louis Trichardt, Skukuza, Swartz et al., 2020b). No trends of $O_3$ were observed at these four sites and the concentrations remained relatively constant over the sampling period. The El Niño–Southern Oscillation (ENSO) made a significant contribution to modelled $O_3$ levels at Amersfoort, Louis Trichardt and Skukuza confirming thus the studies of Balashov et al. (2014) and Thompson et al. (2003b). The influence of local and regional meteorological factors were also evident. Laban et al. (2018) reported $O_3$ levels in northeastern South Africa, and characterized the links between observed NOx and $O_3$ concentrations. These studies were completed by the effect of precursor species and meteorological conditions on ozone formation (Laban et al., 2020). The critical role of regional-scale $O_3$ precursors such as high anthropogenic emissions of NOx (under a limited regime of VOC), coupled with meteorological conditions are well emphasised and is agreement with Swartz et al. (2020a,b). Other works were carried out by Bencherif et al. (2020), Brown et al. (2022), Gaudel et al. (2020), Hamdun and Arakaki (2015), Ihedike et al. (2023), Josipovic et al. (2010), Khoder (2009), Lannuque et al. (2021), Lee et al. (2021), Ngoasheng et al. (2021), Tsivlidou et al. (2023) and Zunckel et al. (2004) in different locations in Africa to characterize $O_3$ pollution levels. The conclusions of these studies reported that an increase of tropospheric column with a mean of 1,2 nmol mol$^{-1}$ decade$^{-1}$ (2,4 % decade$^{-1}$) above Gulf of Guinea and +3.6% over South Africa (Bencherif et al., 2020; Gaudel et al., 2020). A strong diurnal variation of $O_3$ is observed with a maximum in the mid-day time

or afternoon due to the local photochemical production (Hamdun and Arakaki, 2015; Ihedike et al., 2023; Khoder, 2009; Zunckel et al., 2004). The low surface ozone concentrations recorded at the studied sites could be caused by titration of $O_3$ by NOx (Hamdun and Arakaki, 2015; Ngoasheng et al., 2021) but the higher NOx concentrations lead to increased $O_3$ chemical production (Brown et al., 2022). The influence of local climatic as harmattan, temperature, humidity and radiation on ozone formation have been also raised (Balashov et al., 2014); Ihedike et al., 2023; Khoder, 2009). Surface emissions from biomass burning contribute of 24% to boundary layer ozone over Africa (Lee et al., 2021). In the studies of Lannuque et al. (2021), Sauvage et al. (2007), Tsivlidou et al. (2023), it appears clearly that tropical meteorology particularly impacts the $O_3$ distributions through the movement of air masses in the Intertropical Convergence Zone (ITCZ) by the north-easterly Harmattan flow (January) or the southeasterly winds and monsoon flow (July).Other projects such as POLCA (POLlution des Capitales Africaines) and DACCIWA (Interactions Dynamique-Aérosols-Chimie-Nuages en Afrique de l'Ouest), have also been implemented in African capitals such as Bamako, Dakar and Yaoundé (Adon et al., 2016), Abidjan, Cotonou (Bahino et al., 2018) and have provided $O_3$ concentration surface measurements. These studies confirmed that in cities where $NO_2$ is high, $O_3$ is less abundant than in rural areas as reported in Hamdun and Arakaki (2015) and Ngoasheng et al. (2021).

However, despite many African studies about ozone and air quality, it should be noted that these campaigns for the most part are only snapshots in time. The number of measurements publicly available is very small and INDAAF is among the few long-term datasets that are available to the scientific community. With the exception of South Africa, $O_3$ variability is not yet sufficiently documented and very little information is available on the long-term evolution of $O_3$ chemistry in Africa (Fleming et al., 2018; Gaudel et al., 2018; Mills et al., 2018). The constraints of the climate response of isoprene emissions, the temperature sensitivity of NOx and $O_3$ chemistry (Brown et al., 2022) on the one hand and on the other hand the meteorological changes meteorological changes when diagnosing regional tropospheric ozone trends and potential shifts in the timing and spatial patterns of biomass burning and ozone precursor emissions in the tropics (Stauffer et al., 2024) are recommended by these authors. The impact of meteorological parameters (temperature, humidity, rainfall, radiation) and atmospheric chemistry (NOx and VOCs concentrations) on the seasonality of $O_3$ concentrations, and the analysis of long-term $O_3$ trends are only partially explained. Further work is therefore needed to fill the data gaps in Africa, and better understand the mechanisms of $O_3$ formation as a function of ecosystems and their long-term evolution.

As part of the INDAAF program, this study aims to improve the long-term assessment of surface $O_3$ in the western, central, eastern and southern African regions. In the first objective, we document the long-term (1995-2020 depending on the site) monthly, seasonal and interannual variability of $O_3$ concentrations on a regional scale at fourteen sites grouped by ecosystem (dry savannas, humid savannas, forests and agricultural/semi-arid savannas), followed by a comparative study with existing references. The study goes further by discussing the seasonal architecture of anthropogenic and biogenic $O_3$ precursors based on meteorological parameters and emission inventories and the correlation between $O_3$ and these factors. In the third objective, we use non-parametric statistical tests to assess long-term seasonal and annual trends in $O_3$, and discuss the results according to trends in anthropogenic and biogenic emissions of precursors and several new trend studies that include African data. For the first time, the chemical evolution of tropospheric $O_3$ is examined over the long term at all INDAAF and companion sites. This study provides a robust regional mapping of the long-term seasonal cycle $O_3$ formation at the continental scale.

## 2. Materials and methods
### 2.1 Sampling sites

Fourteen $O_3$ measurement sites located in different African ecosystems have been selected for this long-term study of tropospheric $O_3$ chemistry (Fig. 1), among which 8 stations of the INDAAF long-term monitoring network located in 7 West and Central Africa countries (Mali, Niger, Ivory Coast, Senegal, Benin, Congo and Cameroon). These sites are characteristic of dry savanna, wet savanna, forest, agricultural and semi-arid ecosystems (Table 1). A detailed description of INDAAF monitoring stations and land use classes is available in Adon et al. (2010, 2013). Other sites implemented through INDAAF's companion projects and using the same $O_3$ measurements protocols were also selected for this study. The site of Dahra in

Senegal, part of the 'Cycle de l'Azote entre la Surface et l'Atmosphère en afriQUE" (CASAQUE) project is located in dry savanna and used for grazing (Bigaignon et al., 2020). The site of Mbita, part of the Integrated Nitrogen Management system (INMS) is located in East Africa. In South Africa, four long-term DEBITS sites (Louis Trichardt, Skukuza, Cape Point and Amersfoort) are considered. They are regionally representative of the specific ecosystems of southern Africa. Full descriptions of these South African sites can be found in the works of Conradie et al. (2016), Laakso et al. (2012) and Swartz et al. (2020a,b).

All study sites are representative of semi natural rural sites in remote regions. Table 1 presents geographical coordinates and some ecological and climatological characteristics of the sites. In the remainder of the text, the measuring stations will be referred to using the following abbreviations: Banizoumbou (Ba), Katibougou (Ka), Agoufou (Ag), Bambey (Bb), Dahra (Da), Lamto (La), Djougou (Dj), Zoetele (Zo), Bomassa (Bo), Mbita (Mb), Louis Trichardt (LT), Skukuza (Sk), Cape Point (CP) and Amersfoort (Af).

**Table 1.** Geographical, ecological and climatic characteristics of the study sites

| Ecosystem | Station | Latitude, Longitude | Country | Land cover classes | Climate |
|---|---|---|---|---|---|
| Dry savanna | Banizoumbou (Ba) | 13°18' N, 02°22' E | Niger | Open grassland with sparse shrub and culture | Sahelian |
| | Katibougou (Ka) | 12°56' N, 07°32' W | Mali | Deciduous shrubland with sparse trees | Sudano-Sahelian |
| | Agoufou (Ag) | 15°20' N, 01°29' W | Mali | Open grassland with sparse shrub and trees | Sahelian |
| | Bambey (Bb) | 14°42' N, 16°28' W | Senegal | Cultivated grass land with sparse trees | Sahelian |
| | Dahra (Da) | 15°24' N, 15°26' W | Senegal | Open grassland with sparse shrub and trees, sylvopastoral area | Sahelian |
| Wet savanna | Lamto (La) | 06°13' N, 05°02' W | Cote d'Ivoire | Mosaic forest/savanna | Guinean |
| | Djougou (Dj) | 09°39' N, 01°44' E | Benin | Deciduous open woodland | Sudano-Guinean |
| Forest | Zoetele (Zo) | 03°10' N, 11°49' E | Cameroun | Dense evergreen lowland forest | Guinean |
| | Bomassa (Bo) | 02°12' N, 16°20' E | Congo | | Guinean |
| Agricultural field | Mbita (Mb) | 0°25' S, 34°12' E | Kenya | Tropical agricultural area | Subtropical |
| Regional savanna/ Semi-arid | Louis Trichardt (LT) | 22°59' S, 30°01' E | South Africa | Cultivated/Semi-arid regional savanna | Subtropical |
| | Skukuza (Sk) | 24°59' S, 31°35' E | | Semi-arid regional background site surrounded by natural bushveld in a protected area Southern | Subtropical |
| | Cape Point (CP) | 34°21' S, 18°29' E | | Hemispherical marine background site, rocky | Mediterranean |

| | | | | and sparsely vegetated, Fynbos biome | |
| | Amersfoort (Af) | 27°04' S, 29°52' E | | Semi-arid regional savanna, impacted by anthropogenic activities | Warmtemperate |

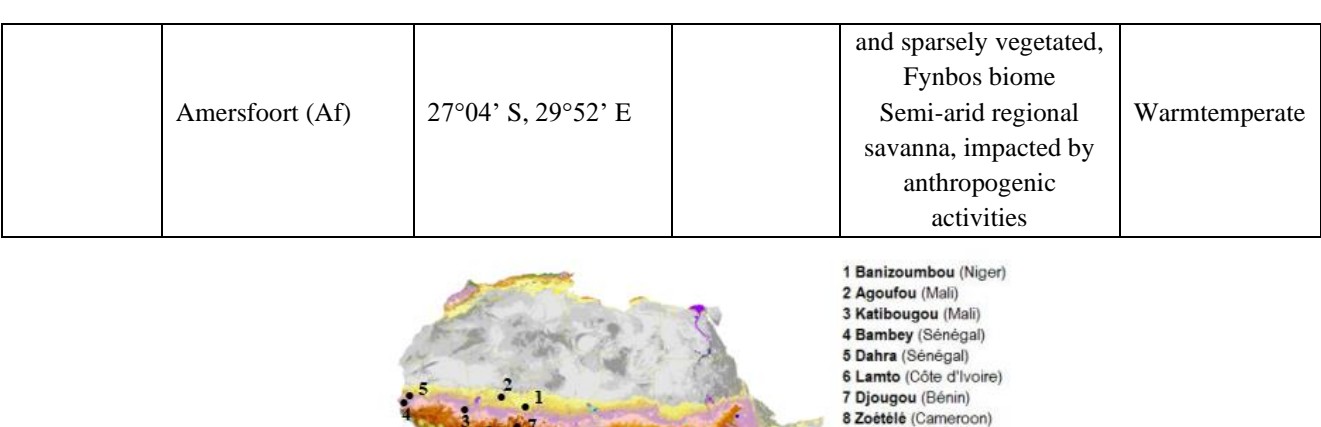

**Figure 1**. Location of the 14 measurement studied sites in Africa on a vegetation map (adapted from Mayaux et al., 2004)


### 2.2 Passive sampling and chemical analysis

#### 2.2.1 Sampling procedure

$O_3$ concentrations were measured using passive samplers developed at the Laboratoire d'Aérologie (LAERO) in Toulouse in the framework of the INDAAF program, and at the North West University in Potchefstroom in South Africa. They are based

on the passive sampling technique, which relies on laminar diffusion and the chemical reaction of the atmospheric pollutant under consideration (Adon et al., 2010; Ferm, 1991 and Martins et al., 2007). These sensors have been tested and validated in different tropical and subtropical regions (Carmichael et al., 2003; Martins et al., 2007). The measurement protocols including passive samplers deployment, analysis by Ionic chromatography and the calculation method of concentrations have been widely described in previous studies (Adon et al., 2010; Bahino et al., 2018; Carmichael et al., 2003; Ferm and Rodhe, 1997;

Galy-Lacaux et al., 2009; Galy-Lacaux and Modi, 1998; Ossohou et al., 2019; Swartz et al., 2020b).

Sampling periods at the measurement sites were coordinated and passive samplers are exposed on a monthly basis using the calendar months. One blank dedicated to ozone is included in the expedition of samplers each two months on sites. In this way, the delay between field deployment and analysis are the same both blanks and exposed samples. All data presented in this paper are blank corrected. A total of 1,317 blanks were assessed at the 14 sites over the studied period. In this paper, we

use monthly database of $O_3$ concentrations. The concentration measurement period runs from the start date of measurements at each site to 2015-2020 (Table 2). $O_3$ concentration collection efficiency (%) on the sampling period (ratio of the number of valid concentrations to the number of filters analysed) was assessed at each of the 14 sites (Table 2). All wet and dry seasons duration are indicated in Table 2. These proportions are fairly representative of high quality measurements as indicated in the work of Laakso et al. (2008, 2012), Laban et al. (2018) and Petäjä et al. (2013). Measurements of $O_3$ concentrations are

continuing at the most of sites and are referenced in the INDAAF website (https://indaaf.obs-mip.fr).

**Table 2**. Sampling period and concentration data collection efficiency

| Ecosystem | Station | Sampling period | Detection limit (ppb) | Data collection efficiency (%) | Total of samplers | Season | Measurement altitude (m) |
|---|---|---|---|---|---|---|---|
| Dry savanna | Ba | 2000-2020 | | 93.5 | 248 | | |
| | Ka | 2001-2020 | | 86.7 | 240 | Dry season: Oct-May | |
| | Ag | 2005-2018 | | 82.6 | 132 | | 1.5 |
| | Bb | 2016-2020 | | 94 | 50 | Wet season: Jun-Sep | |
| | Da | 2012-2020 | | 83.7 | 104 | | |
| Wet savanna | La | 2001-2020 | | 94.2 | 240 | Dry season: Nov-Mar | |
| | Dj | 2005-2020 | | 92.5 | 186 | Wet season: Apr-Oct | 1.5 |
| Forest | Zo | 2001-2020 | | 86.7 | 240 | Dry season: Dec-Feb and July-Aug Wet season: Mar-Jun and Sep-Nov | 3 |
| | Bo | 2001-2020 | 0.1 | 68.3 | 240 | Dry season: Dec-Feb Wet season: Mar-Nov | |
| Agricultural field | Mb | 2017-2020 | | 95.3 | 43 | Dry season: Jun-Oct and Jan-Feb Wet season: Mar-May and Nov-Dec | 1.5 |
| Regional savanna/semi-arid | LT | 1995-2015 | | 95.2 | 248 | Dry season: Apr-Sep Wet season: Oct-Mar | |
| | Sk | 2000-2015 | | 86 | 192 | | |
| | Af | 1997-2015 | 0.02 | 85.5 | 221 | | 1.5 |
| | CP | 1995-2020 | | 90.7 | 248 | Dry season: Oct-Mar Wet season: Apr-Sep | |


**2.2.2 Validation and quality control of INDAAF passive samplers**

To ensure the reliability of the ozone concentrations measurements carried out by the passive sampling monitoring network in West, Central, East and Southern Africa, several validation and quality control tests were carried out as part of the IGAC-

DEBITS program and other collaborations with the Swedish Environmental Research Institute (IVL), the AMMA program, the pilot program for measuring urban meteorology and the environment (GURME) launched by the WMO/GAW and the National University of Singapore. These various performance tests were carried out in Africa at the Banizoumbou (Niger), Zoetele (Cameroon), Lamto (Côte d'Ivoire), Djougou (Benin) and Cape Point (South Africa) sites, in France at Toulouse and in Asia (Singapore). In this assessment, the precision and accuracy of the passive samplers used for $O_3$ monitoring by the

various institutions were determined. $O_3$ detection limits were calculated on the basis of laboratory blank samples. Comparison

of gas concentrations measured by passive samplers (integrated over 15 days) and active analysers was carried out at various sites in Toulouse (1998-2020). The test results indicated a good correlation between the two measurement methods, with an average comparative ratio of 1:0.8 for ozone and a correlation coefficient $R^2 = 0.8$ (Adon et al., 2010). During the 2007 AMMA campaign in Djougou, an intercomparison between measurements from passive samplers and active analysers during the wet season from April to September 2006 revealed that the maximum difference observed between the two techniques (passive/active) was around 6% (Adon et al., 2010). In Banizoumbou, Zoétélé, Lamto and Cape Point, INDAAF and IVL passive samplers were co-located and exposed for a period of one year. The corelation was good between the two types of measurements (Adon et al., 2010; Carmichael et al., 2003). The most recent evaluation of the University of the North West passive samplers used at the South African sites was an international comparison study organised by the National University of Singapore in 2008 (Swartz et al., 2020b). Results indicated that the passive sensors used and operated in INDAAF compared very well with active samplers and had better accuracies. Data quality of the analytical facilities is also ensured through participation in the World Meteorological Organization (WMO) bi-annual Laboratory Intercomparison Study (LIS) (Swartz et al., 2020a,b). The recovery of each ion in standard samples was between 95 % and 105 % (Conradie et al., 2016) and the analysed data were also subjected to the Q test, with a 95 % confidence threshold to identify, evaluate and reject outliers in the datasets (Swartz et al., 2020a). Diffusive samplers have many advantages in the field, including no need for calibration, sampling tubing, electricity or technicians and are small, light, reusable, costefficient and soundless (Adon et al., 2010).

### 2.3 Meteorological parameters and leaf area index

In order to characterize each measurement site, classical meteorological parameters are used such relative humidity, ambient air temperature, rainfall, radiation and leaf area index (LAI). At Ba, Bb, La, Dj and Zo sites, the data on ambient air temperature, relative humidity and rainfall are extracted from the AMMA-CATCH database (Analyse Multidisciplinaire de la Mousson Africaine - Couplage de l'Atmosphère Tropicale et du Cycle Hydrologique; www.amma-catch.org/) and the Observatoire de recherche en environnement "Bassins versants tropicaux expérimentaux" (SO BVET; bvet.obs-mip.fr/) (Ossohou et al., 2019). The measuring devices used at Ka, La and Bo is described in the same work (Ossohou et al., 2019). At Bb site, relative humidity, temperature and rainfall are collected in the INDAAF database (https://indaaf.obs-mip.fr). At Da site, measurements of meteorological parameters come from a measuring station installed by the University of Copenhagen (Bigaignon et al., 2020). In Ka, Ag, Bo, Mb, LT, Af, Sk and CP, the meteorological data are provided by the intermediate reanalysis archive (ERA 5) of the European Center for Medium-Range Weather Forecasts (ECMWF). The time series of global solar radiation used in this study at all sites except Dahra are also ERA 5 reanalysis data obtained from ECMWF. To ensure the reliability of the ERA5 data on the study sites, we determined the estimation errors (RMSE) and the correlation between the reanalysis data and those measured in situ. We chose the Banizoumbou site in Niger (2000-2020), which hosts a meteorological station that provides temperature, humidity and rainfall data, and the Dahra site, where radiation data are measured. We obtained a low error estimate (RMSE) of the order of $9.9 \times 10^{-3}$ °C for temperature, $4.8 \times 10^{-3}$ % for humidity and $2.3 \times 10^{-1}$ mm for rainfall in Niger. At the Bambey site in Senegal, the estimated errors are of the order of $6.4 \times 10^{-2}$ J/m² for radiation. The correlation between in situ and ERA5 data for these two sites is very good and is about of 0.96 for rainfall, 0.99 for humidity, 0.80 for radiation and 0.99 for temperature. LAI data are obtained from MODIS (Moderate Resolution Imaging Spectroradiometer) with a resolution of 0.25 km² for an 8-day time scale centred around each station (Ossohou et al., 2019). All these parameters are collected at the same sampling period as $O_3$ concentrations, with the exception of LAI measurements, which began in 2000 (https://modis.ornl.gov/data.html).

### 2.4 NOₓ and VOC emissions

The emissions of volatile organic compounds (VOCs) and nitrogen oxides ($NO_X$) from biomass combustion were downloaded from the ECCAD (Emissions of atmospheric Compounds and Compilation of Ancillary Data) database in the GFED4 inventories for 0.25°x 0.25° grid cells. BVOC emissions are extracted from the MEGAN-MACC inventory in the ECCAD

database (eccad.aeris-data.fr). The biogenic NO fluxes used are model outputs in reference to the work of Delon et al. (2010, 2012). They were filtered in the eastern grid from 5°S to 20°N in latitude, and 20° W to 30° E in longitude over the period from 2002 to 2007 and cover only the Ba, Ka, Ag, La, Dj, Zo and Bo sites. ECCAD platform is the emissions database of the international GEIA project (Global Emission InitiAtive: geiacenter.org) has been developed within the framework of the French atmospheric data center AERIS (http://www.aeris-data.fr) (Darras et al., 2022). The GFED4 inventory is based on satellite data of fire activities and vegetation productivity observed since 1997 (eccad.aeris-data.fr). MEGAN (Model of Emissions of Gases and Aerosols from Nature) inventory quantifies net biogenic emissions of isoprene and other gases emitted by vegetation into the atmosphere (Guenther et al., 2006; Sindelarova et al., 2014). The determining variables of MEGAN are derived from models and satellite and ground observations, enabling simulations to be carried out on a regional and global scale. They take into account the emission factor, which represents the emission of a compound in the canopy under standard conditions, the emission activity factor, which included changes in emission due to deviations from standard conditions, and the factor that explains production and losses within the plant canopy (Guenther et al., 2006). Isoprene, α-pinene and β-pinene, which account for the largest proportion of BVOCs emitted by vegetation in Africa (Ferreira et al., 2010; Jaars et al. 2016; Liu et al., 2021; Saxton et al., 2007; Serça et al., 2001), were identified and used in this study. A more detailed description of these emission inventories is discussed in the work of García-Lázaro et al. (2018), Guenther et al. (2006) and Vitolo et al. (2018). For each emission category, $NO_x$ (kg m$^{-2}$ s$^{-1}$) and VOC (kg m$^{-2}$ s$^{-1}$), we use the sum of fluxes from all biomass combustion sources (agricultural, waste combustion, savanna, grassland, scrubland, boreal forest, temperate forest, tropical deforestation, peat degradation and peat fires) at a monthly scale and over the study period for each site. These inventories emissions are widely used and Global Fire Emissions Database GFED have been recommended by Stauffer et al. (2024) to study potential shifts in the timing and spatial patterns of biomass burning and ozone precursor emissions in the tropics.

## 2.5 Statistical analysis

The Mann-Kendall and seasonal Kendall tests, associated with the calculation of Sen's slope (Sen, 1968) is applied to all sites with at least 10 years of measure using XLSTAT 2022.2.1.1313 software at 95% confidence intervals. In the case of Kendall's seasonal test, the seasonal nature of the series is taken into account. The literature provides extensive information on Mann Kendall trends calculations (Frimpong et al., 2022; Hirsch et al., 1982; Kendall, 1975; Merabtene et al., 2016). Vectors with p-values less than 0.05 exhibit a very high certainty to obtain the trend, while vectors with p-values in the range of 0.05–0.10 give an indication of a trend (Gaudel et al., 2018). Vectors with p-values in the range of 0.10–0.34 provide a weak indication of change, and p-values greater than 0.34 indicate weak or no change. The vectors with p-values in the range of 0.05–0.34 are very useful for understanding regional trends as they typically follow the same pattern as the very high certainty vectors (Chang et al., 2017; Gaudel et al., 2018). Another non-parametric breakpoint test (Pettitt test) is carried out using Khronostat 1.01 software to assess possible breaks in homogeneity in the $O_3$ concentration series and for optimal application of the trend test.

## 3. Results and discussion
### 3.1 Meteorological and biophysical parameters variation

The monthly variations of meteorological parameters and leaf area indexes (LAI) are shown in Fig. 2 for all sites. In dry savanna, the rainfall regime is unimodal, with the greatest amounts of rain recorded from July to September corresponding at the maxima of LAI. Mean air temperature ranged from 22.1 ± 0.9°C to 34.9 ± 0.4°C, with air relative humidity from 68% to 82%. The most elevated solar radiation is found at Ag (23.1 ± 0.5 MJ.m$^{-2}$). In wet savanna and forest sites, the rainfall pattern and LAI follow a quasi-bimodal distribution. The mean annual LAI varies from 1.2 ± 0.3 m$^2$.m$^{-2}$ (Dj) to 4.7 ± 0.7 m$^2$.m$^{-2}$ (Bo). The most significant monthly variations in relative humidity are found in Dj (23.2 - 84.1%). At Mb site, the maximum rainfall occurs between March and May, reaching 255 mm in April. The vegetation cover is denser at the end of the first wet season (1.3 ± 0.3 m$^2$.m$^{-2}$ in May) with an average value of humidity around 70%. In Southern Africa, the humidity varies from 13% to 22% year-round except at CP where variations are very low. The maximum of rain is collected between December and

January (432 mm on average) at LT, Af and Sk sites and in August at CP (343 mm). LAI maxima are of the order of 1.6 ± 0.2

315    m².m⁻² at Af ; 3.1 ± 0.6 m².m⁻² at LT; 4.0 ± 0.1 m².m⁻² at CP and 1.7 ± 0.2 m².m⁻² at Sk in wet season. In wet savanna and forest, the temperature variations are low, as well as in Mb (23.6 ± 0.5°C). On the other hand, at the South African sites, the temperature reaches amplitudes ranging from 6°C to 10°C. From wet savanna to semi-arid savanna (South Africa), the average solar radiation is below at 22 MJ.m⁻². Along the north-south transect for the study sites, the gradient of humidity, leaf area index and rainfall are positive, whereas it is negative for temperature and radiation. The variations in meteorological parameters

are strongly influenced by the alternating seasons. These characteristics are dependent on the type of climate. Indeed, in West and Central African climate (and its variability) is a function of the position of the Inter Tropical Convergence Zone (ITCZ), which is a band separating the hot and dry continental air coming from the Sahara desert (Harmattan) from the cooler, humid maritime air masses (Monsoon) originating from the equatorial Atlantic Ocean (Adon et al., 2010; Lannuque et al., 2021; Sauvage et al., 2005). Its geographical shift from the Northern Hemisphere during the boreal summer to the Southern

Hemisphere during the boreal winter with different positions throughout the year (for example in January, around 5°N and in August, around 22°N) define the seasons in this region of Africa and explain the marked seasonal variations observed in West and Central Africa (Sauvage et al., 2005). The position of the convergence zone gives rise to the "wet" seasons. Compared with West Africa, East Africa exhibits slightly different regimes due to the topography and the proximity of the Indian Ocean. The climate of southern Africa is characterized by alternating wet and dry periods which are also modified by the position of

the ITCZ (Lannuque et al., 2021). Within the ITCZ warm and humid surface air masses converge and are convectively uplifted into the upper troposphere. The uplifted air masses are then advected polewards in the upper branches of the Hadley cells. The dry air in the descending branches of these cells creates the conditions for wildfires and the resulting emission of ozone precursors (Lannuque et al., 2021).

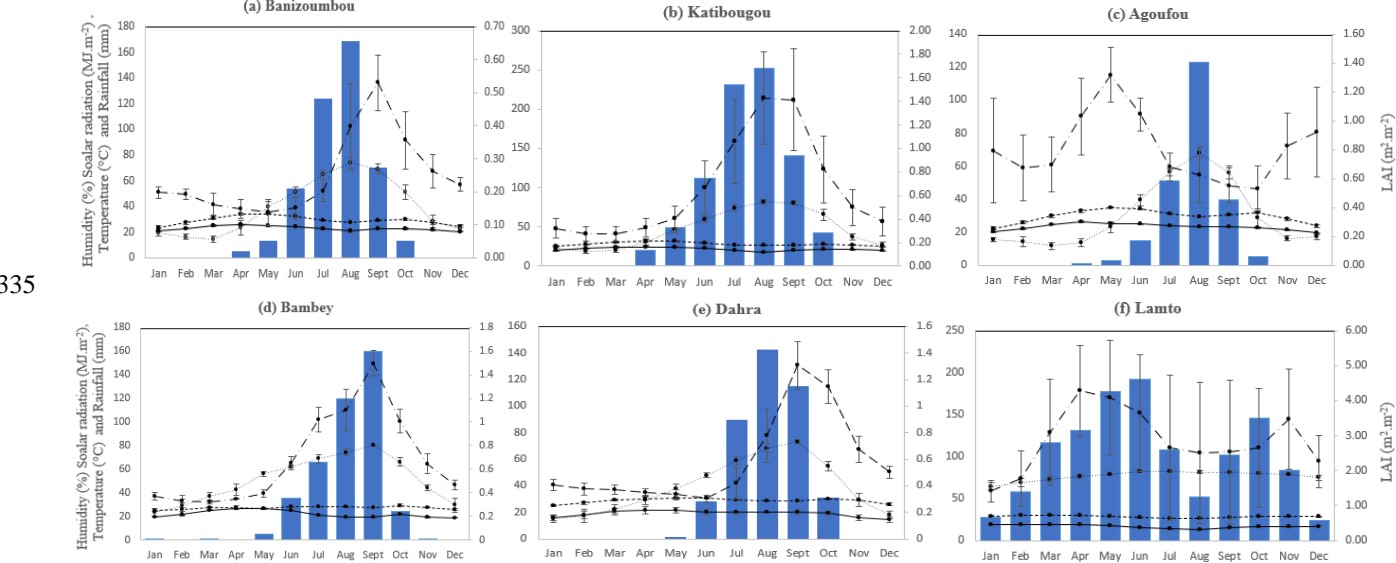

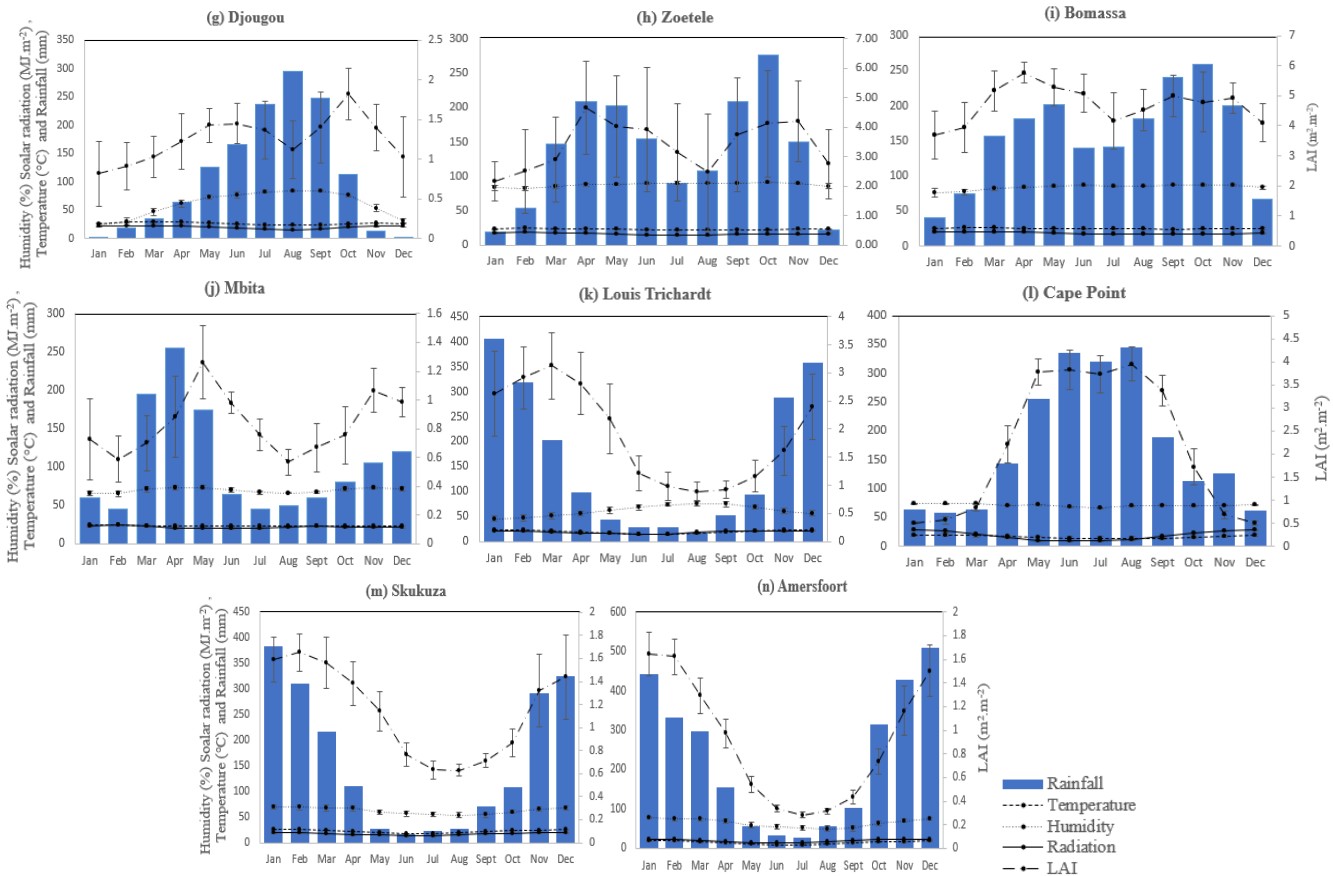

**Figure 2.** Mean monthly variation in air temperature (°C), rainfall (mm), relative humidity (%), solar radiation (MJ.m$^{-2}$) and leaf area index (m$^2$.m$^{-2}$). The mean absolute deviation is represented by the vertical bars.

## 3.2 Characterization of O$_3$ levels

### 3.2.1 Seasonal and annual variation in O$_3$ levels
#### 3.2.1.1 Dry savanna

Figure 3 presents monthly O$_3$ surface concentrations measured in Ba, Ka, Ag (Fig. 3a), Bb and Da (Fig. 3b) representative of dry savannas in Niger, Mali and Senegal. The seasonal variability of O$_3$ is well marked: O$_3$ levels during the wet season are higher than in the dry season (Table 3). On most sites, from January to May, the O$_3$ concentrations gradually increase to finally reach annual peaks at the start of the wet season (May-June-July). The mean annual cycle of monthly O$_3$ concentrations at Ba, Ka, Ag (Fig. 4a), Bb and Da (Fig. 4b) are obtained from averages of monthly in situ measurements over the whole studied period. The annual distribution is similar to the regional rainfall pattern. O$_3$ concentrations decrease as the rainy season progresses, but remain at higher levels compared to the dry season. At dry savanna sites, monthly average surface O$_3$ concentrations range from 6.1±2.4 ppb to 14.5±2.6 ppb during the dry season, and from 13.9±5.1 ppb to 19.4±3.9 ppb during the rainy season (Table 3). From dry to wet season, O$_3$ levels increased from 18.7% to 68.5%. The annual O$_3$ concentrations ranged from 10.5±5.4 ppb at Ka to 14.8±4.3 ppb at Bb (Table 3).

The high O$_3$ concentrations observed at the start of the rainy season are due to soil humidification during this period, which generates biogenic NO emissions pulses in the region. Indeed, the accumulated nitrogen in soils (in the form of ammonium and nitrate ions) from traditional agricultural practices, such as grazing, manure spreading and decomposition of crop residues

(Delon et al., 2015; Laville et al., 2005) is released to the atmosphere when the first rains fall on dry soils. Bacterial nitrification is thus activated, leading to nitrogen consumption and consequent release of large pulses of NO (Adon et al. 2010; Delon et al., 2015; Jaegle et al., 2004; Laville et al., 2005; Ludwig et al., 2001; Ossohou et al., 2019). During the wet season, the decrease in $O_3$ levels may be attributed to a decrease in NOx concentrations. Indeed, soil mineral N is used by plants during their root growth phase, and is therefore less available for the production of NO to be released to the atmosphere (Homyak et

al., 2014). On the Fig. 5, which presents the monthly variation of NOx and VOCs (natural and anthropogenic emissions) in dry savanna, biogenic NO fluxes (Fig 5e, f and g) show a bell-shaped variation, peaking in August (wet season). We observe a good dependence of $O_3$ with NO ($0.73 < r < 0.92$) at Ba, Ka and Ag in the presence of high relative humidity and precipitation ($0.64 < r < 0.95$) (Table 4), that is agreement with the high values of $O_3$ observed in wet season over these sites. Monthly profile of BVOC fluxes (isoprene, α pinene and β pinene) in dry savanna (Fig. 5) shows a maximum at the end of the dry

season/beginning of wet season at Ba, Ka and Ag (Fig. 5e, f and g), or during the wet season at Bb and Da (Fig. 5h and i). Isoprene fluxes are more obvious at Da ($214.2 \pm 30$) ng.m$^{-2}$.s$^{-1}$ whereas α pinene, and β pinene exhibit larger values at Ka site ($11,2 \pm 1,8$; $5.2 \pm 0.8$ ng.m$^{-2}$.s$^{-1}$ respectively). The fluxes of β pinene ($0.70 < r < 0.79$) and isoprene ($r = 0.79$) correlates well with $O_3$ respectively at (Ba, Ka, Ag) and (Bb, Da) under the influence of the humidity, rainfall in Mali and Niger and the temperature, radiation and humidity ($0.50 < r < 0.76$) in Senegal (Table 4).

These observations in dry savanna are confirmed by Stewart et al. (2008) who correlated $O_3$ production in the Sahel during the wet season with high NOx concentrations attributed to biogenic emissions during the AMMA (Analyse Multidisciplinaire de la Mousson Africaine) campaign. In dry savanna, Oluleye et al. (2013) estimated that rain was responsible for 62% of the $O_3$ distribution in the West African region, excluding the precursors NO, CO and hydrocarbons, as also illustrated in our results. Saunois et al. (2009) have shown that soil NOx emissions, combined with the northward advection of volatile organic

compounds (VOCs), play a key role in $O_3$ production in dry savanna regions. This large-scale impact of biogenic emissions has also been verified by Williams et al. (2009), who estimate that 2-45% of tropospheric $O_3$ over equatorial Africa may originate from NOx emissions from African soils. All these works are in agreement with the results of this study. Monthly variation in anthropogenic NOx and VOC emissions (Fig. 5a, b, c and d) indicates during the wet season, NOx and VOC fluxes are very low. On the other hand, maxima are observed in the dry season with the highest emissions found in Ka and could be

the cause of ozone production in the dry season. Indeed, the monthly averaged biomass combustion emissions (GFED4) over the 18-year period (1998-2015) in the Sahel show that Ka is significantly affected by the biomass combustion source in November (Ossohou et al., 2019).

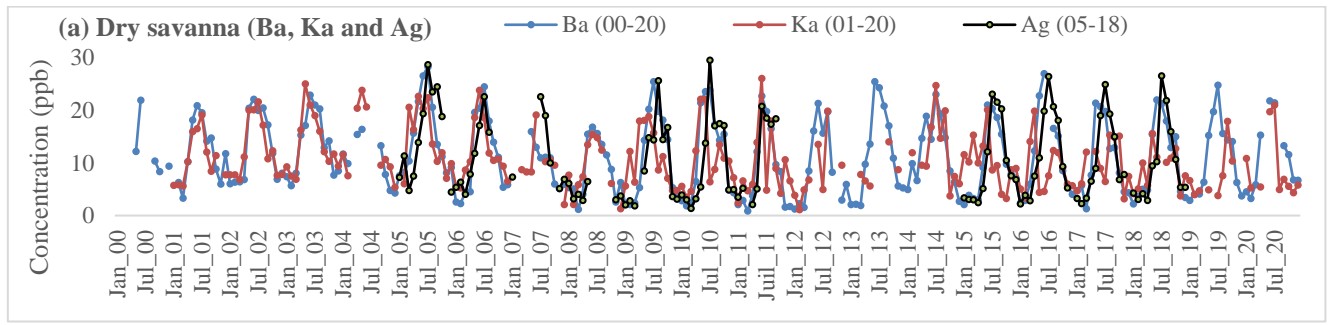

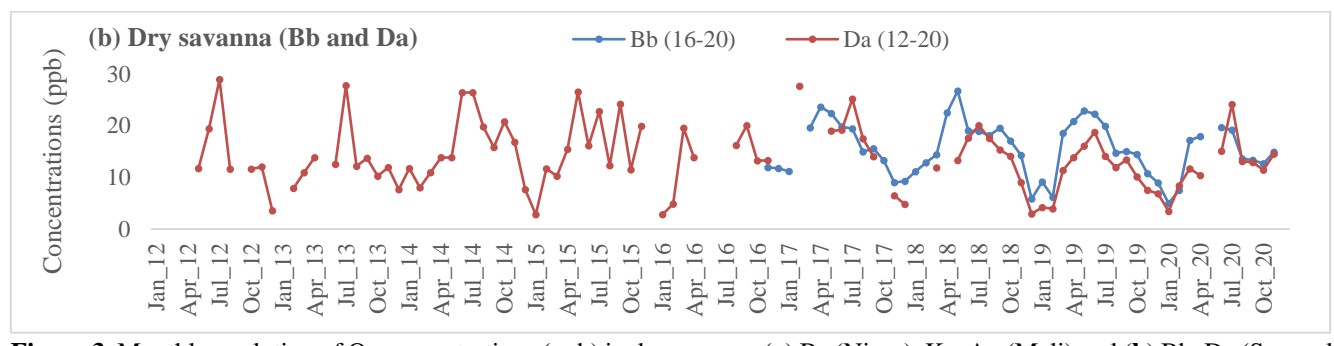


**Figure 3.** Monthly evolution of O₃ concentrations (ppb) in dry savanna (**a**) Ba (Niger), Ka, Ag (Mali) and (**b**) Bb, Da (Senegal).

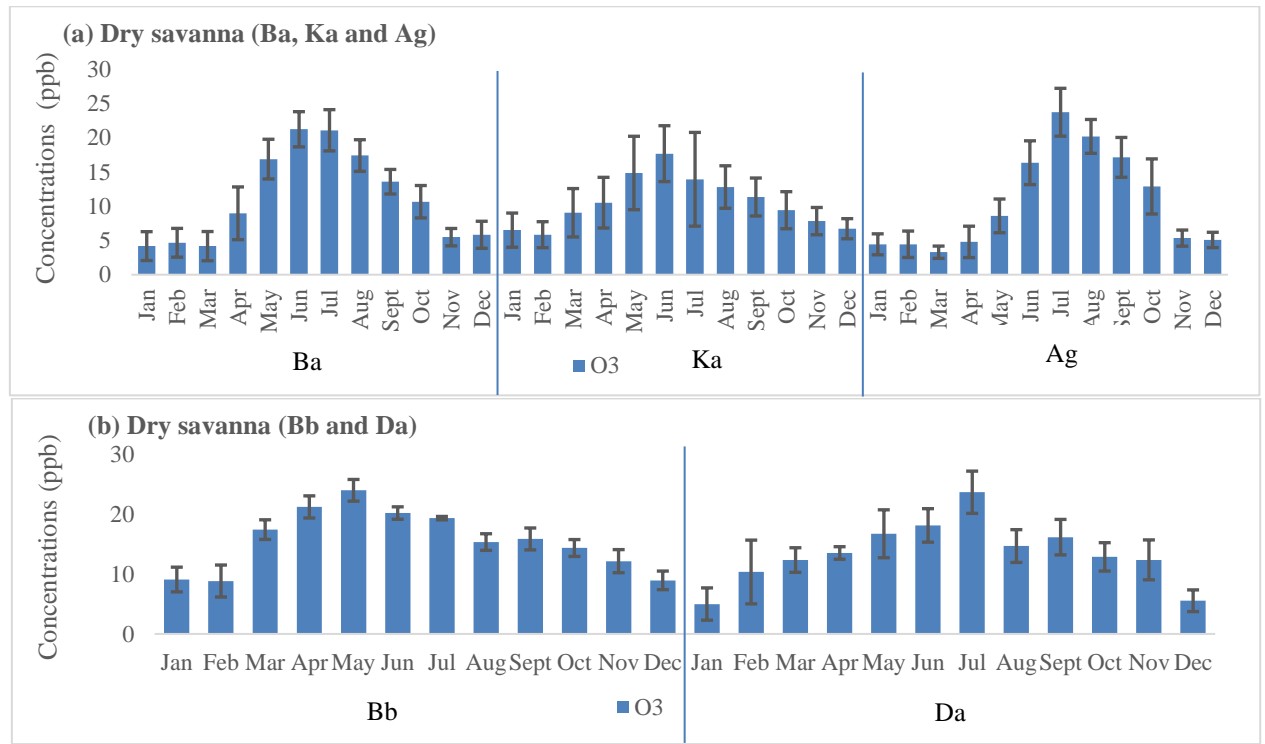

**Figure 4.** Mean monthly averages of O₃ concentrations (ppb) in dry savanna (**a**) Ba (Niger), Ka, Ag (Mali) and (**b**) Bb, Da (Senegal). Mean monthly averages are calculated from the long ozone data series of Fig. 3. Bars represent mean absolute deviation.

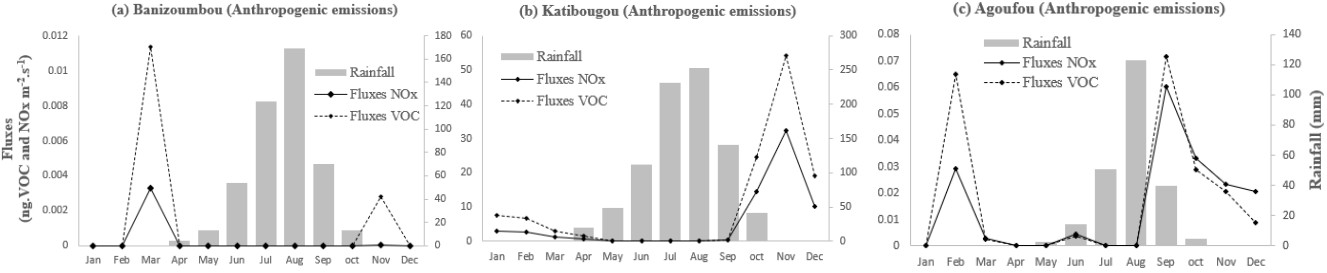

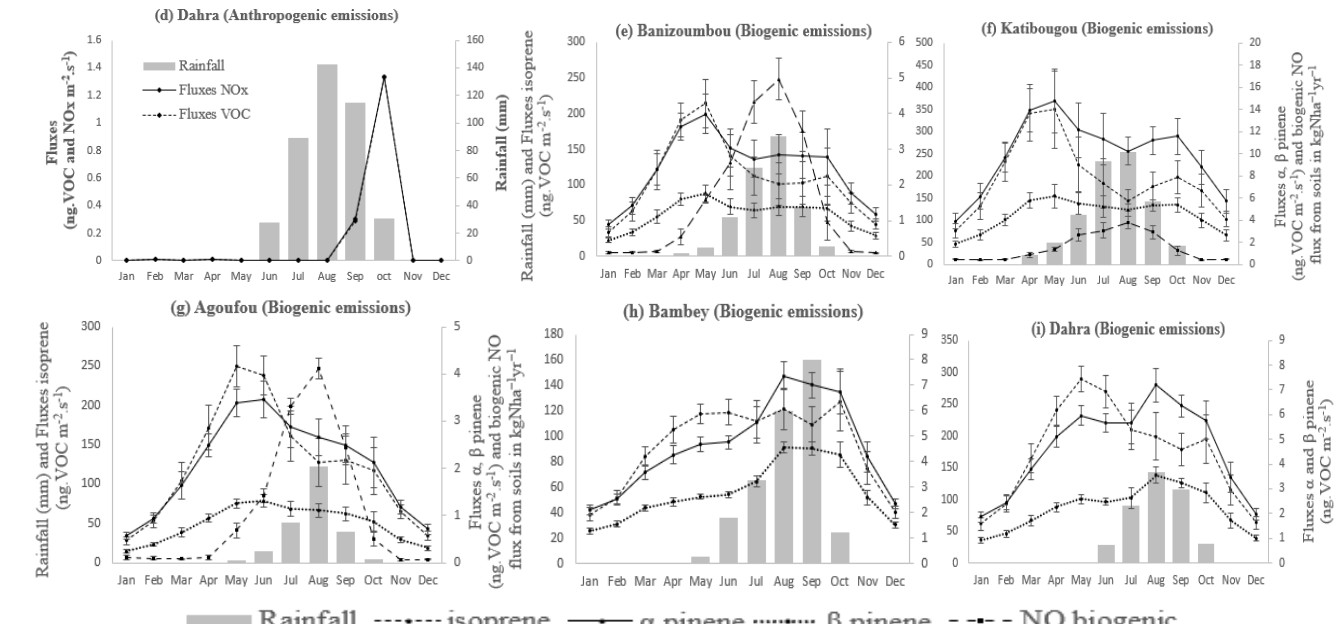

Figure 5. Mean monthly fluxes of natural and anthropogenic NOx and VOC estimated by the MEGAN and GFED4 inventories for 0.25° x 0.25° grid cells centered on each of dry savanna



**Table 3**. Minimum, maximum and average of monthly, annual and seasonal O$_3$ concentrations at all sites (1995-2020)

| Ecosystem | | Monthly | | Annual | Dry season | | | Wet season | | |
|---|---|---|---|---|---|---|---|---|---|---|
| | | min | max | Avg | min | max | Avg | min | max | Avg |
| Dry savanna | Ba | 0.9 | 28.3 | 11.2±6.9 | 4.2±2.7 | 16.9±3.7 | 7.6±2.9 | 13.6±2.5 | 21.2±3.5 | 18.3±3.2 |
| | Ka | 1.2 | 26.1 | 10.5±5.4 | 5.8±2.6 | 14.9±6.6 | 8.8±3.7 | 11.3±3.8 | 17.7±5.4 | 13.9±5.1 |
| | Ag | 1.4 | 29.5 | 10.5±7.3 | 3.3±1.1 | 12.9±4.6 | 6.1±2.4 | 16.4±3.8 | 23.7±4.6 | 19.4±3.9 |
| | Bb | 4.96 | 26.7 | 14.8±4.3 | 8.8±3.5 | 24.0±2.4 | 14.5±2.6 | 15.4±1.9 | 20.2±1.4 | 17.7±1.6 |
| | Da | 2.8 | 29.0 | 13.9±6.3 | 5.0±3.8 | 16.7±5.4 | 11.1±4.0 | 14.7±3.1 | 23.7±4.8 | 18.2±4.0 |
| Wet savanna | La | 4.25 | 20.6 | 10.8±3.3 | 9.9±1.4 | 15.1±2.3 | 13.5±2.3 | 6.7±1.1 | 12.2±1.9 | 9.0±1.5 |
| | Dj | 3.3 | 24.8 | 13.5±4.8 | 10.8±4.5 | 18.7±2.7 | 14.1±4.0 | 9.0±2.0 | 18.4±2.4 | 13.2±2.8 |
| Forest | Zo | 1.2 | 11.1 | 5.2±2.1 | 7.1±2.1 | 7.8 ±1.6 | 7.5±2.1 | 3.5±1.3 | 6.6±1.9 | 4.6±1.6 |
| | Bo | 1.5 | 8.3 | 3.9±1.1 | 4.0±1.0 | 5.2±1.2 | 4.7±1.4 | 2.8±1.0 | 5.4±1.0 | 3.7±1.0 |
| Agricultural or semi-arid savanna | Mb | 10.5 | 30.2 | 19.9±4.7 | 13.8±3.4 | 25.7±5.7 | 20.9±4.0 | 14.1±5.0 | 22.5±2.7 | 18.5±3.9 |
| | Sk | 6.3 | 64.1 | 22.8±7.3 | 23.0±9.6 | 30.2±6.0 | 25.9±7.3 | 14.5±1.9 | 29.2±5.6 | 20.3±5.4 |
| | Af | 3.2 | 55.5 | 26.9±6.3 | 19.9±6.2 | 31.2±7.6 | 24.5±6.3 | 23.8±4.9 | 34.4±7.4 | 29.0±7.3 |
| | CP | 3.3 | 67.4 | 26.8±6.2 | 17.3±5.3 | 30.4±6.0 | 23.4±5.6 | 25.9±6.3 | 32.1±5.7 | 29.8±6.1 |
| | LT | 9.0 | 86.6 | 30.8±8.0 | 24.7±7.9 | 40.1±10.8 | 32.0±8.9 | 21.3±5.2 | 36.0±9.1 | 28.3±8.2 |

### 3.2.1.2 Wet savanna and forest

Figure 6 presents the mean monthly surface O$_3$ concentrations in Dj, La (Fig. 6a), Zo and Bo (Fig. 6b). O$_3$ concentrations
present a seasonality during the year. The maximum of the data series in La is 20.6 ppb in March (dry season) and the minimum

is 4.3 ppb in October (wet season). In Dj, the $O_3$ levels are higher than in La. The monthly highest value recorded in Dj is 24.8 ppb in April (start of the wet season). At the forested ecosystems sites, $O_3$ concentrations are lower than in dry, wet and semi-agricultural/semi-arid savannas (Table 3). In Zo and Bo, the highest annual peaks are found in February (11.1 ppb and 8.3 ppb respectively). Monthly averages in the dry season ranged from $4.7\pm1.4$ ppb (Bo) to $14.1\pm4.0$ ppb (Dj), and in the wet season from $3.7\pm1.0$ ppb (Bo) to $13.2\pm2.8$ ppb (Dj) (Table 3). The $O_3$ mean annual cycle is shown in Fig. 7.

The high $O_3$ concentrations in dry season in these two ecosystems could be related to the biomass burning source, which is generally recorded during the months of December-February in rural tropical environments and BVOC emissions. Indeed, in wet savannas (La, Dj), and forest (Zo) (Fig. 8), NOx and VOC anthropogenic fluxes reach their maxima during the dry season. The mean flux estimates are respectively 14.5; 24.2; 4.9 ng.m$^{-2}$.s$^{-1}$ for NOx and 26.8; 29.4; 5.5 ng.m$^{-2}$.s$^{-1}$ for anthropogenic VOCs at La, Dj and Zo (Fig 8a, b and c). BVOC maxima fluxes are obtained at the end of the dry season/beginning of the wet season (Fig. 8e, f, g and h). A drop in these fluxes is then observed during the wet season. Strong Pearson correlations are observed between $O_3$, NOx and the VOCs ($0.49 < r < 0.92$) (Table 4). Temperature and radiation are also well correlated with $O_3$ ($0.54 < r < 0.89$) in these two ecosystems. These results are corroborated by the literature. Indeed, radiation and humidity facilitate the propagation of radical chain reactions and the production of hydroxyl radicals (OH) at these sites (Graedel and Crutzen, 1993). According to several authors, $O_3$ levels tend to increase under warm, sunny conditions favorable to photochemical $O_3$ production (Hamdun and Arakaki, 2015; Morakinyo et al., 2020). Moreover, Aghedo et al. (2007), Mari et al. (2011), Saunois et al. (2009) and Saxton et al. (2007) have reported that vegetated areas emit large quantities of biogenic organic compounds that influence $O_3$ production in the presence of light and temperature. Others authors such as Abbadie (2006), Adon et al. (2010), Galanter et al. (2000) and Tsivlidou et al. (2023) have linked the high $O_3$ concentrations recorded in the dry season to the presence of NOx emitted by biomass combustion in the wet savanna (Gulf of Guinea). According to Adon et al. (2010), Baldy et al. (1996), Clain et al. (2009), Cros et al. (1992), Hamdun and Arakaki (2015), Martins et al. (2007), Oluleye et al. (2013) the biomass burning is likely to contribute significantly to $O_3$ production through precursor emissions (NOx and CO) in the dry season (wet savanna) with nearly 30% to 80% of the savanna ground surface burnt annually between December and February. The high $O_3$ concentrations measured in Tranquebar (India) have been linked to increased emissions of NOx and other precursors from various sources (Debaje et al., 2003). Compared to wet and dry savannas, forest sites recorded the lowest $O_3$ amounts due to significant dry deposition of $O_3$ on the ground, on foliage and trees (Mari et al., 2011; Rummel et al., 2007; Saunois et al., 2009). Tropical forests are shown to be important $O_3$ sinks. A strong gradient of $O_3$ between forest and dry savanna in West Africa has been observed from aircraft measurements (Saunois et al., 2009).

In Bo, high NOx and VOC fluxes are observed in the wet season (Fig. 8d), unlike in Zo (Fig. 8c) and corroborated by Ossohou et al. (2019) over the period 1998-2015. The source of these recorded anthropogenic during this period of year at Bo emissions could be biomass combustion. According to the work of Sauvage et al. (2005), the period from August to September corresponds to a peak in biomass burning activity in the southern African countries (Mozambique, Zimbabwe, South Africa). Moving air masses over Central Africa via the northern edge of the continental anticyclone could explain such high emissions at Bo in August-September.

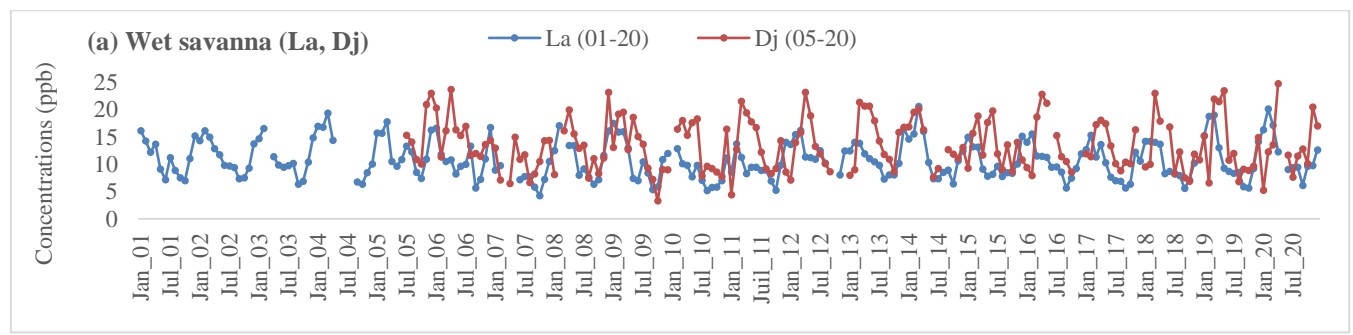

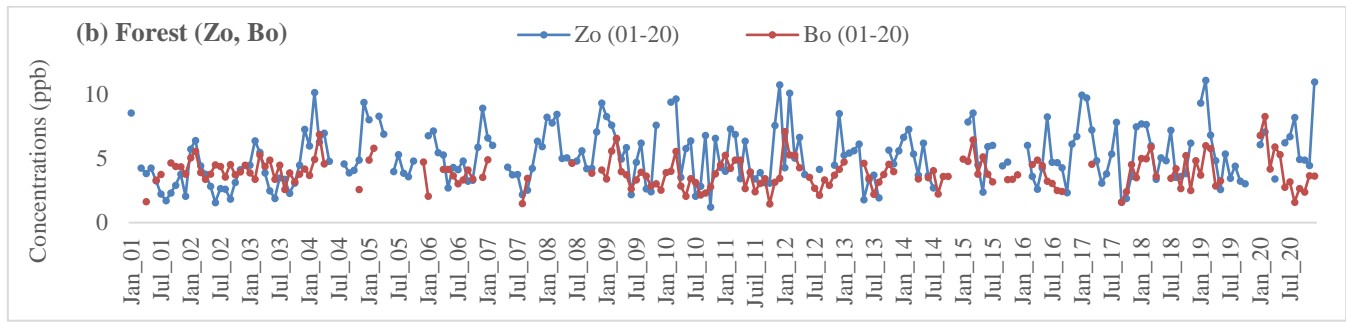

**Figure 6.** Monthly evolution of O$_3$ concentrations (ppb) in (**a**) humid savanna, La (Cote d'Ivoire) and Dj (Benin) and (**b**) in forest, Zo (Cameroon) and Bo (Congo).

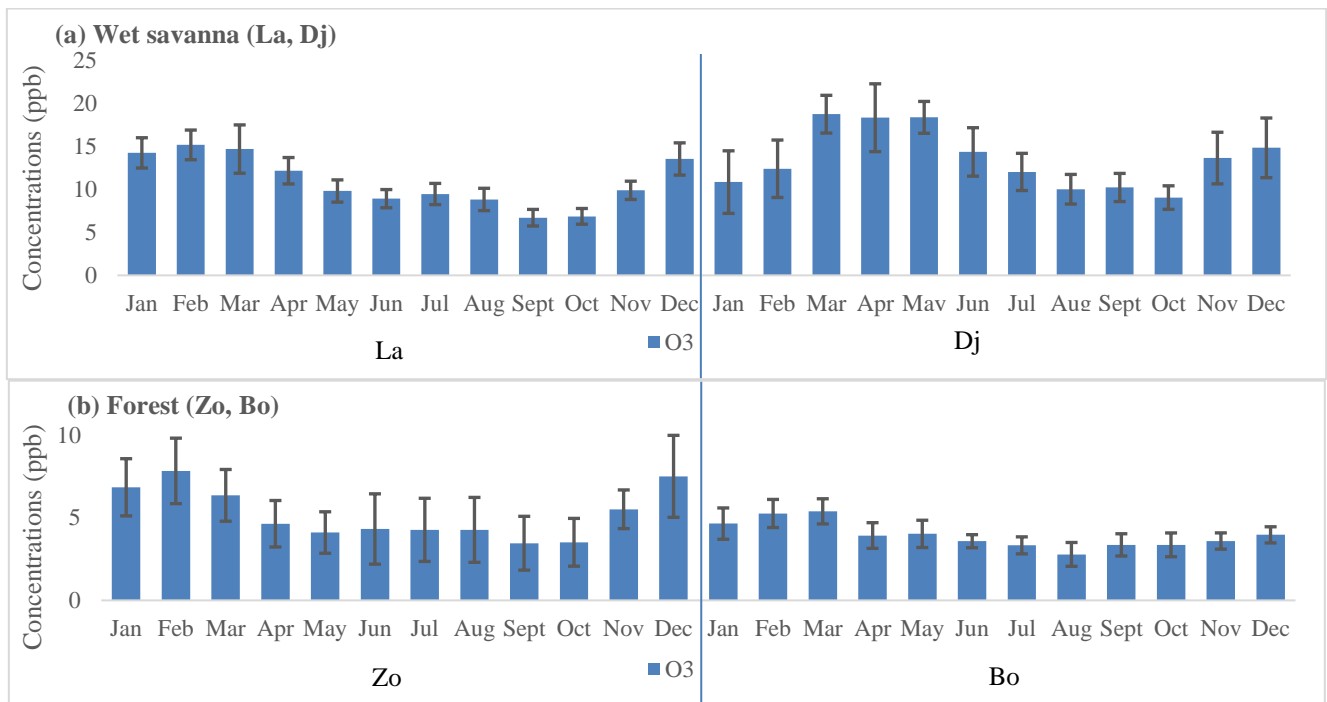

**Figure 7.** Mean monthly averages of O$_3$ concentrations (ppb) a) in humid savanna, La (Cote d'Ivoire) and Dj (Benin) and b) in forest, Zo (Cameroon) and Bo (Congo). Mean monthly averages are calculated from the long ozone data series of Fig. 6.

Bars represent mean absolute deviation.

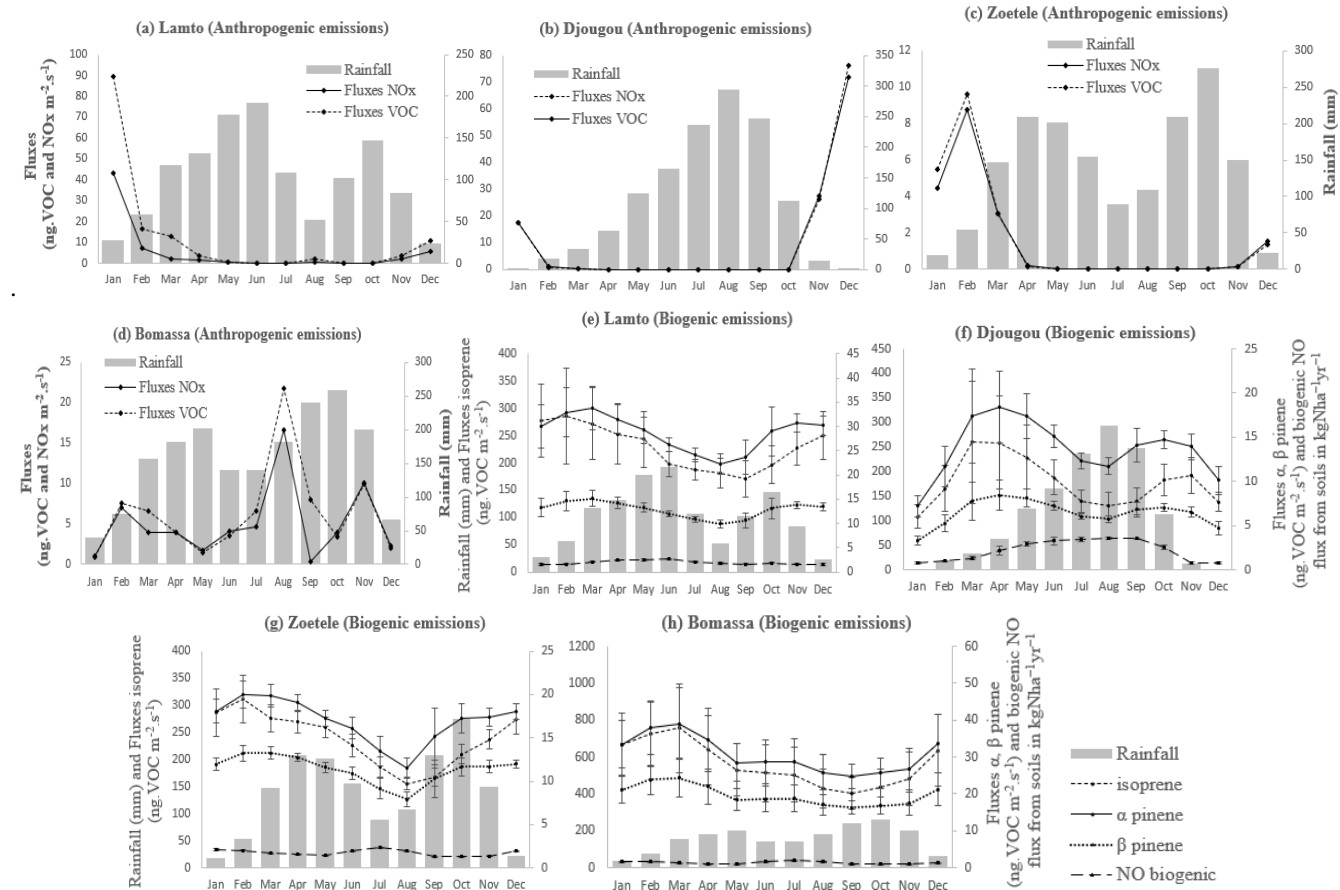

**Figure 8.** Mean monthly fluxes of natural and anthropogenic NOx and VOC estimated by the MEGAN and GFED4 inventories for 0.25° x 0.25° grid cells centered on each of wet savanna and forests.

**Table 4.** Correlation r between $O_3$, its precursors and meteorological variables at different sites. Blank spaces in the table indicate the absence of data on this site for the precursor concerned over the study period

| Ecosystem | Dry savanna | | | | | Wet savanna | | Forest | | Agricultural/semi-arid savanna | | | | |
|---|---|---|---|---|---|---|---|---|---|---|---|---|---|---|
| Sites | Ba | Ka | Ag | Bb | Da | La | Dj | Zo | Bo | Mb | LT | CP | Af | Sk |
| | $O_3$ | | | | | | | | | | | | | |
| NO biogenic | 0.85 | 0.73 | 0.92 | - | - | -0.15 | -0.24 | 0.43 | $-10^{-3}$ | - | - | - | - | - |
| NOx_C | -0.33 | -0.43 | 0.04 | - | 0.001 | 0.49 | 0.059 | 0.80 | -0.33 | 0.31 | 0.37 | -0.67 | 0.62 | 0.61 |
| VOC_C | -0.40 | -0.47 | -0.003 | - | $-5.10^{-4}$ | 0.54 | 0.05 | 0.79 | -0.40 | 0.31 | 0.60 | -0.87 | 0.52 | 0.65 |
| isoprene | 0.51 | 0.54 | 0.46 | 0.79 | 0.78 | 0.92 | 0.81 | 0.80 | 0.92 | 0.42 | -0.39 | -0.91 | 0.29 | -0.64 |
| α pinene | 0.67 | 0.76 | 0.66 | 0.45 | 0.77 | 0.74 | 0.64 | 0.63 | 0.90 | 0.36 | -0.45 | -0.86 | 0.29 | -0.67 |
| β pinene | 0.70 | 0.79 | 0.72 | 0.34 | 0.72 | 0.70 | 0.56 | 0.60 | 0.89 | 0.33 | -0.48 | -0.85 | 0.27 | -0.71 |
| Temperature | 0.45 | 0.47 | 0.42 | 0.51 | 0.76 | 0.72 | 0.69 | 0.68 | 0.89 | 0.47 | -0.46 | -0.90 | 0.49 | -0.62 |
| Humidity | 0.82 | 0.64 | 0.95 | 0.52 | 0.70 | -0.85 | -0.19 | -0.89 | -0.79 | -0.61 | -0.79 | -0.68 | 0.1 | -0.80 |
| Rainfall | 0.74 | 0.64 | 0.75 | 0.15 | 0.51 | -0.48 | -0.39 | -0.76 | -0.54 | -0.18 | -0.49 | 0.74 | 0.53 | -0.69 |

| Radiation | 0.16 | 0.15 | 0.26 | 0.75 | 0.69 | 0.73 | 0.54 | 0.65 | 0.87 | 0.21 | -0.19 | -0.76 | 0.71 | -0.43 |

### 3.2.1.3 Agricultural and semi-arid savanna

Figure 9 presents the monthly evolution of surface $O_3$ concentration in agricultural site (Mb) and semi-arid savanna sites (LT, CP, Sk and Af). At Mb site, monthly $O_3$ concentrations do not exceed 30.2 ppb (Table 3). At the CP, LT, Sk and Af sites, $O_3$ levels are almost twice as high as in West African sites. The mean annual cycle of $O_3$ concentrations (Fig. 10a and b) shows that at Mb, $O_3$ levels are almost similar between seasons (Table 3). In southern African ecosystems, dry-season $O_3$ concentrations are the highest at LT and Sk. The annual averages are around 19.9±4.7 ppb at Mb; 22.8±7.3 ppb at Sk; 26.9±6.3 ppb at Af; 26.8±6.2ppb at CP and 30.8±8.0ppb at LT (Table 3).

These $O_3$ levels observed could be associated at the combustion sources and natural emissions. Indeed, on the Fig. 11a, we observe NOx and VOC fluxes are quantifiable in February (dry season) at Mbita. Based on the analysis of burned surface areas, Bakayoko et al. (2021) indicated that Mb is strongly influenced by biomass burning from northern and southern sides during both dry seasons. High $O_3$ levels measured at Mb site during the dry season show similar values as in Nairobi, Kenya, during the same months (Kimayu et al., 2017). At the South African sites, the Fig. 11b, c, d and e shows the mean fluxes of anthropogenic emissions vary from 0.3 ng.m$^{-2}$.s$^{-1}$ (LT) to 10.2 ng.m$^{-2}$.s$^{-1}$ (Sk) for NOx and from 2.9 ng.m$^{-2}$.s$^{-1}$ to 11.8 ng.m$^{-2}$.s$^{-1}$ (Sk) for VOCs with the maxima recorded in dry season. As for the BVOC emissions (Fig. 11 g, i and j), more specifically at LT, Af and Sk, the highest values are reached in the wet season. At CP (Fig. 11h), the maximum emissions are measured in the dry months of January/February. The calculations of correlation indicate ozone is linked to anthropogenic combustion sources at LT, Af and Sk and are anti-correlated with temperature, humidity, and radiation (- 0.90 < r < - 0.43) at LT, CP and Sk (Table 4). High $O_3$ concentrations are therefore measured at these sites during the driest and coldest months (Swartz et al., 2020b). Except at Af where $O_3$ has a weak link with BVOC, the increase of isoprene, α pinene, β pinene emissions rate is positively correlated with $O_3$ decrease at the others sites. At the CP and Af sites, rainfall and $O_3$ are correlated (0.53 < r < 0.74) and ozone production could therefore also be linked to microbial activity of soils on these two sites. At CP site during the same study period, Swartz et al. (2020b) emphasised higher $NO_2$ concentrations were attributed to increased microbial activity in the wet season and $O_3$ seasonal pattern corresponded to the $NO_2$ seasonality, which was attributed to their related chemistry. The importance of humidity and temperature in $O_3$ photochemistry observed at almost all South African sites has been highlighted by Balashov et al. (2014) and Laban et al. (2018, 2020).

At southern African sites, $O_3$ levels could be also attributed to a combination of regional and local influences, including emissions from industrial, vehicular and domestic biomass combustion to biomass combustion events in sub-Saharan Africa (Mozambique, Zambia, Zimbabwe and Angola) recirculated by anticyclonic air mass processes (Baldy et al., 1996; Laban et al., 2018; Martins et al., 2007; Swap et al., 2003; Tiitta et al., 2014). Biomass combustion is considered as a major source of $O_3$ precursors in South Africa (Ngoasheng et al., 2021; Vakkari et al., 2013) and in Southern Africa (Heue et al., 2016) and may explain the $O_3$ levels observed in dry season at LT and Sk. The high $O_3$ levels in the wet season (at Af and CP) due to soil microbial activity (Swartz et al., 2020a) could be also explained by long-range transport of air pollutants emitted from the industrialized Highveld region (Abiodun et al., 2014; Ojumu, 2013). During the austral winter, $O_3$ concentrations in the boundary layer are higher (e.g. at CP and Af) due to a systematic increase in $O_3$ precursors from households, combustion for space heating (Bencherif et al., 2020; Laban et al., 2018; Lourens et al., 2011; Oltmans et al., 2013; Swartz et al., 2020b). The high concentrations measured at Af could also be due to industrial activities located near this site (Lourens et al., 2011).

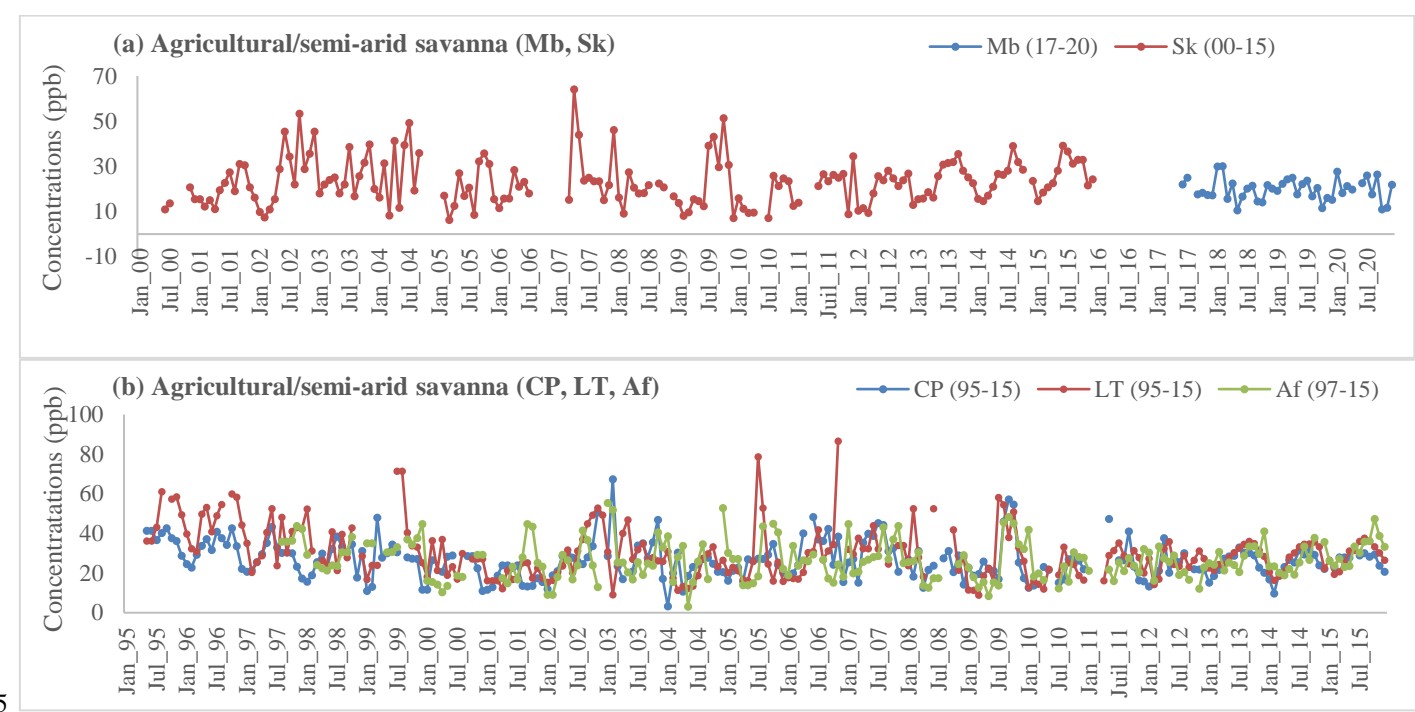

**Figure 9.** Monthly evolution of O₃ concentrations (ppb) in Agricultural/semi-arid savanna (**a**) Mb (Kenya) and Sk (South Africa) and (**b**) CP, LT and Af (South Africa).

**Figure 10.** Mean monthly averages of O₃ concentrations (ppb) in Agricultural/semi-arid savanna (**a**) Mb (Kenya) and Sk (South Africa) and (**b**) CP, LT and Af (South Africa). Mean monthly averages are calculated from the long ozone data series of Fig. 9. Bars represent mean absolute deviation.


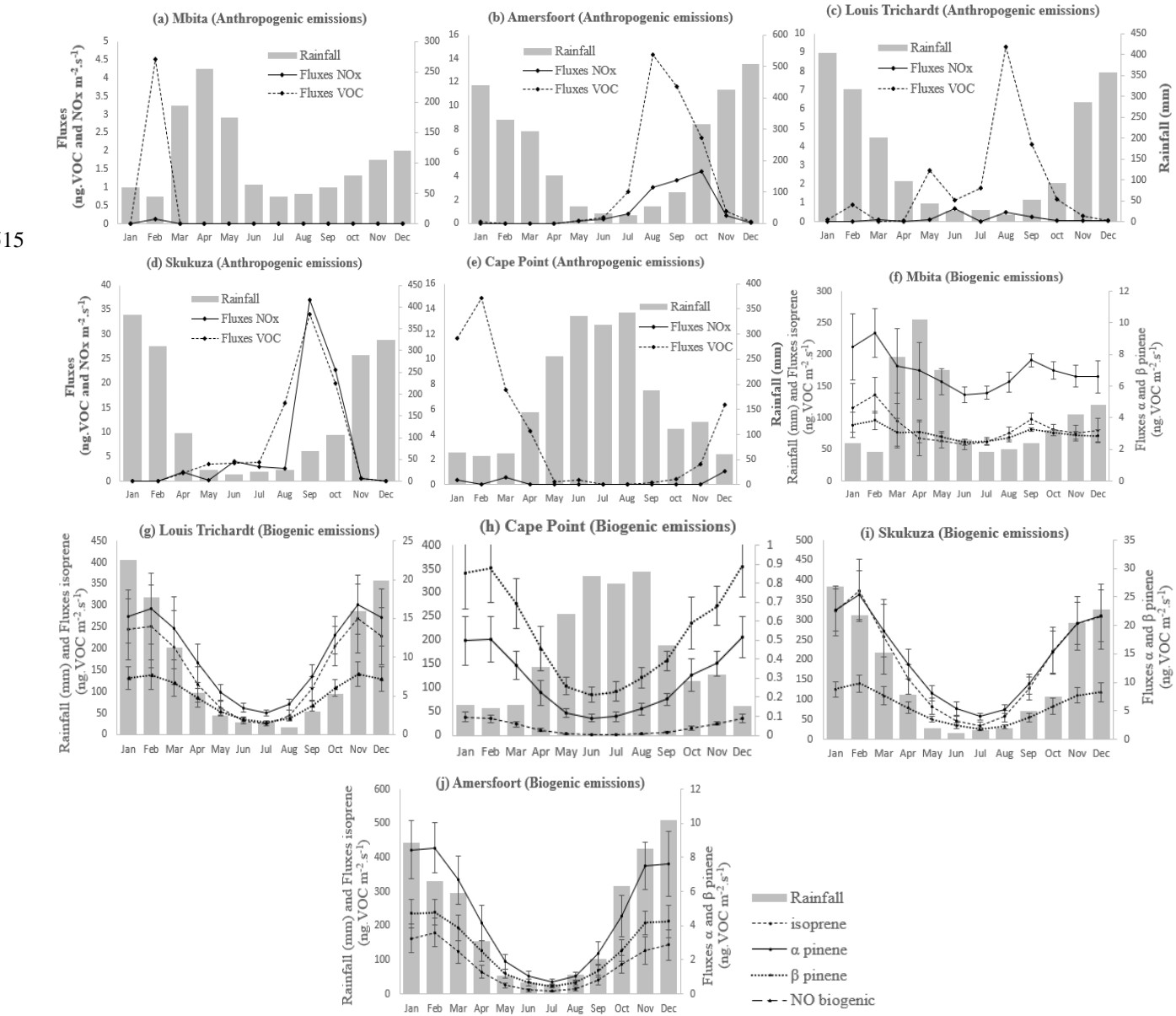

**Figure 11.** Mean monthly fluxes of natural and anthropogenic NOx and VOC estimated by the MEGAN and GFED4 inventories for 0.25° x 0.25° grid cells centered on each of agricultural/semi-arid savanna sites

### 3.2.2 O₃ levels in Africa, set in a global context

O₃ concentrations measured at the 14 studied site are mapped on a seasonal and annual scale (Fig. 12).

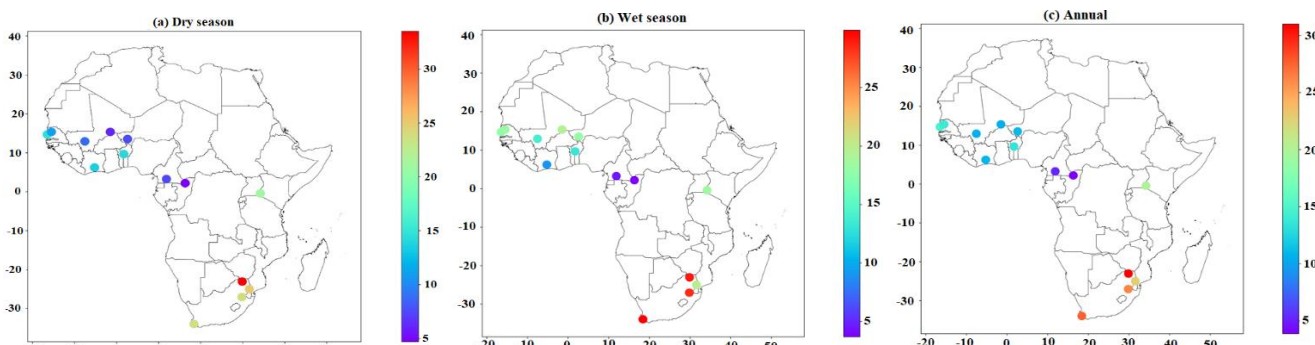

**Figure 12.** Seasonal and annual mapping of O₃ concentration levels in (**a**) Dry season (Ba, Ka, Ag, Da and Bb: October to May ; La and Dj: November to March ; Zo : December to February and July to August ; Bo: December to February ; Mb: June
to October and January to February ; LT, Sk and Af : April to September; CP: October to March), (**b**) Wet season (Ba, Ka, Ag, Da and Bb: June to September; La and Dj: April to October ; Zo : March to June and September to November ; Bo: March to November ;  Mb: March to May and November to December ; LT, Sk and Af : October to March ; CP : April to September) and (**c**) Annual over the 14 studied sites.

We compared African ozone levels related in this study with studies carried out in Africa and around the world over the last 20 years (Table 5, Fig. 13). The bibliographical synthesis takes into account studies where data measurement methodology has been clearly described. Sites where concentrations have been measured by passive samplers are listed. We have identified among others, sites in Nepal, North America, North-Eastern Europe, Asia and Africa. Figure 13 focuses more on O₃ monitoring studies in Africa.


**Table 5.** O₃ concentrations at various sites worldwide as reported in literature

| Sites | Type | Period | O₃ (ppb) | References |
|-------|------|--------|----------|------------|
| India, Tranquebar | Rural | May 1997 -Oct 2000 | 17±7 - 23 ± 9 | Debaje et al. (2003) |
| Sweden, Malmö (20 sites) | Rural | (16–24 Apr 2012 ; 28 May–4 Jun; 20–27 Aug 2012 | 37.0±5.4 | Hagenbjörk et al. (2017) |
| | Urban | | 35.0±3.9 | |
| | Traffic | | 33.6±3.5 | |
| Sweden, Umeå (20 sites) | Rural | | 27.7±8.4 | |
| | Urban | | 26.7±7.3 | |
| | Traffic | | 25.2±6.9 | |
| China, Waliguan mountain | Rural | Sept 99-May 2001 | 44.9 | Carmichael et al. (2003) |
| Taiwan, Shui-Li | Rural | | 25.0 | |
| Malaysia, Tanah Rata | Rural | | 16.0 | |
| Indonesia, Bukit Kototabang | Rural | | 10.7 | |
| Inde, Agra | Rural | | 30.8 | |
| Argentina, Isla Redonda | Rural | | 15.9 | |
| Brazil, Arembepe | Rural | | 19.2 | |
| Turkey, Camkoru | Rural | | 35.4 | |

| | | | | |
|---|---|---|---|---|
| Arab Emirates, Al-Ain | Rural | Apr 2005-Apr 2006 | 8.7 | Salem et al. (2009) |
| | Industrial | | 5.9 | |
| | Traffic | | 4.4 | |
| | Commercial | | 5.9 | |
| | Residential | | 7.3 | |
| American Samoa Island, Tutuila | Urban and Rural | 1990-2015 | 13.7 ± 1.0 | Lu et al. (2018) |
| Chili, El Tololo | | | 32.0 ± 1.3 | |
| South Africa, Cape point | | | 24.3 ± 1.1 | |
| Australia, Cape grim | | | 24.9 ± 0.8 | |
| New Zealand, Baring head | | | 21.4 ± 1.6 | |
| Syowa | | | 25.2 ± 0.9 | |
| Neumayer-G | | | 24.3 ± 1.5 | |
| Arrival heights | | | 25.9 ± 1.5 | |
| South Pole | | | 28.4 ± 1.7 | |
| Barrow Atmospheric Baseline Observatory | Remote site | 1973-2015 | 15-44 | Cooper et al. (2020) |
| Mauna Loa Observatory (MLO) | Remote site | 1973-2015 | 26-65 | |
| American Samoa Observatory | Remote site | 1973-2015 | 5-20 | |
| South Pole Observatory | Remote site | 1973-2015 | 17-40 | |
| China | Rural site | 2014-2017 | 34 | Dufour et al. (2021) |
| Central East China | | | 36 | |
| Beijing–Tianjin–Hebei region | | | 39 | |
| Yangtze River Delta | | | 35 | |
| Pearl River Delta | | | 31 | |
| North America, Europa and Est Asia (Korea et Japan) | 3136 Rural sites | 2010-2014 | 0-56 and more | Gaudel et al. (2018) |
| North America, Europa and Est Asia | 3348 Rural sites and 1453 urban sites | 2010-2014 | 0-100 and more | Fleming et al. (2018) |
| North America, Europa and Est Asia | Rural site | 1996-2005 | 15-55 | Young et al. (2018) |
| North America, Europa and Est Asia | Rural and urban site | 2010-2014 | 10-60 | Schultz et al. (2017) |
| Eastern North America | Rural site | 2000-2014 | 26-38 | Chang et al. (2017) |
| | Urban site | | 28-38 | |

Several sites reported in Table 5 and Fig. 13 are exposed to high $O_3$ concentrations. These different levels observed in Africa
and around the world are in most cases above values displayed in this study, with the exception of sites in southern Africa.
Indeed, in Africa particularly, the earlier studies investigated have mentioned high ozone concentrations measured in southern
Africa. Laban et al. (2018) have observed the elevated surface ozone ($O_3$) concentrations over four sites in South Africa
(Botsalano (2006-2008), Marikana (2008-2010), Welgegund (2010-2015) and Elandsfontein (2009- 2010)) with an annual
mean ranging around from 5 to 70 ppb. The temporal $O_3$ patterns observed at the four sites resembled typical trends for $O_3$ in
continental South Africa, with $O_3$ concentrations peaking in late winter and early spring (Laban et al., 2018). The assessment
of long-term seasonal and inter-annual trends of a 21-year ozone passive sampling (monthly means) dataset collected at the
Cape Point Global Atmosphere Watch (CPT GAW) station indicated that annal mean ozone level at this coastal area is of 26
ppb (Swartz et al., 2020b) while at Louis Trichardt (1995-2015), Amersfoort (1997-2015) and Skukuza (2000-2015) the level
ranging from 22 ppb to 31 ppb (Swartz et al., 2020a). Over the sites of Botswana (1999-2001) and the Mpumalanga highveld,

Zunckel et al. (2004) emphasized the springtime maximum of $O_3$ concentrations is between 40 and 60 ppb, but reached more than 90 ppb as a mean in October 2000. At the background stations at Cape Point (2000-2002), in Namibia (2000-2002) and areas adjacent to the highveld the maximum concentrations are between 20 and 30 ppb with minimums between 10 and 20 ppb. Ngoasheng et al. (2021) have investigated on the surface ozone concentrations during the period from 2014 to 2015 and 2018 to 2019 over ten sites located at North West in South Africa (8 ppb à 48 ppb). In addition, a more intensive campaign

was conducted in June, July and August 2019 during what, 15 additional sites were also monitored. During the campaign from September 1999 to June 2001 of the newly established WMO/GAW Urban Research Meteorology and Environment (GURME) project, the mean values of ozone concentrations over the sites of Elandsfontein (35.1 ppb), Cape point (24.2 ppb) in South Africa, Tamanrasset in Algeria (33.2 ppb), Mt. Kenya (31.5 ppb) were evaluated and indicated high values over many sites (Carmichael et al 2003). Over a period of nine to 11-year (1995-2005) at four remote sites: Louis Trichardt (South Africa),

Cape Point (South Africa), Amersfoort (South Africa) and Okaukuejo (Namibia) in southern Africa, Martins et al. (2007) exhibited a fairly constant high mean value of ozone about 27 ppb throughout the region except for the Louis Trichardt site, with a relatively high 10-year mean of 35 ppb. These values are approximately two times higher compared to Western and Central Africa INDAAF ozone data. The main reason could be the proximity of South Africa sites to $O_3$ precursor sources. They are generally located close to industrial, commercial and residential areas, not far from road traffic, garbage dumps, etc.,

where precursor emissions are high. Some sites may also be influenced by continental air masses containing gaseous pollutants. From 2012 to 2015, Hamdun and Arakaki (2015) measured surface ozone levels at three urban sites (Mapipa, Ubungo, and Posta) and two suburban sites (Kunduchi and Vijibweni) in the city of Dar es Salaam and in the village of Mwetemo, a rural area of Bagamoyo, Tanzania. Ozone levels at suburban (7.9-23.6) ppb sites were generally higher than at urban sites (10.3-18.6) ppb. In the context of the POLCA (Pollution of African Capitals) program, $O_3$ was measured using a passive sampling

technique from Jan. 2008 to Dec. 2009 at Dakar and from Jun. 2008 to Dec. 2009 at Bamako (Adon et al., 2016). The mean annual concentrations of $O_3$ are 7.7 ppb in Dakar and 5.1 ppb in Bamako, respectively. At Abidjan during an intensive campaign within the dry season (15 December 2015 to 16 February 2016), using INDAAF (International Network to study Deposition and Atmospheric chemistry in AFrica) passive samplers exposed in duplicate for 2- week periods (Bahino et al., 2018), the highest $O_3$ concentration measured is at the two coastal sites of Gonzagueville and Félix-Houphouët-Boigny

International Airport located in the southeast of the city, with average concentrations of 19.1 ± 1.7 and 18.8 ± 3.0 ppb, respectively. At urban sites such as Al-Ain, Bamako, Dakar, Abidjan and Cotonou, the low $O_3$ levels are due to the saturated NOx regime observed at these sites, which limits photochemical $O_3$ production (Adon et al., 2013; Bahino et al., 2018; Salem et al., 2009). At most INDAAF sites, concentrations are lower because of their rural characteristics, generally far from anthropogenic sources, and much more influenced by biogenic activities from soils and vegetation. In the framework of IDAF

program, Adon et al. (2010) analysed ozone concentrations from 2000 to 2007 over the sites of Banizoumbou (Niger), Katibougou and Agoufou (Mali), Djougou (Benin), Lamto (Cote d'Ivoire), Zoetele (Cameroon) and Bomassa (Congo). Annual mean O3 concentrations are lower for all ecosystems and range from 4.0±0.4 ppb (Bomassa) to 14.0±2.8 ppb (Djougou) and are the same order of magnitude over period 2000-2020 (INDAAF program) where concentrations ranging from 3.9±1.1 ppb (Bomassa) to 14.8±4.3 ppb at Bambey in Western and Central Africa. Results are fairly illustrative of the various mitigation

or vigilance measures that need to be adopted to ensure the environmental well-being of each ecosystem. The additional efforts must therefore be made, through projects or programs, to densify monitoring networks for polluting gases in general and $O_3$ in particular, especially in Africa, where very few long-term monitoring exist.

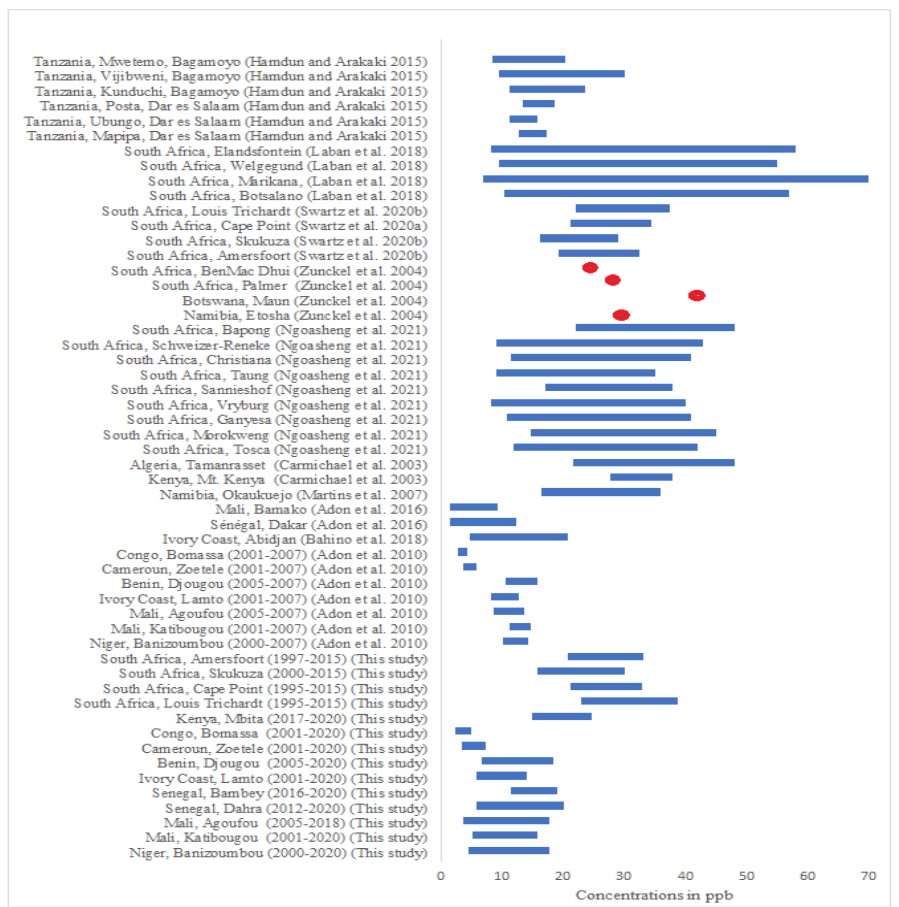

**Figure 13.** Overview of O₃ monitoring studies in Africa. Blue bars represent lower and upper range of means if reported. Red points represent average concentration of O₃.

### 3.3 Annual and seasonal trends of O₃ and its precursors
### 3.3.1 Annual trends

Annual trends in $O_3$ concentrations were calculated all sites (except Bambey and Mbita which do not have 10 years of measurements) according to 95% confidence intervals in Mann-Kendall test in dry savanna, wet savanna, forest and Agricultural/semi-arid savanna (Fig. 14). At the annual scale, Katibougou site in Mali shows a decrease in $O_3$ concentrations around 2.4 ppb decade$^{-1}$ from 2001 to 2020 (pvalue = 0.002) at the 95% confidence level. This trend at Katibougou is confirmed in both the dry and wet seasons. Ozone concentrations decrease by -1.8 ppb decade$^{-1}$ (pvalue = 0.03) in the dry season and -3.3 ppb decade$^{-1}$ (pvalue < 0.01) in the wet season. At the same site, the trend in nitrogen oxide ($NO_2$) over the 1998-2020 period shows a decline in annual concentrations and annual seasonal means. Ossohou et al. (2019) observed a decrease in $NO_2$ concentrations at Katibougou in the wet season (-0.4 ppb decade$^{-1}$) over the period 1998-2015. These downward trends in $NO_2$ could therefore explain the downward trends in $O_3$. At the Banizoumbou site (Niger) a medium certainty for a decrease trend by -0.8 ppb decade$^{-1}$ of $O_3$ concentrations (pvalue = 0.1) is noted at a 95% confidence level. During the dry period, this downward trend is calculated with high certainty with $O_3$ decreasing by -1.5 ppb decade$^{-1}$ (pvalue = 0.04). Calculation of trends on biogenic VOCs at Banizoumbou indicate a decrease in biogenic emissions of alpha pinene (τ = -0.37, p value = 0.020) and beta pinene (τ = -0.39, p value = 0.01). Chen et al. (2018) indicated that trends in global tree cover from 2000 to 2015 have

led to clear decreases, particularly in West Africa with a reduction of around 10% in regional BVOC emissions due to agricultural expansion. At the sites of Agoufou (Mali) and Bambey (Senegal), a low certainty of the $O_3$ concentrations decrease is observed (respectively pvalue = 0.21; pvalue = 0.35). In wet savanna and forests, Lamto (Cote d'Ivoire) and Djougou (Benin)
sites show a downward trend by -0.3 ppb decade$^{-1}$. At 95% confidence interval, there is very low certainty that this trend will occur. In West Africa generally, we note a decrease in surface ozone concentrations even if this trend is not significant at almost all sites. On these remote sites, this decrease could be partly linked to a large decrease in burned area in tropical savannas in Africa, particularly those with low and intermediate levels of tree density (Andela et al., 2017). At Bomassa site (Congo) (pvalue =1), we have no trend contrary to Zoetele (Cameroon) where an increase of $O_3$ concentrations by 0.5 ppb decade$^{-1}$ is
recorded, with a medium certainty. To explain the trends observed at Zoetele, we apply the Kendall rank correlation between $O_3$ concentrations and its precursors. We obtain a significant positive rank correlation at the 95% confidence level. For NOx and anthropogenic VOCs, $\tau = 0.7$ (pvalue= 0.03) while with biogenic VOCs, the correlation varies from 0.54 to 0.69 (pvalue <0.01). In addition, we observe increasing trends for biogenic VOCs. Isoprene increases by 18 ng.m$^{-2}$s$^{-1}$ per decade (pvalue = 0.001), alpha pinene by 1 ng.m$^{-2}$s$^{-1}$ per decade (pvalue < 0.001) and beta pinene by 0.4 ng.m$^{-2}$s$^{-1}$ per decade (pvalue = 0.003).
Similar trends were also observed in the wet season for alpha pinene and beta pinene, and in the dry season for isoprene. The rise in $O_3$ concentrations in Zoetele could therefore be explained by these increasing trends observed in isoprene, alpha pinene and beta pinene from one season to the other and in anthropogenic emissions in African forest regions. These results are corroborated by 18 years of satellite data (1998-2015) by Andela et al. (2017) who noted an increasing trend in burned areas close-canopy forests. In South Africa sites, at Louis Trichardt (pvalue = 0.48) and Amersfoort (pvalue = 0.44), a downward
trend is reported with a very low certainty respectively by -3.4 ppb decade$^{-1}$ and -1.1 ppb decade$^{-1}$ at scale confidence of 95%. The same negative trend is recorded at Cape Point (pvalue = 0.14) with a low certainty (by -2.7 ppb decade$^{-1}$). At Skukuza (pvalue = 0.49), an increase of $O_3$ concentrations by 2.21 ppb decade$^{-1}$ with no evidence of trends at a confidence threshold of 95% is observed. The absence of annual trends at South African sites assessed with certainty confirms the results obtained by Swartz et al. (2020a, 2020b) at the Louis Trichardt, Amersfoort, Skukuza and Cape Point sites using multiple linear regression
model approach. Indeed, the trend lines for the $O_3$ concentrations measured during the entire sampling periods indicate slight negative slopes at Amersfoort and Louis Trichardt and a small positive slope at Skukuza (Swartz et al., 2020a). Gaudel et al. (2020) and Wang et al. (2022) have observed that annual trends of median ozone values have increased in the tropospheric column (950 to 250 hPa) respectively around 2 nmol mol$^{-1}$ decade$^{-1}$ during 22 years (1994-2016) of measurement and of 2.61 $\pm$ 0.34 ppbv per decade (1995-2017) above Gulf of Guinea. These results are contrary to the decrease in ozone observed at the
INDAAF sites in the Gulf of Guinea (Lamto and Djougou), which show no trend. To partly explain the increase of ozone in recent decades, Gaudel et al. (2020) pointed out that although NOx emissions from biomass combustion have decreased in the tropics, this decrease has been overcompensated by the increase in fossil fuel emissions. However, we believe that the discrepancy between these studies could be explained by the proximity of the measurement sites to the sources of precursor emissions. Indeed, the data measured from the commercial aircraft monitoring network used in the work of Gaudel et al. (2020)
are taken at airports closer to cities, whereas INDAAF sites are rural sites, far from fossil fuel combustion sources. In addition, Gaudel et al. (2018) reported that spatially, global surface ozone trends are highly variable depending on time period, region, elevation and proximity to fresh ozone precursor emissions. The distributions of ozone annual trends in the recent two decades (2000-2019) explored by Hou et al. (2023) over six regions on the world including Africa (25°S–25°N, 17°W–51°E) showed a significant increase in these six regions of the Tropospheric Column Ozone (OMI/MLS satellite data), with the smallest
value of ~0.07 DU/yr in the African while with MET+2015EMIS model, the annual trends of ozone over Africa turn to insignificant decreases (0.04 DU/yr). These trends, which contrast with the changes observed at most INDAAF sites, could support that ozone surface data do not necessarily represent the free troposphere where the radiative forcing effects of ozone are concentrated. We also think that the ozone trends could be estimate more accurately in combining of the soundings and long-term surface observations is necessary. Breaks in the annual concentration data were observed at Ba and Ka in 2006 as a
result of the Pettitt test. During the dry and wet seasons, further breaks were recorded in the annual series in 2006 (Ka) and

2007 (Ba) (dry season) and in 2014 at Ka (wet season). However, no trend inversion was induced in these break years. At the other study sites, no breaks were observed.

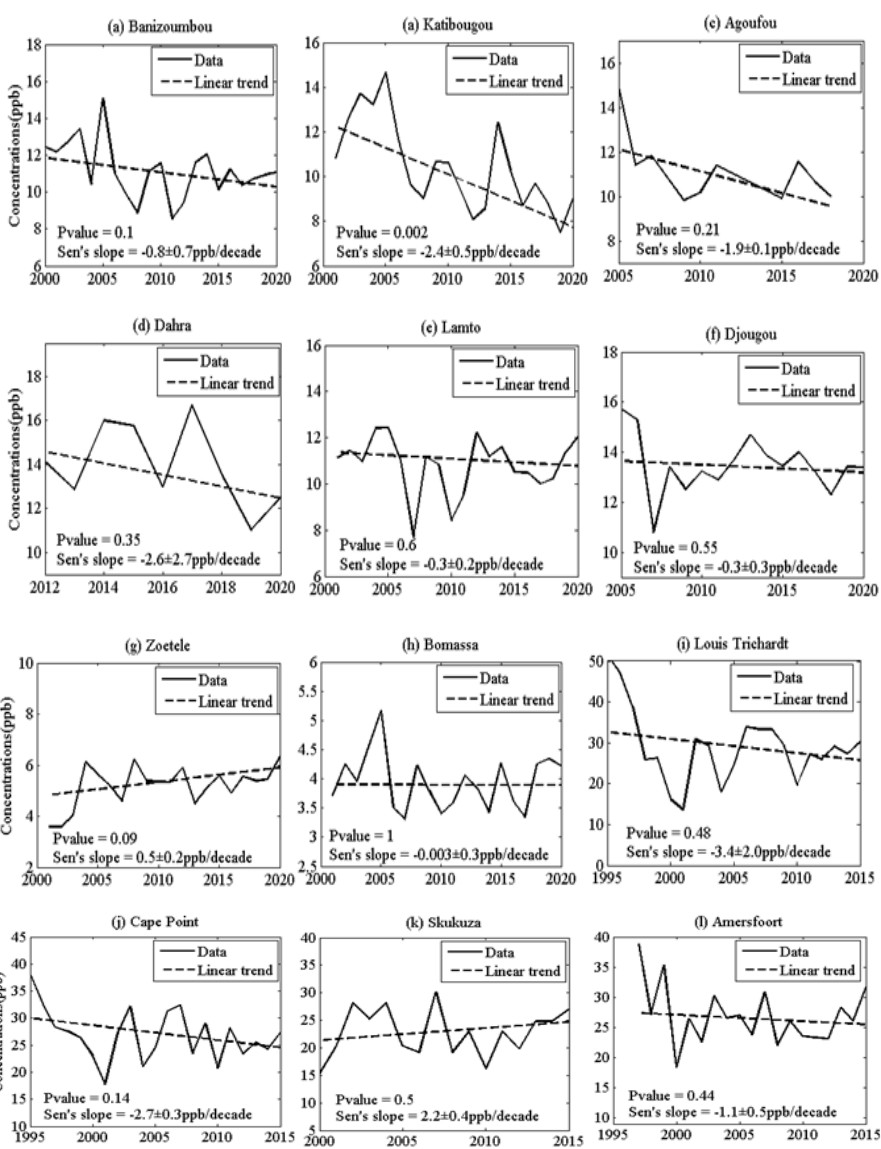

**Figure 14.** Long-term annual linear trend of in situ $O_3$ concentrations over the period 1995-2020 at 95% confidence intervals calculated for 12 measurement sites representative of African dry savanna, wet savanna, forest and agricultural/semi-arid savannas and p-value for each trend.

### 3.3.2 Seasonal trends

Trend tests were performed on monthly mean $O_3$ concentrations using seasonal Kendall test and all trend results are presented in Fig. 15. Test reveals a downward trend of -0.7 ppb decade[-1] at Banizoumbou (pvalue = 0.02), with high certainty at 95% confidence level. At Ka (pvalue <0.001), $O_3$ concentrations decrease by -2.4 ppb decade[-1] with very high certainty. At the other

sites in West Africa, a downward trend of $O_3$ of -0.7 ppb decade$^{-1}$ at Agoufou, -0.6 ppb decade$^{-1}$ at Dahra, -0.23 ppb decade$^{-1}$ at Lamto and -0.11 ppb decade$^{-1}$ at Djougou are calculated. However, the certainty is medium at Agoufou, low at Dahra and Lamto and very low at Djougou at 95% confidence interval. The trends in Bomassa (pvalue = 0.17) and Cape Point (pvalue =0.13), are similar to sites of West Africa with low certainty. At the Louis Trichardt and Amersfoort sites, the results show no ozone trend (pvalue = 0.67 and 0.93) but respectively -2.7 ppb and -0.1 ppb per decade decrease. In contrast, an upward trend

with very high certainty is reported at Zoetele in Cameroon (Sen slope = 0.7 ppb decade$^{-1}$; pvalue = 0.001), and at Skukuza in South Africa (Sen slope = 3.4 ppb decade$^{-1}$; pvalue = 0.001) at 95% confidence interval. All the annual trends observed at INDAAF sites are confirmed by Kendall's seasonal trends.

    The $O_3$ mixing ratio in the lower troposphere is slightly higher in central Africa in July than in northern Africa (in January), likely indicating rapid photochemical $O_3$ production by biomass burning precursors (Singh et al., 1996) during the Southern

Hemisphere fires (Tsivlidou et al., 2023).  Biomass burning air mass transport from the hemisphere where fires occur (where the highest CO is measured) to the opposite hemisphere is allowed by either the north-easterly Harmattan flow (January) or the southeasterly winds and monsoon flow (July) (Sauvage et al., 2005; Tsivlidou et al., 2023) and could be also explain the upward trend observed at Zoetele. At Skukuza, the upward trend is thought to be due to anthropogenic emissions. At the Irene site in the North-eastern interior of South Africa, Bencherif et al. (2020) obtained an upward trend in the tropospheric $O_3$

column (1998–2017) at a rate of around 2.4% per decade. At Irene an increase of tropospheric ozone is reported respectively by Thompson et al. (2014) in spring, and Mulumba et al. (2015) in summer, from ozonesonde records (1990 to 2008) and ozone tropospheric columns (1998 à 2013). An upward trend in $NO_2$ levels was also evident at Skukuza, signifying the influence of growing rural communities on the Kruger National Park border (Swartz et al., 2020a). At the South African sites, the high values of NOx and COV fluxes are observed at Skukuza (Fig. 11d). The population growth and the associated increase

in anthropogenic activities (like domestic biomass burning and solid fuel combustion) result in high levels of pollutants in South Africa's Highveld region (Kai et al., 2022; Keita et al., 2021) and could therefore justify the upward trend obtained. The increase in $O_3$ in South Africa could be also explained by biomass burning (agriculture reason) and greenhouse (urban-industry reason) activities implemented in Africa during summer and spring seasons (Bencherif et al., 2020; Diab et al., 2004; Sivakumar et al., 2017). Furthermore, pollutants emissions from domestic biofuel, brought by long-range transport of pollution

in the Southern Hemisphere have an impact at the continental scale (Thompson et al., 2014). Moreover, Thompson et al. (2021) have observed an increase mean trend of + 1.2% per decade (pvalue = 0.119) from 22-year SHADOZ record (1998–2019) of ozone profiles in the free tropospheric (5–15 km) at Nairobi in East Africa. This trend result is similar with surface ozone change observed in Skukuza. The increase in tropospheric ozone in the tropics, is partly of dynamic origin with the movement of air masses and not solely due to growing anthropogenic emissions (Thompson et al., 2021), could also explain this similarity.

Balashov et al. (2014) reported from 1990 to 2007 over the South African Highveld that the sites of Palmer and Makalu exhibit statistically significant negative trends in surface ozone over the spring season and the month of September respectively, whereas Verkykkop and Elandsfontein show no statistically significant change in surface ozone. At most INDAAF sites (remote and rural) we found, as in the work of Balashov et al. (2014), that the surface ozone data still showed no trend. These results confirm however that surface ozone trends in Africa are not uniform regionally or seasonally as mentioned by

Thompson et al. (2021).

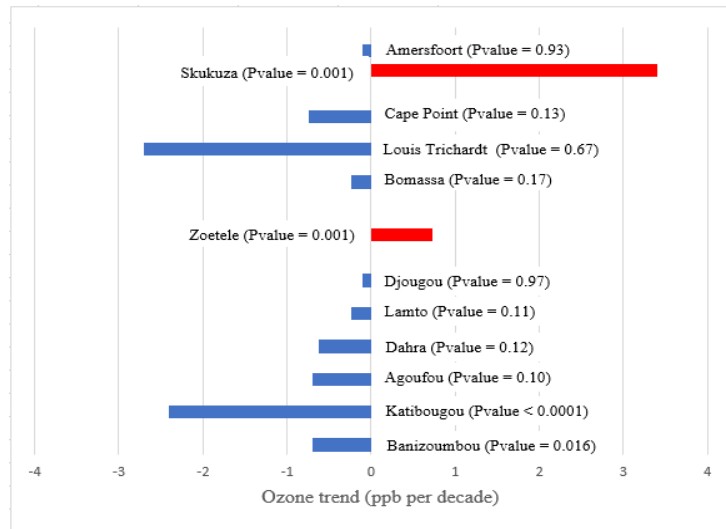

**Figure 15** Kendall's seasonal trend of in situ O₃ concentrations over the period 1995-2020 at 95% confidence intervals calculated for 12 measurement sites representative of African dry savanna, wet savanna, forest and agricultural/semi-arid savannas and p-value for each trend.


## 4. Conclusion

This work presents an original database of long-term $O_3$ concentrations at fourteen African sites belonging to the INDAAF program and companion projects. This database gives a better understanding of $O_3$ concentration levels at remote tropical sites representative of the major African biomes. In this study, we establish a mean annual cycle by site and ecosystem type, and
investigate the seasonal variability of $O_3$ concentrations over the period 1995-2020. Our analysis of the seasonality of anthropogenic and biogenic NOx and VOC emissions then highlights the significant factors contributing to $O_3$ formation. Finally, we calculate the $O_3$ long-term trends, which provide an insight into the long-term evolution of $O_3$ levels and the local and regional dynamics of the emission sources of its precursors.

The results indicated that $O_3$ levels are the highest during the rainy season in dry savannas and during the dry season in wet
savannas and forests. In agricultural fields, no seasonal variations of $O_3$ concentrations are observed. In semi-arid savanna (South Africa), dry season $O_3$ levels are the highest at Louis Trichardt and Skukuza. At Cape Point and Amersfoort sites, maxima occur during the rainy season. Mean annual $O_3$ concentrations range from 10.5±5.4 to 14.8±4.3 ppb in dry savannas, from 10.8±3.3 to 13.5±4.8 ppb in wet savannas, from 3.9±1.1 to 5.2±2.1 ppb in forest ecosystems and from 19.9±4.7 to 29.1±8.4 ppb in semi-arid/agricultural savannas. BVOC (under the influence of air temperature), NO emissions (in the presence
of humidity) and precipitation, are the main contributors to $O_3$ formation in dry savannas. The seasonality of $O_3$ measurements and dominant precursors confirm the important role of microbial processes leading to high NO emissions at the beginning of the wet season for $O_3$ production. Furthermore, the influence of air temperature and solar radiation on woody emissions from shrubs in the Sahel, and the presence of sparse vegetation (short grasses, forbs and dicotyledonous shrubs with perennial ground cover) in this region could be at the origin of BVOC emissions. The photochemical $O_3$ regime in savannas and
rainforests (heavily vegetated areas) is strongly linked to BVOC emissions from vegetation, and to temperature, radiation and humidity. At Lamto and Zoetele, anthropogenic NOx and VOC also contribute to $O_3$ formation. The most dominant precursor species in southern Africa are mainly NOx emissions (anthropogenic and biogenic), humidity and temperature, as well as anthropogenic VOC at a few sites. They are due of Biomass and fuel combustion, large-scale transport of pollutants, domestic combustion in winter and biogenic emissions from vegetation. At INDAAF sites, which are rural sites far from many
anthropogenic sources, $O_3$ concentrations are below most of the values reported in the literature. At 95% confidence intervals,

annual and seasonal Mann-Kendall trends at all sites indicate that the Katibougou site in Mali and the Banizoumbou site in Niger experience a decrease in $O_3$ concentrations (around -2.4 ppb decade$^{-1}$ and -0.8 ppb decade$^{-1}$) with a high certainty over the period 2000 to 2020 justified by downward trends of $NO_2$ trends observed at Katibougou and the BVOC emissions at Banizoumbou. In contrast, a significant upward trend is reported at Zoetele (0.7 ppb decade$^{-1}$) in Cameroon and Skukuza (3.4

ppb decade$^{-1}$) in South Africa.  These trends could be attributed to the increase in BVOCs in Zoetele and anthropogenic and biogenic emissions in Skukuza.

This study described in details the $O_3$ levels in representative African biomes, as well as the photochemical regimes and conditions leading to the observed concentrations. The results presented in this article constitute a robust database showing the importance of developing or maintaining long-term observation projects and observatories. This database could be used to

assess the impact of $O_3$ dry deposition fluxes on African crops and the potential yield losses because of $O_3$ absorption by crops. An assessment of the various agricultural losses during the growing season will help to better orient actions to improve crop yield and achieve food security. In addition, this documentation is invaluable for modeling chemical processes in the atmosphere and for projecting future changes in tropospheric $O_3$. It could limit the uncertainties of these models and facilitate their validation, which is mainly based on data measured in situ.


**Data availability.** Dataset DOIs of $O_3$ observations for INDAAF sites (see complete citation in the reference list), available in the INDAAF database at https://indaaf.obs-mip.fr :

| Banizoumbou (Niger) | Katibougou (Mali) | Agoufou (Mali) | Bambey (Senegal) |
|---|---|---|---|
| https://doi.org/10.25326/608 Laouali et al. (2023) | https://doi.org/10.25326/604 Galy-Lacaux et al. (2023a) | https://doi.org/10.25326/610 Galy-Lacaux et al. (2023b) | https://doi.org/10.25326/609 Galy-Lacaux et al. (2023c) |
| Dahra (Senegal) | Lamto (Cote d'Ivoire) | Djougou (Benin) | Zoetele (Cameroon) |
| https://doi.org/10.25326/606 Galy-Lacaux et al. (2023d) | https://doi.org/10.25326/275 Galy-Lacaux et al. (2023e) | https://doi.org/10.25326/605 Akpo et al. (2023) | https://doi.org/10.25326/603 Ouafo-Leumbe et al. (2023) |
| Bomassa (Congo) | Mbita (Kenya) | Louis Trichardt (South Africa) | Skukuza (South Africa) |
| https://doi.org/10.25326/607 Galy-Lacaux et al. (2023f) | https://doi.org/10.25326/642 Galy-Lacaux et al. (2023g) | https://doi.org/10.25326/646 van Zyl et al. (2023a) | https://doi.org/10.25326/645 van Zyl et al. (2023b) |
| Cape Point (South Africa) | Amersfoort (South Africa) | | |
| https://doi.org/10.25326/644 van Zyl et al. (2023c) | https://doi.org/10.25326/647 van Zyl et al. (2023d) | | |

GFED4 (NOx, COV) and MEGAN-MACC (Isoprene, pinene-a, pinene-b) data are available from https://eccad.sedoo.fr/#/data.       ERA5       reanalysis       data       are       available       from https://cds.climate.copernicus.eu/cdsapp#!/dataset/reanalysis-era5-single-levels?tab=overview.

**Author contributions.** CGL designed the study, wrote the protocol and edited the paper. HEVD conducted data processing,

the statistical analysis and wrote the paper. CD made conceptual contributions and edited the paper. ABA and MO contributed at the statistical analysis, assisted in sample collection and edited the paper. VY, DL, MO-L, ON, PGVZ assisted in sample collection and edited the paper. EG and MDA analysed the samples.

**Competing interests.** The contact author has declared that none of the authors has any competing interests.


**Disclaimer.**

**Acknowledgements.** The authors would like to acknowledge the INDAAF project (International Network to study Deposition and Atmospheric chemistry in Africa), and especially all its local technicians for their maintenance and sampling work. We would also like to acknowledge the "Cycle de l'Azote entre la Surface et l'Atmosphère en afrique" (CASAQUE) project and the Integrated Nitrogen Management system (INMS) for providing us with data on $O_3$ concentrations at Dahra and Mbita. This study was supported by the INSA (Integrated Nitrogen Studies in Africa) project, which funded the research carried out for this paper at the "Laboratoire d'Aérologie de Toulouse". We are indebted to the AMMA-CATCH project, the University of Copenhagen and the Observatoire de recherche en environnement "Bassins versants tropicaux expérimentaux" (SO BVET) for providing us with meteorological data from Niger, Benin, Senegal and Cameroon. We are also grateful to the ECCAD platform and the European Centre for Medium-Range Weather Forecasts (ECMWF) for biogenic and anthropogenic emissions and ERA5 reanalysis data.

**Financial support.** This study has received funding from the European Union's Horizon 2020 research and innovation programme under the Marie Skłodowska-Curie grant agreement no. 871944.

**Review statement**.

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
