# Peer review of "Measurement report: Long-term measurements of ozone concentrations in semi-natural African ecosystems"

_EGUsphere, 2024_

## Referee Comment (RC1)

This paper "Long-term measurement of ozone concentrations in semi-natural African ecosystems", presents long-term (1995-2020) data on ozone concentrations across diverse African ecosystems. The analysis, based on passive samplers deployed at 14 sites, explores seasonal variations, long-term trends, and the influence of local factors like temperature, precipitation, and precursor emissions on ozone formation. This research fills a critical data gap for understanding ozone behavior in Africa, a region with limited such measurements. Therefore, I recommend publication after the points are considered.

**Abstract**
Line 25, page 1:
*Atmospheric levels*: could you be more precise? How do you define the atmosphere in terms of height?

Line 30, page 1:
*Over the period 1995-2020, monthly ozone concentrations were measured at these sites using passive samplers*
How do you get ozone levels in the atmosphere using passive samplers?

Line 32, Page 1:
*Ozone levels in the wet season (in dry savanna) are higher and comparable to concentrations in the dry season (in wet savanna and forest).*

I find this sentence a bit confusing. Does it compare ozone levels in dry savanna with wet savanna and forest? Does ozone levels refer to the whole troposphere?

Lines 33-34, page 1:
You mention East Africa, Southern Africa and Sahel, while before you group the sites depending on the ecosystem they belong to. How are these geographical regions defined? Does each region correspond to one ecosystem? With other words, it's not clear to me if the characteristics are due to the geographical location of the sites or because they belong to the same ecosystem.

Lines 34-39, page 1:
How do you support the dependence of ozone levels to different factors such as temperature, NOx emissions etc? How do you distinguish the effect of anthropogenic NOx emissions? Do you use any chemical transport model and sensitivity tests or these results are based on other studies?

Line 44, page 2:
*The increasing trends are consistent with the increase in biogenic emissions at Zoétélé and NO2 levels at Skukuza.*

The increase in biogenic emissions and NO2 levels results from this study?

**Introduction**
Lines 94-105, page 3:

This part could be more informative by discussing the main finding for ozone in Africa, rather than listing the previous studies.

Line 112-114, page 3:
*In the first objective, we first document the long-term (1995-2020 depending on the site) monthly, seasonal and interannual variability of O3 concentrations on a regional scale at fourteen sites grouped by ecosystem (dry savannas, humid savannas, forests and agricultural/semi-arid savannas),*

Grouped by ecosystem: does it mean that you merge measurements from different sites if they belong to the same ecosystem? Or do you analyse the measurements of each site separately?

**Material and methods**
2.1 Sampling sizes
Lines 141-143, page 4:
The abbreviations for the sites are confusing and hard to remember. It makes it hard for the reader to understand the site you are referring to.

Line 151, page 5:
Figure 1: how are the sites ordered on the map? The map would be easier to read if you order the sites by increasing longitude.

2.2 Passive sampling and chemical analysis
Table 2, page 5:
Why is the measurement altitude different for the forest sites?

In Lamto, the sampling period is 2001-2020 and the data collection efficiency is 226/240. What do the numbers mean exactly? Please clarify. The number of months of the sampling period is 228. Is it 226 monthly measurements taken into account for the period 2001-2020?

The sampling period is quite different depending on the site (e.g 2017-2020 and 1995-2020). How do you assess the statistical significance of the measurements or the trends? How do you compare trends or ozone levels while the sampling period is different depending on the site?

2.3 Meteorological parameters and leaf area index
Lines 173-184, page 6:
What difference does it make that you use different datasets for the meteorological parameters depending on the site?
Have you tried to compare data from AMMA-CATCH database/INDAAF database with the ERA5 data for the common location? What is the consistency between the different datasets?

2.4 NOX and VOC emissions
Lines 197-199, page 7:
*The biogenic NO fluxes used are model outputs in reference to the work of Delon et al. (2010, 2012). They were filtered in the eastern grid from 5° S to 20° N in latitude, and 20° W*

*to 30° E in longitude over the period from 2002 to 2007 and cover only the Ba, Ka, Ag, La, Dj, Zo and Bo sites.*

What does the phrase "*model outputs in reference to the work of Delon et al. (2010, 2012)."* mean? Please clarify.
I interpreted this phrase as you used the model output produced by Delon et al. 2010 and 2012? If yes, have you considered using more up-to-date model output and emission inventories?
Why do you use data for the period from 2002 to 2007? Why does this period not match the sampling period of the sites (Table 2)?

2.5 Statistical analysis
General comment:
The application of PCA to identify factors controlling ozone concentration raises significant concerns. PCA assumes linearity in variable relationships, which may not hold true for ozone formation processes involving precursor emissions and meteorological conditions. This raises concerns about the interpretability and validity of the results. Considering alternative methods like machine learning techniques designed to handle non-linear relationships would be more appropriate for this type of analysis. At least, it's crucial to acknowledge the limitations of PCA in establishing causal relationships between ozone and its controlling factors.

Line 221, page 7:
The use of some references (e.g. Tsuyuzaki et al., 2020) to justify PCA analysis seems outside the usual scope of atmospheric research. Considering established applications of PCA in atmospheric science for analyzing ozone surface data would be more relevant.

**Results and discussion**
3.1 Meteorological and biophysical parameters variation
Lines 231-245, pages 7-8:
This paragraph focuses on describing the content of Figure 2. It would be more informative to analyze and explain the underlying mechanisms driving the patterns observed in Fig. 2. Several relevant studies by LAERO (e.g. Adon et al. 2010; Lannuque et al., 2021; Sauvage et al., 2005; 2007; Tsivlidou et al. 2023) explore the African meteorology (e.g. ITCZ, Harmattan, monsoon). Using these explanations, the authors could enrich the discussion and explain the seasonal variations evident in Figure 2.

Adon, M., Galy-Lacaux, C., Yoboué, V., Delon, C., Lacaux, J. P., Castera, P., Gardrat, E., Pienaar, J., Al Ourabi, H., Laouali, D., Diop, B., Sigha-Nkamdjou, L., Akpo, A., Tathy, J. P., Lavenu, F., and Mougin, E.: Long term measurements of sulfur dioxide, nitrogen dioxide, ammonia, nitric acid and ozone in Africa using passive samplers, Atmos. Chem. Phys., 10, 7467–7487, https://doi.org/10.5194/acp-10-7467-2010, 2010.

Lannuque, V., Sauvage, B., Barret, B., Clark, H., Athier, G., Boulanger, D., Cammas, J.-P., Cousin, J.-M., Fontaine, A., Le Flochmoën, E., Nédélec, P., Petetin, H., Pfaffenzeller, I., Rohs, S., Smit, H. G. J., Wolff, P., and Thouret, V.: Origins and characterization of CO and O3 in the African upper troposphere, Atmos. Chem. Phys., 21, 14535–14555, https://doi.org/10.5194/acp-21-14535-2021, 2021.

Sauvage, B., Thouret, V., Cammas, J.-P., Gheusi, F., Athier, G., and Nédélec, P.: Tropospheric ozone over Equatorial Africa: regional aspects from the MOZAIC data, Atmos. Chem. Phys., 5, 311–335, https://doi.org/10.5194/acp-5-311-2005, 2005

Sauvage, B., Gheusi, F., Thouret, V., Cammas, J.-P., Duron, J., Escobar, J., Mari, C., Mascart, P., and Pont, V.: Medium-range mid-tropospheric transport of ozone and precursors over Africa: two numerical case studies in dry and wet seasons, Atmos. Chem. Phys., 7, 5357–5370, https://doi.org/10.5194/acp-7-5357-2007, 2007.

Tsivlidou, M., Sauvage, B., Bennouna, Y., Blot, R., Boulanger, D., Clark, H., Le Flochmoën, E., Nédélec, P., Thouret, V., Wolff, P., and Barret, B.: Tropical tropospheric ozone and carbon monoxide distributions: characteristics, origins, and control factors, as seen by IAGOS and IASI, Atmos. Chem. Phys., 23, 14039–14063, https://doi.org/10.5194/acp-23-14039-2023, 2023.

Figure 2, page 9: the range of the left and right axes differ depending on the site. This makes it harder for the reader to make comparisons between the different locations.

Figure 3, page 10:
Too low O3 surface concentrations compared to Tsivlidou et al 2023- Sahel (their fig. S3 panel 1a). Do you have any idea why?

Table 3, page 11: What does 'moy' stand for?

Figure 6, page 13:
Too low concentrations compared to Tsivlidou et al., 2023- Central Africa (their fig. 3 panel 1b )? Any idea why?

Figure 9, page 15:
The location markers in Fig. 9 (especially locations 1–7) do not match the ones from fig. 1 (page 5).

3.3.1 NOx and VOC anthropogenic emissions & 3.3.2 NOx and VOC natural emissions:

To streamline the flow of the text, it might be beneficial to consider consolidating sections 3.3.1 and 3.3.2. Some of the key reasons for the ozone peaks and their seasonal patterns have already been established earlier. Merging these sections would reduce redundancy and maintain a focused narrative.

3.4.2 Characterisation of O3 precursor emission sources and studies of correlations

Similar to the previous comment above, there appears to be some repetition of explanations and citations within 3.4.2. To enhance readability, particularly given the paper's length, combining this section with other paragraphs discussing ozone features (e.g., 3.2.1, 3.3, 3.4.2) could be explored. This would allow for a more concise and focused presentation of the findings.

For improved readability and reduced redundancy, I recommend exploring the possibility of consolidating sections discussing ozone features. This could involve merging 3.2.1, 3.3, and 3.4.2 into a single, well-organized section that succinctly explains the observed phenomena.

3.5.1 Annual trends
Figure 15, page 26:
What could cause the maxima of $O_3$ over Katibougou (a and b) and Banizoumbou (d) in 2014?

**Conclusions:**
Line 628, page 28:
What do you mean that the different *photochemical regimes* are discussed in the study?

**Technical suggestions:**
Line 76, page 2:
(Gaudel et al., 2018; Fleming et al., 2018; Mills et al., 2018) should be (Fleming et al., 2018; Gaudel et al., 2018; Mills et al., 2018)

---

## Referee Comment (RC2)

**REVIEW OF Hagninou Elagnon Venance Donnou et al.: "Measurement report: Long-term measurements of ozone concentrations in semi-natural African ecosystems"**

**SUMMARY –** This paper analyzes 26 years of trace gas measurements (1995-2020) collected by passive filters from 14 nonurban stations across Africa, mostly in west Africa south to South Africa in the International Network to study Deposition and Atmospheric chemistry in Africa (INDAAF). There are three sections to the paper. (1) climatology of monthly ozone data from the 14 stations, organized by 4 general ecosystem types. (2) Seasonality is linked to chemical and meteorological parameters- BVOC, combustion VOC (industrial, biomass fires) and NO emissions, humidity, precipitation. (3) Trends in ozone over the period 1995-2020 are computed by typical statistical methods.

**OVERALL RECOMMENDATION** – The goals of the paper encompass a large range of topics which are too much for a single article because several aspects of the study show incomplete analyses. Of the three topics listed above, the correlations with NO and VOC (Topic 2) may be the most novel. However, the origins of those "values" and "trends" raise more questions than can be answered in one article. This Reviewer recommends dropping that material (Sections specified below) from a revised submittal and focusing on two topics: (1) ozone climatology and the relationship of seasonality and variability with the wet/dry meteorological variables and the LAI; (2) the trends and comparison with several new trend studies that include African data.

The Introduction needs to be revised to briefly review prior work that is most relevant to the INDAAF project. Here is the best place to include reference to the many campaigns held in west and southern Africa (Listed in 3c below) that examine processes, particularly related to fire impacts on ozone, but that are only "snapshots in time" compared to the long-term INDAAF measurements. The authors should put some of their existing references to meteorological and seasonal influences in the Introduction to give context and to motivate the reader to understand how their seasonal analyses (Section 3.1) extend and give new insights into the earlier work. (The goal here is to highlight WHY your study is original and important!)

In revising subsequent Sections: (1) For the ozone climatology, a priority request is more detail about quality control in passive sampler datasets. (2) For ozone trends, the results shown (Figure 15) should be augmented with trends to display all 14 sites (Section 3.5). Additional context for interpretating these results is needed, including comparisons with other publications on African ozone trends. How do the trends over South Africa compare, for example, with other studies? Similar, different, why? In summary, the Reviewer recommends the paper for ACP publication \*after\* major revision. The latter must include better evidence of INDAAF ozone data quality and more thorough analyses of the 1995-2020 trends at the 14 stations.

"Measurement Report" does not sound like an appropriate classification for a paper that presents more original scientific results than simply documenting the existence and archiving of a dataset. The reviewer suggests dropping the "Measurement Report" heading. A more suitable title might be: "Surface ozone seasonality and trends over Africa (1995-2020) from the INDAAF project".

**MAIN COMMENTS**

1.  Abstract. A major concern about the ozone measurements is quality control of the passive sampler data (more below). A sentence or two that documents the reliability of the INDAAF station ozone observations is needed in the Abstract. Lines 33-34 – Sentence is ambiguous as written. Do you mean southern Africa has higher mean ozone concentrations or greater seasonal variations in ozone in east Africa? The Sahel region sites are not defined although it is recommended that the two sentences in Lines 34 to 37 be omitted. Line 40 –

specify the dates of the trend calculations. The lines 42 to 44 refer to changes in VOC and NO2 that appear to be assumptions. That is, the paper gives no measurements to support this statement for these 2 INDAAF sites. These lines should be removed.

2.  Sections on NO, VOC, BVOC.  Three reasons for *recommending nearly all the material in Sections 2.4, 3.3, and 3.4 be removed from the current paper.* (a) The paper is too long and yet, important information is still lacking. Example: explanation of the method in 3.4.1 needs to be expanded. (b) It is not clear to what extent values for the species NOx, VOC, BVOC, their seasonality of emissions, changes over time, are from model(s) and/or experiments and if so, which ones?  Explain. It is important when referring to published relationships between ozone and the precursors to distinguish between links based on measurements (clarify with references to the data sources and publications) and "trends" based on models that are of unknown accuracy. Related to this point, the relationships for species and ozone in Figure 12 are not convincing, in contrast to those in Figure 2 where there are definite links of ozone with climatic/meteorological parameters. (c) The correlations in Figure 14 and Table 5 are intriguing but not very clear. For example, previous studies demonstrate biomass fire impacts on sites in northeast South Africa but how is the attribution to VOC and NO combustion made here?  *In summary, the Reviewer finds good potential in linking ozone with the precursors and processes (biogenic vs combustion NOx and VOC, for example) but the analyses in this paper are confusing in some places and are not convincing in other sections.* Recommend that the authors move these analyses to a second paper that presents ozone and precursor relationships with more rigor and *observational* evidence.

3.  There are a number of places where statements should be made more accurately. Many suitable references are given in the paper but sometimes in a misleading way.  Examples from **Section 1 follow**:

a.  Line 61.  Since Zhang et al. (2016) both modeling and observational studies have shown that ozone trends are not uniform regionally or seasonally, i.e. even in the tropics a number of sites with ozonesonde profiles exhibit no trend (Thompson et al., 2021). A study with sondes over equatorial southeast Asia, published in the TOAR II collection by Stauffer et al. (2024), shows no definite ozone trend annually but a 6-8%/decade increase limited to 3 months/year.  *Insert words to this effect on Line 62 after Adon et al., 2010.*  The references are here:

Thompson, A. M., R. M. Stauffer, K. Wargan, J. C. Witte, D. E. Kollonige, J. R. Ziemke, Regional and seasonal trends in tropical ozone from SHADOZ profiles: Reference for models and satellite products, J. Geophys. Res., https://agupubs.onlinelibrary.wiley.com/doi/10.1029/2021JD034691, 2021.

Stauffer, R. M., A. M. Thompson, D. E. Kollonige, N. Komala, H. Khirzin Al-Ghazali, D. Y. Risdianto, A. Dindang, A. F. bin Jamaluddin, M. Kumar Sammathuria, N. Binti Zakaria, B. J. Johnson, P. D. Cullis, Dynamical drivers of free-tropospheric ozone increases over equatorial Southeast Asia, Atmos. Chem. Phys., https://doi.org/10.5194/acp-24-5221-2024

b.  Line 63-65. Laban et al. (2018), which describes patterns of South African ozone at a number of sites that are both rural and anthropogenically influenced, emphasizes that the highest ozone is in the dry season in the semi-arid areas in the Highveld, largely due to a maximum in domestic fuel usage for winter heating, not biosphere interactions, as the paper states.

c.  Lines 70-75.  The Reviewer is not sure if the continents of South America and Africa were compared that Africa (at least sub-Saharan Africa) is "the least studied." On line 73 it is more accurate to say "*may be* the least studied continent." Over the past ~30 years there have been many large African field campaigns conducted (AMMA, EXPRESSO, SAFARI/TRACE-A, ORACLES, SAFARI-2000) such that hundreds of articles have been published on African air quality and environment.  Links to dynamical factors affecting ozone seasonality (Diab et al.,

1996; 2003; 2004), interannual variability in ozone related to ENSO (Balashov et al., 2014) and widespread impact of biomass and domestic fires in southern Africa are well-established.  On the latter, refer to detailed analyses of ozone and related measurements in Special Issues of *J. Geophysical Res.* on SAFARI/TRACE-A, and SAFARI-2000. Southern Africa has been the major arena of these ozone studies and South Africa has had several high-quality monitoring programs. In Line 65 it is more accurate to say "With the exception of South Africa, ozone variability is poorly documented on the African continent." For example, in North American and European journals, North Africa probably has the smallest number of articles. However, having said that there are many African studies about ozone and air quality, the number of measurements *publicly available* may be very small. The authors can point out that the INDAAF is among the few datasets that are available to the scientific community.  Diab et al: Diab, R. D., A. M. Thompson, M. Zunckel, G. J. R. Coetzee, J. B. Combrink, G. E. Bodeker, J. Fishman, F. Sokolic, D. P. McNamara, C. B. Archer, and D. Nganga, Vertical ozone distribution over southern Africa and adjacent oceans during SAFARI-92, *J. Geophys. Res.*, 101, 23,809-23,821, 1996;  Diab, R. D., A. Raghunandran, A. M. Thompson, V. Thouret, Classification of tropospheric ozone profiles over Johannesburg based on MOZAIC aircraft data, Atmos. Chem. Phys., 3, 713-723, 2003;  Diab, R. D., A. M. Thompson, K. Mari, L. Ramsay, G. J. R. Coetzee, Tropospheric Ozone Climatology over Irene, South Africa from 1990-1994 and 1998-2002, J, Geophys. Res., 109, D20, D20301, doi: 10.1029/2004JD004293, 2004.

    d.    Lines 100-105.  A number of references are given about studies and campaigns but little information about the findings of each that are relevant to the authors' study. What the reader wants to see is "what do we know from the prior campaigns?" "Do they agree with one another?"  "What are new INDAAF results that confirm, contradict or complement the earlier findings?" The list in these lines is not useful without connecting the background to the current paper.

    e.    In like manner to comment (b) above about Line 65, LINE 106 is not correct. There have been a number of studies with South African ozone and related data.  Line 106 should say "With the exception of South Africa very little information is available on the long-term evolution of O3 chemistry over Africa. The impact of meteorological parameters and atmospheric chemistry… and the analysis of long-term trends *is only partially explained.*" *In other words it is not correct to say the trends are "unexplored.*" The authors cite Balashov et al. (2014) (line 537) which determines trends for 5 South African sites with high quality surface O3 data, over ~15 years; see also Martins et al, 2007. Trends over a longer period, ~1990 to 2011 in ACP, described the seasonality of free tropospheric O3 trends over Irene [Pretoria], South Africa with both IAGOS and ozonesonde data (Thompson et al., Atmos. Chem. Phys., **14,** 9855-9869, 2014; the latter paper corrected a sampling error in Clain et al., 2009). In ACP, Gaudel et al. (egusphere-2023-3095) use IAGOS African aircraft data to estimate trends over a number of sub-Saharan African cities for the period ~1995-2019.

4.  Section 2.2.  This section, fundamental to the quality of the paper, is inadequate. It is not enough to cite previous papers to establish the accuracy of the INDAAF ozone record. Uncertainties in ozone mixing ratios for typical samples need to be provided. Uncertainties are also needed in the graph of trends in Figure 15. To give further confidence in the ozone time series for the INDAAF sites, comparisons of sampler ozone with independent ozone measurements should be made for sites where the latter data are available within the 1995-2020 period. Examples: ozone from Irene, South Africa, sondes at the surface and Welgegund (continuous ozone monitor) can be compared to LT; if ozone from the WMO/GAW station at Cape Point is available from an independent analyzer, a comparison of trends from such an instrument during the INDAAF period should be made.  Although Nairobi observations are not co-located with the Mbita INDAAF data, their ozone should display similar seasonal patterns. Nairobi ozonesonde data are available from SHADOZ (1998-2023; https://tropo.gsfc.nasa.gov/ shadoz); surface monitor ozone data may also be available.

5. Sections 3.1 and 3.2.

The analyses corresponding to Figures 3-9 (with Table 3) are very good. In later Sections, e.g., 3.4.2.3, there are references to meteorological influences on South African ozone seasonality (humidity, temperature; Balashov et al., 2014; Laban et al., 2018; 2020). The Introduction and/or earlier sections would be strengthened by moving some of these references forward.

Figures 4, 6, 8. These Figures originate from averaging the values in Figures 3, 5, 7, respectively, no? Please explain in the captions.

Figure 9. Likewise, although parts of the text define the periods of "wet", "dry" season, it would be useful to repeat or summarize those definitions in the Figure 9 caption.

Page 12. Discussion. References to Nepal climatology are not relevant here. Remove.

Pages 16 and 17. These comparisons are not relevant to the paper. There is no value to Table 4 and it should be deleted along with Lines 380-391. We don't know that the same analytical methods are used as at INDAAF sites. The dates are not a match for the INDAAF period in many cases. What is the point of making comparisons with Arctic, Antarctic, midlatitude sites?

Figure 10. This Figure *is* relevant to the discussion of INDAAF ozone climatology. In particular, the INDAAF data for South Africa should be added to the Figure and those values should be compared to earlier publications with South African data. Reasons for similarities and differences should be discussed, including how sampling systems for the non-INDAAF data compare to INDAAF measurements in similar parts of South Africa.

6. Sections 3.3, 3.4

As mentioned above, most of this material should be removed or saved for a separate manuscript because it is about precursor variability and trends for which necessary explanations, references and background are beyond the scope of this paper.

Section 3.5

The focus on O3 trends is one of two major results of this paper but the discussion is incomplete (no South African data displayed, for example). Some points raised in Section 3.5.1 refer to BVOC trends (? From models; if so give short explanation). There have been trends papers with African ozone data that cover most of the INDAAF period analyzed here, ~mid-late 1990s through 2020: Gaudel et al., 2018; 2020; in review- egusphere-2023-3095; Thompson et al., 2021. How do the authors' INDAAF trends agree? Discuss in more depth. There should be references on seasonality of fire impacts from the Piketh and van Zyl-Beukes groups. Expand the literature search. It may be possible to add truly relevant articles to the bibliography and remove references on biogenic emissions or model "trends" that are not data.

**Miscellaneous Comments and Questions:**

**Abstract.** Place names. Location, Country; - separate with semi-colon. Banizoumbou, Niger; Katibougou and Agoufou, Mali; etc

**Line 53** Don't begin sentence with O3, use "Ozone"

**Line 94.** "Previous studies" – delete "existing"

**Line 112.** Delete the 2nd occurrence of "first"

**Lines 116-120**. Recommend omitting 3 sentences. *A second objective is then: "In the 2nd objective, we use non-parametric statistical tests to assess seasonal and annual trends in O3 in the context of other trend analyses of Africa ozone (Thompson et al., 2014; Gaudel et al., 2018; Gaudel et al., 2024)."* Option to add: "In a companion/separate article INDAAF ozone trends are linked to potential changes in NO2, VOC, BVOC."

**Table 2**. For South Africa LT, Sk, Af – End point is 2015. Did the program end there or are there data taken since 2015 that are not publicly available?

---

## Community Comment (CC2)

Comments by Owen R. Cooper (TOAR Scientific Coordinator of the Community Special Issue) on:

**Measurement report: Long-term measurements of ozone concentrations in semi-natural African ecosystems**

Hagninou Elagnon Venance Donnou, Aristide Barthélémy Akpo, Money Ossohou, Claire Delon, Véronique Yoboué, Dungall Laouali, Marie Ouafo-Leumbe, Pieter Gideon Van Zyl, Ousmane Ndiaye, Eric Gardrat, Maria Dias-Alves, and Corinne Galy-Lacaux

EGUsphere [preprint], https://doi.org/10.5194/egusphere-2024-284, 2024
Discussion started: 16 April 2024;  Discussion closes 8 June, 2024

This review is by Owen Cooper, TOAR Scientific Coordinator of the TOAR-II Community Special Issue. I, or a member of the TOAR-II Steering Committee, will post comments on all papers submitted to the TOAR-II Community Special Issue, which is an inter-journal special issue accommodating submissions to six Copernicus journals:  ACP (lead journal), AMT, GMD, ESSD, ASCMO and BG. The primary purpose of these reviews is to identify any discrepancies across the TOAR-II submissions, and to allow the author teams time to address the discrepancies.  Additional comments may be included with the reviews. While O. Cooper and members of the TOAR-II Steering Committee may post open comments on papers submitted to the TOAR-II Community Special Issue, they are not involved with the decision to accept or reject a paper for publication, which is entirely handled by the journal's editorial team.

**General Comments:**

TOAR-II has produced two guidance documents to help authors develop their manuscripts so that results can be consistently compared across the wide range of studies that will be written for the TOAR-II Community Special Issue.  Both guidance documents can be found on the TOAR-II webpage: https://igacproject.org/activities/TOAR/TOAR-II

*The TOAR-II Community Special Issue Guidelines*:   In the spirit of collaboration and to allow TOAR-II findings to be directly comparable across publications, the TOAR-II Steering Committee has issued this set of guidelines regarding style, units, plotting scales, regional and tropospheric column comparisons, tropopause definitions and best statistical practices.

*Guidance note on best statistical practices for TOAR analyses*:  The aim of this guidance note is to provide recommendations on best statistical practices and to ensure consistent communication of statistical analysis and associated uncertainty across TOAR publications. The scope includes approaches for reporting trends, a discussion of strengths and weaknesses of commonly used techniques, and calibrated language for the communication of uncertainty. Table 3 of the TOAR-II statistical guidelines provides calibrated language for describing trends and uncertainty, similar to the approach of IPCC, which allows trends to be discussed without having to use the problematic expression, "statistically significant".

**Specific Comments:**

TOAR-II Steering Committee member, Dr. Erika von Schneidemesser is an expert on small sensors, and she has independently posted a few comments on this paper to the EGUsphere discussion webpage.

Discussion of trends:

The expression "statistically significant" is used throughout the submitted manuscript, however this expression is now recognized as being problematic and it should be abandoned and replaced by the more useful method of reporting all trends (with uncertainty, e.g. 95% confidence intervals) and all *p*-values, followed by a discussion of the trends and the author's opinion regarding their confidence in the trend values. This advice comes from a highly influential paper by Wasserstein et al. (2019), published in the journal, The American Statistician, that has already been cited over 1800 times (according to Web of Science). This advice was adopted by the first phase of TOAR (Tarasick et al., 2019) and is also used by TOAR-II. Some other recent papers on ozone trends that have taken this advice are: Chang et al., 2020; Cooper et al., 2020; Gaudel et al., 2020; Chang et al., 2022; Wang et al., 2022. Because these papers report all trend values, uncertainties, and all p-values, and also discuss the trend results, there is no confusion regarding the findings, and one does not even notice that the term "statistically significant" is not used at all. Table 3 of the TOAR-II statistical guidelines provides calibrated language for describing trends and uncertainty, similar to the approach of IPCC. In particular, Figure 16 only shows trend values that were deemed "significant" based on the p-value; this approach is inconsistent with the TOAR guidelines and the figure needs to report trend values from all sites, with 95% confidence intervals and p-values. Another TOAR guideline is that trends be reported in units of ppbv per decade so that trends can be compared between different studies. It's fine if you want to report your trend s in units of % $yr^{-1}$, but please also report the trends in ppbv $decade^{-1}$.

The "Data Availability" statement provides links for each of the monitoring stations, allowing the observations to be downloaded. However, it's not clear which link belongs to which station. It would be helpful if the links can be listed in a table, which also provides the station name and country.

**References:**

Chang, K.-L., et al. (2020), Statistical regularization for trend detection: An integrated approach for detecting long-term trends from sparse tropospheric ozone profiles, Atmos. Chem. Phys., 20, 9915–9938, https://doi.org/10.5194/acp-20-9915-2020

Chang, K.-L., O. R. Cooper, A. Gaudel, M. Allaart, G. Ancellet, H. Clark, S. Godin-Beekmann, T. Leblanc, R. Van Malderen, P. Nédélec, I. Petropavlovskikh, W. Steinbrecht, R. Stübi, D. W. Tarasick, C. Torres (2022), Impact of the COVID-19 economic downturn on tropospheric ozone trends: an uncertainty weighted data synthesis for quantifying regional anomalies above western North America and Europe, *AGU Advances, 3*, e2021AV000542. https://doi.org/10.1029/2021AV000542

Cooper, et al. 2020. Multi-decadal surface ozone trends at globally distributed remote locations. Elem Sci Anth, 8: 23. DOI: https://doi.org/10.1525/elementa.420

Gaudel, A., et al. (2020), Aircraft observations since the 1990s reveal increases of tropospheric ozone at multiple locations across the Northern Hemisphere. Sci. Adv. 6, eaba8272, DOI: 10.1126/sciadv.aba8272

Wang, H., Lu, X., Jacob, D. J., Cooper, O. R., Chang, K.-L., Li, K., Gao, M., Liu, Y., Sheng, B., Wu, K., Wu, T., Zhang, J., Sauvage, B., Nédélec, P., Blot, R., and Fan, S. (2022), Global tropospheric ozone trends, attributions, and radiative impacts in 1995–2017: an integrated analysis using aircraft (IAGOS) observations, ozonesonde, and multi-decadal chemical model simulations, Atmos. Chem. Phys., 22, 13753–13782, https://doi.org/10.5194/acp-22-13753-2022

Wasserstein, R. L., Schirm, A. L., and Lazar, N. A.: Moving to a world beyond p < 0:05, Am. Stat., 73, 1–29, https://doi.org/10.1080/00031305.2019.1583913, 2019.

---

## Author Comment (AC1)

**Response to Comment CC1**

Title: Long-term measurement of ozone concentrations in semi-natural African ecosystems
Author(s): Donnou et al.
MS No.: acp-2024-284

The authors would like to thank Erika von Schneidemesser for its relevant and helpful comments to improve the manuscript and give more informations about passive samplers monitoring.

**Comment 1** : Table 2: It would be more helpful if instead of the absolute numbers listed for data collection efficiency, a % was given, plus the n-value for total samples. This makes it more comparable across sites regardless of the n-value.

We will propose in the review process to include in the manuscript a modified table 2 as follow:

| Ecosystem | Station | Sampling period | Detection limit (ppb) | Data collection efficiency (%) | Total of samplers | Season | Measurement altitude (m) |
|---|---|---|---|---|---|---|---|
| Dry savanna | Ba
Ka
Ag
Bb
Da | 2000-2020
2001-2020
2005-2018
2016-2020
2012-2020 | | 93.5
86.7
82.6
94
83.7 | 248
240
132
50
104 | Dry season: Oct-May
Wet season: Jun-Sep | 1.5 |
| Wet savanna | La
Dj | 2001-2020
2005-2020 | | 94.2
92.5 | 240
186 | Dry season: Nov-Mar
Wet season: Apr-Oct | 1.5 |
| Forest | Zo | 2001-2020 | 0.1 | 86.7 | 240 | Dry season: Dec-Fev and July-Aug
Wet season: Mar-Jun and Sept-Nov | 3 |
| | Bo | 2001-2020 | | 68.3 | 240 | Dry season: Dec-Fev
Wet season: Mar-Nov | |
| Agricultural field | Mb | 2017-2020 | | 95.3 | 43 | Dry season: Jun-Oct and Jan-Fev
Wet season: Mar-May and Nov-Dec | 1.5 |
| Regional savanna/semi-arid | LT
Sk
Af | 1995-2015
2000-2015
1997-2015 | | 95.2
86
85.5 | 248
192
221 | Dry season: Apr-Sep
Wet season: Oct-Mar | 1.5 |
| | CP | 1995-2020 | | 90.7 | 248 | Dry season: Oct-Mar
Wet season: Apr-Sep | |

**Comment 2** : section 2.2. while the method for passive ozone sampling is well established, it would still be important to include text on the number of blanks that were evaluated and if/how any blank correction was done. Also, while monthly is mentioned, it is not explicit if the samples were all monthly and did this correspond to calendar months or were there different start and end dates for sampling rather than the first of the month - last day of the month? Finally, were these sampling times coordinated across the sites or did they vary?

We will add mentions in the revised text section 2.2 to answer the comment including these informations:

Sampling periods at the measurement sites were coordinated and passive samplers are exposed on a monthly basis using the calendar months. One blank dedicated to ozone is included in the expedition of samplers each two months on sites. In this way, the delay between field deployment and analysis are the same both blanks and exposed samples. All data presented in this paper are blank corrected.

---

## Author Comment (AC2)

**Response to Comment CC2**

Title: Long-term measurement of ozone concentrations in semi-natural African ecosystems

Author(s): Donnou et al.

MS No.: acp-2024-284

The authors would like to thank Owen COOPER for its relevant and helpful comments to improve the manuscript.

**Comment 1** : The expression "statistically significant" is used throughout the submitted manuscript, however this expression is now recognized as being problematic and it should be abandoned and replaced by the more useful method of reporting all trends (with uncertainty, e.g. 95% confidence intervals) and all p- values, followed by a discussion of the trends and the author's opinion regarding their confidence in the trend values. This advice comes from a highly influential paper by Wasserstein et al. (2019), published in the journal, The American Statistician, that has already been cited over 1800 times (according to Web of Science). This advice was adopted by the first phase of TOAR (Tarasick et al., 2019) and is also used by TOAR-II. Some other recent papers on ozone trends that have taken this advice are: Chang et al., 2020; Cooper et al., 2020; Gaudel et al., 2020; Chang et al., 2022; Wang et al., 2022. Because these papers report all trend values, uncertainties, and all p-values, and also discuss the trend results, there is no confusion regarding the findings, and one does not even notice that the term "statistically significant" is not used at all. Table 3 of the TOAR-II statistical guidelines provides calibrated language for describing trends and uncertainty, similar to the approach of IPCC.

Considering Wassertein et al. (2019) and recent ozone trends publications, we will propose in the revision process to include in the manuscript:
- Trends calculation per decade
- Modifications in the trend's discussion section taking into account the TOAR-II statistical guideline and presenting all calculated trends.

**Comment 2** : In particular, Figure 16 only shows trend values that were deemed "significant" based on the p-value; this approach is inconsistent with the TOAR guidelines and the figure needs to report trend values from all sites, with 95% confidence intervals and p-values. Another TOAR guideline is that trends be reported in units of ppbv per decade so that trends can be compared between different studies. It's fine if you want to report your trend s in units of % yr$^{-1}$, but please also report the trends in ppbv decade$^{-1}$

We will propose in the review process to include in the manuscript modified figures 15 and 16 as follow:

- Annual trends

[Figure]

**Figure 15.** Long-term annual linear trend of in situ O₃ concentrations over the period 1995-2020 at 95% confidence intervals calculated for 12 measurement sites representative of African dry savannas, wet savannas, forests and agricultural/semi-arid savannas.

- Seasonal trends

[Figure]

**Figure 16** Kendall's seasonal trend values of in situ O$_3$ concentrations over the period 1995-2020 at 95% confidence intervals calculated for 12 measurement sites representative of African dry savannas, wet savannas, forests and agricultural/semi-arid savannas.

**Comment 3:** The "Data Availability" statement provides links for each of the monitoring stations, allowing the observations to be downloaded. However, it's not clear which link belongs to which station. It would be helpful if the links can be listed in a table, which also provides the station name and country

We will propose in the reviewed manuscript a new table in the "Data availability" section to present sites and associated DOIs (links) (with the complete citation reference that will be included in the reference list.

**Data availability.** Dataset DOIs of O$_3$ observations for INDAAF sites (see complete citation in the reference list), available in the INDAAF database at https://indaaf.obs-mip.fr :

| Banizoumbou (Niger) | Katibougou (Mali) | Agoufou (Mali) | Bambey (Senegal) |
|---|---|---|---|
| https://doi.org/10.25326/608 Laouali et al. (2023) | https://doi.org/10.25326/604 Galy-Lacaux et al. (2023a) | https://doi.org/10.25326/610 Galy-Lacaux et al. (2023b) | https://doi.org/10.25326/609 Galy-Lacaux et al. (2023c) |
| Dahra (Senegal) | Lamto (Cote d'Ivoire) | Djougou (Benin) | Zoetele (Cameroon) |
| https://doi.org/10.25326/606 Galy-Lacaux et al. (2023d) | https://doi.org/10.25326/275 Galy-Lacaux et al. (2023e) | https://doi.org/10.25326/605 Akpo et al. (2023) | https://doi.org/10.25326/603 Ouafo-Leumbe et al. (2023) |
| Bomassa (Congo) | Mbita (Kenya) | Louis Trichardt (South Africa) | Skukuza (South Africa) |
| https://doi.org/10.25326/607 Galy-Lacaux et al. (2023f) | https://doi.org/10.25326/642 Galy-Lacaux et al. (2023g) | https://doi.org/10.25326/646 van Zyl et al. (2023a) | https://doi.org/10.25326/645 van Zyl et al. (2023b) |
| Cape Point (South Africa) | Amersfoort (South Africa) | | |
| https://doi.org/10.25326/644 van Zyl et al. (2023c) | https://doi.org/10.25326/647 van Zyl et al. (2023d) | | |

---

## Author Response (AR1)

**Response to Reviewers and Community**

Title: Measurement report: Long-term measurements of ozone concentrations in semi-natural African ecosystems

Author(s): Donnou et al.
MS No.: acp-2024-284

The authors would like to thank the editor, the referees and the scientific community for their valuable time and comments. They raise constructive and helpful comments for improving the manuscript. In general, we have modified the summary of the manuscript by adding information on the control and quality of the INDAAF data and presented all trend results by decade. In the introduction, we have included some previous projects during which ozone measurement campaigns were carried out, particularly in South Africa, and we have made the synthesis of the resulting work. In section 2.2, a further description is given of the analysis of passive sensors. The various reliability and quality control tests on the measurements made by the INDAAF passive sensors have been added to this section. Section 2.4 presenting the NOx and VOC emission inventories has been improved. The statistical tests of the Principal Component Analysis have been removed from section 2.5. In the "Results and Discussions" section, the analysis of seasonal variations in meteorological parameters has been discussed, and the results from the ozone climatology have been merged with the biogenic and anthropogenic emissions that explain the seasonality of ozone. The contextualisation of ozone concentrations in Africa and in the world has been improved and the discussion of annual and seasonal trends observed on a decadal scale at all INDAAF sites has been extended with relevant references in Africa.

Below, we provide a point-by-point response explaining how we have addressed each of the referees' comments. Note that our responses to the comments are in blue. We are confident that this second review stage has improved the manuscript. We look forward to the referees' opinion of the revised article, and to the editor's decision.

With our best regards, and our deepest gratitude to the referees and editor.

On behalf of all the co-authors

Sincerely yours
Venance Donnou and Corinne Galy-Lacaux

**REFEREES' REPORTS:**

**REFEREE RC1**

**COMMENTS 1: ABSTRACT**

Line 25, page 1: *Atmospheric levels*: could you be more precise? How do you define the atmosphere in terms of height?

**Author's response**: Line 25, page 1, we talked about ''atmospheric ozone concentrations'' and not ''*Atmospheric levels''*. We did not used the term ''Atmospheric levels'' in the manuscript.

Line 30, page 1: *Over the period 1995-2020, monthly ozone concentrations were measured at these sites using passive samplers*. How do you get ozone levels in the atmosphere using passive samplers?

**Author's response**: The passive sampling technique relies on laminar diffusion and on the chemical reaction of the atmospheric pollutant. The procedure using passive samplers is based on the following phenomena:
- the physical phenomenon of molecular diffusion;
- the chemical reaction between the studied gas molecules and those of the substance impregnated into the cellulose filter.
The gas transported into the passive sampler by molecular diffusion is then chemically trapped on an impregnated filter. The reaction product is recovered by extraction in a small volume of milli-Q water before being analysed by ion chromatography (ammonium, nitrates and sulphates). The measured dose is proportional to the gas concentration and can be expressed in ppb using the equation:

$$C=[(L/A).X.R.T]/(t.D.P)$$

L/A is the air resistance coefficient and depends on the passive sampler size and geometry. T is the monthly mean temperature over the sampling period (°K), X is the number of gas molecules trapped on the cellulose filter (μmol) during one month and R (constant gas) = 0.08206 $L.atm\ mol^{-1}\ K^{-1}$, P is the mean atmospheric pressure during the sampling period and t is the exposure time (one month, in s), D ($m^2\ s^{-1}$) is the diffusion coefficient of the gas in air. These details were mentioned in the first version of manuscript in section 2.2 at line. These details were mentioned in the studies of Adon et al. (2010), Ferm, (1991) and Martins et al. (2007) mentioned at section 2.2, line 156 in the first version of manuscript.

Line 32, Page 1: *Ozone levels in the wet season (in dry savanna) are higher and comparable to concentrations in the dry season (in wet savanna and forest)*. I find this sentence a bit confusing. Does it compare ozone levels in dry savanna with wet savanna and forest? Does ozone levels refer to the whole troposphere?

**Author's response**: In this sentence, we're not trying to make a comparison between ecosystems. The measurement level does not vary (1.5 m or 3 m) and the ozone levels mentioned refer to concentration levels. This sentence was rephrased in the revised version to avoid any confusion.

**Author's changes in manuscript**: Modified Line 34 to 35.

Lines 33-34, page 1: You mention East Africa, Southern Africa and Sahel, while before you group the sites depending on the ecosystem they belong to. How are these geographical regions defined? Does each region correspond to one ecosystem? With other words, it's not clear to me if the characteristics are due to the geographical location of the sites or because they belong to the same ecosystem.

**Author's response**: The ecosystem of a region is closely linked to its geographical location (latitude, altitude, proximity to water, relief, soil type, ocean currents, prevailing winds). In other words, the geographical location determines its climate, relief, water availability, soil composition, and other ecological factors that together shape local ecosystems. We can therefore say that the characteristics mentioned are due to the geographical situation and therefore to the ecosystem. In Table 1, section 2.1, we have listed the dry savanna sites in the Sahel region (Sahelian climate).

Lines 34-39, page 1: How do you support the dependence of ozone levels to different factors such as temperature, NOx emissions etc? How do you distinguish the effect of anthropogenic NOx emissions? Do you use any chemical transport model and sensitivity tests or these results are based on other studies?

**Author's response**: To support the dependence of ozone levels on various factors such as temperature, NOx emissions etc., we have carried out a principal component analysis (PCA) in this study, which is a method often used in air quality analyses to identify the main sources of pollutants emitted into the atmosphere. This statistical technique specifies the relationships between variables, the phenomena behind these relationships and the similarities between individuals. Anthropogenic NOx and VOC emissions were obtained from ECCAD emission inventories (https://eccad.aeris-data.fr).

Line 44, page 2: *The increasing trends are consistent with the increase in biogenic emissions at Zoétélé and $NO_2$ levels at Skukuza.* The increase in biogenic emissions and $NO_2$ levels results from this study?

**Author's response**: The increase in biogenic emissions is the result of this study. Trends were calculated for VOCs at Zoétélé and showed an increasing significance. On the other hand, the increase of $NO_2$ concentrations at Skukuza results from the work of Swartz et al, (2020a) carried out at the same sites during the same study period. Nevertheless, trends of $NO_2$ concentrations for all other sites have been calculated in this study.

*Swartz, J-S., van Zyl, P. G., Beukes, J. P., Galy-Lacaux, C., Ramandh, A., and Pienaar, J. J.: Measurement report: Statistical modelling of long-term trends of atmospheric inorganic gaseous species within proximity of the pollution hotspot in South Africa, Atmos. Chem.990 Phys., 20, 10637–10665, https://doi.org/10.5194/acp-20-10637-2020, 2020a.*

**COMMENTS 2: INTRODUCTION**

Lines 94-105, page 3: This part could be more informative by discussing the main finding for ozone in Africa, rather than listing the previous studies.

**Author's response**: This comment is now included in the revised manuscript.

**Author's changes in manuscript:** Modified Line 104 to 113, 116 to 121, 123 to 131, 133 to 145 and 148 to 149.

Line 112-114, page 3: *In the first objective, we first document the long-term (1995-2020 depending on the site) monthly, seasonal and interannual variability of $O_3$ concentrations on a regional scale at fourteen sites grouped by ecosystem (dry savannas, humid savannas, forests and agricultural/semi-arid savannas).* Grouped by ecosystem: does it mean that you merge measurements from different sites if they belong to the same ecosystem? Or do you analyse the measurements of each site separately?

**Author's response**: Measurements from different sites are not merged if they belong to the same ecosystem. We analyse ozone concentration measurements for each site separately.

**COMMENTS 3: MATERIAL AND METHODS**

2.1 Sampling sizes
Lines 141-143, page 4: The abbreviations for the sites are confusing and hard to remember. It makes it hard for the reader to understand the site you are referring to.

**Author's response**: The aim of using abbreviations was to simplify the reading. However, we have tried to adjust and minimize the use of abbreviations in the revised manuscript.

**Author's changes in manuscript:** Modified section 3.3 from Line 819 to 942 and section 4. Line 960, 970, 975 and 977 to 980.

Line 151, page 5: Figure 1: how are the sites ordered on the map? The map would be easier to read if you order the sites by increasing longitude.

**Author's response**: Sites are classified on the map by ecosystem. First, we have the dry savanna (from Banizoumbou to Dahra), the humid savanna (from Lamto to Djougou), the forests (from Zoétélé to Bomassa) and the agricultural/semi-arid savannas from Mbita to Cape Point.

2.2 Passive sampling and chemical analysis
Table 2, page 5: Why is the measurement altitude different for the forest sites?

**Author's response**: The measurement altitude is different for the forested sites because of the canopy with is more important at these sites. The forests present significant dry depositions fluxes on soil, foliage and trees. We decided also to co-locate aerosol measurements and meteorological parameters at this height.

In Lamto, the sampling period is 2001-2020 and the data collection efficiency is 226/240. What do the numbers mean exactly? Please clarify. The number of months of the sampling period is 228. Is it 226 monthly measurements taken into account for the period 2001-2020?

**Author's response**: In Lamto for example, the collection efficiency is 226/240. The 240 corresponds to the total number of monthly ozone concentration data over 20 years (12 x 20) and 226 indicates the number of ozone concentration data validated after measurement. Indeed, certain data are extracted from the series when they are aberrant or below the detection limit. These are therefore 226 monthly measurements taken into account for the period 2001-2020.
We have proposed to include more details in the new text about the efficiency of data collection.

**Author's changes in manuscript:** Modified Line 226 to 229 (table 2).

The sampling period is quite different depending on the site (e.g 2017-2020 and 1995-2020). How do you assess the statistical significance of the measurements or the trends? How do you compare trends or ozone levels while the sampling period is different depending on the site?

**Author's response**: The sampling period is different depending on the date of the first measurements implemented on sites. To assess the statistical significance of measurements or trends, the Mann Kendall statistical method used in this paper requires that the used data cover at least 10 years. On this basis, the trends were calculated at 95% confidence intervals. When pvalue ≤ 0.05, the trend is statistically significant. Following the TOAR guidelines we can use the following expressions: pvalue ≤ 0.01, the tendance is with very high certainty, if 0.01 < pvalue ≤ 0.05, the trend is with high certainty, if 0.05 < pvalue ≤ 0.10, the trend is with medium certainty, if 0.10 < pvalue ≤ 0.33, the trend is with low certainty and if pvalue > 0.33, the trend is with very low certainty.
Therefore, the Dahra site, with a total of 9 years of measurement, is an exception for the calculation of trends. Bambey site (05 years of measurement) and Mbita (04 years of measurement) were not included in trend calculations. The trends or ozone concentrations are calculated with their standard deviations and comparisons are made with other works including these differences in measurement periods which could also explain the differences observed between the average concentrations.

2.3 Meteorological parameters and leaf area index

Lines 173-184, page 6: What difference does it make that you use different datasets for the meteorological parameters depending on the site? Have you tried to compare data from AMMA-CATCH database/INDAAF database with the ERA5 data for the common location? What is the consistency between the different datasets?

**Author's response**: The sites where ERA5 datasets are used are those where meteorological data are not available in situ. When we compare the data from ERA5 with those measured in situ in the AMMA-CATCH and INDAAF databases at the same site, we see that the data are consistent with small margins of error. For example, for the Banizoumbou site, we obtained error estimates (RMSE) of the order of $9.9 \times 10^{-3}$ °C for temperature, $4.8 \times 10^{-3}$ for humidity and $2.3 \times 10^{-1}$ mm for rainfall. At the Bambey site, where in situ radiation measurements are available, the estimated errors are of the order of $6.4 \times 10^{-2}$ J/m². The correlation between in situ and ERA5 data at these sites is 0.96 for precipitation, 0.99 for humidity, 0.80 for radiation and 0.99 for temperature.

**Author's changes in manuscript:** Modified Line 268 to 275.

2.4 NOX and VOC emissions

Lines 197-199, page 7: *The biogenic NO fluxes used are model outputs in reference to the work of Delon et al. (2010, 2012). They were filtered in the eastern grid from 5° S to 20° N in latitude, and 20° W to 30° E in longitude over the period from 2002 to 2007 and cover only the Ba, Ka, Ag, La, Dj, Zo and Bo sites.* What does the phrase "*model outputs in reference to the work of Delon et al. (2010, 2012)."* mean? Please clarify. I interpreted this phrase as you used the model output produced by Delon et al. 2010 and 2012? If yes, have you considered using more up-to-date model output and emission inventories? Why do you use data for the period from 2002 to 2007? Why does this period not match the sampling period of the sites (Table 2)?

**Author's response**: The biogenic NO data used in this paper come from the modelling results produced by Delon et al (2010 and 2012). These data were used due to a lack of inventory data on the production of biogenic NO emissions. New simulations for our periods of study are not available.

2.5 Statistical analysis

General comment: The application of PCA to identify factors controlling ozone concentration raises significant concerns. PCA assumes linearity in variable relationships, which may not hold true for ozone formation processes involving precursor emissions and meteorological conditions. This raises concerns about the interpretability and validity of the results. Considering alternative methods like machine learning techniques designed to handle non-linear relationships would be more appropriate for this type of analysis. At least, it's crucial to acknowledge the limitations of PCA in establishing causal relationships between ozone and its controlling factors.

**Author's response**: In this study, the Principal Component Analysis method was used to identify the dominant meteorological precursors and variables in each of the ecosystems studied. However, we agree with the reviewer that our results deserve further investigation due to the non-linear relation observed between ozone and its precursors. However, under certain specific (remote sites for example) and simplified conditions, it is possible to observe a linear or quasi-linear relationship with low precursor concentrations or in regimes of low sensitivity (e.g. an area with low NOx concentrations but sufficient VOCs may lead to a linear increase in ozone up to a certain point). The presence of these conditions was not examined in this study. We therefore propose, as part of the revision process, to

remove PCA analysis in the revised manuscript (section 3.4 in the first version) and we just keep corelations coefficients calculation between ozone, precursors and meteorological variables (table 5 in the first version). These parts will be merged with ozone climatology (section 3.2).
This decision is also coherent with the comments and the demand of reviewer 2 to globally shorten the paper.

**Author's changes in manuscript:** Modified Line 167 to 171, 306 to 314, 400 to 422, 454 to 465, 503 to 507 and 515 to 534.

Line 221, page 7: The use of some references (e.g. Tsuyuzaki et al., 2020) to justify PCA analysis seems outside the usual scope of atmospheric research. Considering established applications of PCA in atmospheric science for analyzing ozone surface data would be more relevant

**Author's response**: It is now in the revised manuscript in relation to the previous observation.

**Author's changes in manuscript:** Modified Line 306 to 314.

**COMMENTS 4: RESULTS AND DISCUSSION**

3.1 Meteorological and biophysical parameters variation
Lines 231-245, pages 7-8: This paragraph focuses on describing the content of Figure 2. It would be more informative to analyze and explain the underlying mechanisms driving the patterns observed in Fig. 2. Several relevant studies by LAERO (e.g. Adon et al. 2010; Lannuque et al., 2021; Sauvage et al., 2005; 2007; Tsivlidou et al. 2023) explore the African meteorology (e.g. ITCZ, Harmattan, monsoon). Using these explanations, the authors could enrich the discussion and explain the seasonal variations evident in Figure 2.

Adon, M., Galy-Lacaux, C., Yoboué, V., Delon, C., Lacaux, J. P., Castera, P., Gardrat, E., Pienaar, J., Al Ourabi, H., Laouali, D., Diop, B., Sigha-Nkamdjou, L., Akpo, A., Tathy, J. P., Lavenu, F., and Mougin, E.: Long term measurements of sulfur dioxide, nitrogen dioxide, ammonia, nitric acid and ozone in Africa using passive samplers, Atmos. Chem. Phys., 10, 7467–7487, https://doi.org/10.5194/acp-10-7467-2010, 2010.

Lannuque, V., Sauvage, B., Barret, B., Clark, H., Athier, G., Boulanger, D., Cammas, J.-P., Cousin, J.-M., Fontaine, A., Le Flochmoën, E., Nédélec, P., Petetin, H., Pfaffenzeller, I., Rohs, S., Smit, H. G. J., Wolff, P., and Thouret, V.: Origins and characterization of CO and O3 in the African upper troposphere, Atmos. Chem. Phys., 21, 14535–14555, https://doi.org/10.5194/acp-21-14535-2021, 2021.

Sauvage, B., Thouret, V., Cammas, J.-P., Gheusi, F., Athier, G., and Nédélec, P.: Tropospheric ozone over Equatorial Africa: regional aspects from the MOZAIC data, Atmos. Chem. Phys., 5, 311–335, https://doi.org/10.5194/acp-5-311-2005, 2005

Sauvage, B., Gheusi, F., Thouret, V., Cammas, J.-P., Duron, J., Escobar, J., Mari, C., Mascart, P., and Pont, V.: Medium-range mid-tropospheric transport of ozone and precursors over Africa: two numerical case studies in dry and wet seasons, Atmos. Chem. Phys., 7, 5357–5370, https://doi.org/10.5194/acp-7-5357-2007, 2007.

Tsivlidou, M., Sauvage, B., Bennouna, Y., Blot, R., Boulanger, D., Clark, H., Le Flochmoën, E., Nédélec, P., Thouret, V., Wolff, P., and Barret, B.: Tropical tropospheric ozone and carbon monoxide

distributions: characteristics, origins, and control factors, as seen by IAGOS and IASI, Atmos. Chem. Phys., 23, 14039–14063, https://doi.org/10.5194/acp-23-14039-2023, 2023.

**Author's response**: Considering Adon et al., (2010), Lannuque et al., (2021), Sauvage et al., (2005, 2007), Tsivlidou et al. (2023) publications and recent papers about African meteorology, we propose in the revised manuscript a section to analyze and explain the underlying mechanisms driving the observed patterns observed in Fig. 2.

**Author's changes in manuscript:** Modified Line 343 to 357.

Figure 2, page 9: the range of the left and right axes differ depending on the site. This makes it harder for the reader to make comparisons between the different locations.

**Author's response:** It is now in the revised manuscript.

**Author's changes in manuscript:** Modified 358 to 365.

Figure 3, page 10: Too low $O_3$ surface concentrations compared to Tsivlidou et al 2023- Sahel (their fig. S3 panel 1a). Do you have any idea why? Figure 6, page 13: Too low concentrations compared to Tsivlidou et al., 2023- Central Africa (their fig. 3 panel 1b)? Any idea why?

**Author's response:** In the work of Tsivlidou et al (2023), the ozone concentration values obtained by the authors are higher than those obtained in our study in West and Central Africa. This finding can be explained by the fact that the sites studied in Tsivlidou et al (2023) are urban or close to cities where concentrations of ozone precursor pollutants are higher (NOx and anthropogenic VOCs). In addition, the measurements taken during aircraft take-off and landing are instantaneous measurements, unlike the INDAAF measurements, which represent continuous monthly integration and monthly average. In addition to the above explanations, in Central Africa, INDAAF measurements are taken in the heart of the forest in the presence of the canopy, which constitutes a sink for ozone concentrations, unlike measurements by aircraft, where ground recordings are taken near the runway.

Table 3, page 11: What does 'moy' stand for?

**Author's response:** The average ozone concentration is referred to as 'moy'. It should be 'Avg' and has been corrected in the revised version of the manuscript.

**Author's changes in manuscript:** Modified Line 443 (Table 3).

Figure 9, page 15: The location markers in Fig. 9 (especially locations 1–7) do not match the ones from fig. 1 (page 5).

**Author's response:** On Figure 9, the location markers are larger to make it easier for readers to distinguish the colours. This caused the locations to overflow. We have corrected this aspect, which was raised by the evaluator in the revised manuscript. On Figure 1, however, the map has been reworked. Not all the markers had been positioned correctly.

**Author's changes in manuscript:** Modified Line 571 to 573 (Fig. 12) and 197 to 200 (Fig. 1).

3.3.1 NOx and VOC anthropogenic emissions & 3.3.2 NOx and VOC natural emissions: To streamline the flow of the text, it might be beneficial to consider consolidating sections 3.3.1 and 3.3.2. Some of the key reasons for the ozone peaks and their seasonal patterns have already been established earlier. Merging these sections would reduce redundancy and maintain a focused narrative.

3.4.2 Characterisation of $O_3$ precursor emission sources and studies of correlations. Similar to the previous comment above, there appears to be some repetition of explanations and citations within 3.4.2. To enhance readability, particularly given the paper's length, combining this section with other paragraphs discussing ozone features (e.g., 3.2.1, 3.3, 3.4.2) could be explored. This would allow for a more concise and focused presentation of the findings. For improved readability and reduced redundancy, I recommend exploring the possibility of consolidating sections discussing ozone features. This could involve merging 3.2.1, 3.3, and 3.4.2 into a single, well-organized section that succinctly explains the observed phenomena.

**Author's response:** The reviewer's comments are relevant and we decided to merge these sections in the revised manuscript.

**Author's changes in manuscript:** Modified Line 400 to 409, 412 to 423, 436 to 440, 454 to 465, 477 to 482, 495 to 500, 503 to 507, 515 to 517, 519 to 534 and 560 to 566.

3.5.1 Annual trends. Figure 15, page 26: What could cause the maxima of $O_3$ over Katibougou (a and b) and Banizoumbou (d) in 2014?

**Author's response:** The maxima observed at Katibougou in 2014 can be attributed to high NO emissions during the rainy season (June 2014 to September 2014). This is confirmed by the trend observed during the wet season, unlike the dry season, where the maximum in the data series over the last decade was in 2015. 2014 was a year marked by one of the lowest rainfall amounts recorded after 2010 (789 mm, compared with the highest amount estimated at 1289 mm in 2020). This could mean a delay in the start of the rainy season. When the dry season is longer, the start of the rainy season leads to more intense biogenic NO emissions. At Banizoumbou during the dry season, the 2014 maximum would indicate greater quantities of anthropogenic NOx emissions during that year.

**Conclusions.** Line 628, page 28: What do you mean that the different *photochemical regimes* are discussed in the study?

**Author's response:** By 'photochemical regimes' we mean the chemical mechanisms by which ozone is formed in the atmosphere under the influence of the meteorological variables that act as catalysts for these chemical reactions. These regimes in the case of this study vary from site to site and have been addressed in the manuscript by calculating correlations between ozone, its precursors and meteorological variables.

**Technical suggestions:** Line 76, page 2: (Gaudel et al., 2018; Fleming et al., 2018; Mills et al., 2018) should be (Fleming et al., 2018; Gaudel et al., 2018; Mills et al., 2018)

**Author's response:** This comment has been included in the revised version of the manuscript.

**Author's changes in manuscript:** Modified Line 153 to 154.

**COMMENTS 1**

**Abstract.** A major concern about the ozone measurements is quality control of the passive sampler data (more below). A sentence or two that documents the reliability of the INDAAF station ozone observations is needed in the Abstract.

**Author's response**: In the abstract, the revised manuscript includes information about the reliability of the INDAAF station ozone observation.

**Author's changes in manuscript:** Modified Line 30 to 32.

Lines 33-34 – Sentence is ambiguous as written. Do you mean southern Africa has higher mean ozone concentrations or greater seasonal variations in ozone in east Africa?

**Author's response**: In this sentence, we mean that seasonal variations are low in East Africa and that ozone concentrations are higher in Southern Africa.

**Author's changes in manuscript:** Modified Line 36 to 37.

The Sahel region sites are not defined although it is recommended that the two sentences in Lines 34 to 37 be omitted.

**Author's response**: This comment was taken into account in the revised version.

**Author's changes in manuscript**: Modified Line 37 to 43.

Line 40 – specify the dates of the trend calculations.

**Author's response**: It is now in the revised manuscript.

**Author's changes in manuscript**: Modified Line 43 to 44 and 49.

The lines 42 to 44 refer to changes in VOC and $NO_2$ that appear to be assumptions. That is, the paper gives no measurements to support this statement for these 2 INDAAF sites. These lines should be removed

**Author's response**: We were calculated trends test for VOCs at Zoétélé in this study. The results are given in section 3.5.2 of the first version. For $NO_2$, these changes are indicated in the work of Swartz et al, (2020a) who investigated $NO_2$ concentrations trends at the same sites over the same period of study.

*Swartz, J-S., van Zyl, P. G., Beukes, J. P., Galy-Lacaux, C., Ramandh, A., and Pienaar, J. J.: Measurement report: Statistical modelling of long-term trends of atmospheric inorganic gaseous species within proximity of the pollution hotspot in South Africa, Atmos. Chem.990 Phys., 20, 10637–10665, https://doi.org/10.5194/acp-20-10637-2020, 2020a.*

**COMMENTS 2**

**Sections on NO, VOC, BVOC**: Three reasons for recommending nearly all the material in Sections 2.4, 3.3, and 3.4 be removed from the current paper. (a) The paper is too long and yet, important information is still lacking. Example: explanation of the method in 3.4.1 needs to be expanded. (b) It is not clear to what extent values for the species NOx, VOC, BVOC, their seasonality of emissions, changes over time, are from model(s) and/or experiments and if so, which ones? Explain. It is important when referring to published relationships between ozone and the precursors to distinguish between links based on measurements (clarify with references to the data sources and publications) and "trends" based on models that are of unknown accuracy. Related to this point, the relationships for species and ozone in Figure 12 are not convincing, in contrast to those in Figure 2 where there are definite links of ozone with climatic/meteorological parameters. (c) The correlations in Figure 14 and Table 5 are intriguing but not very clear. For example, previous studies demonstrate biomass fire impacts on sites in northeast South Africa but how is the attribution to VOC and NO combustion made here? In summary, the Reviewer finds good potential in linking ozone with the precursors and processes (biogenic vs combustion NOx and VOC, for example) but the analyses in this paper are confusing in some places and are not convincing in other sections. Recommend that the authors move these analyses to a second paper that presents ozone and precursor relationships with more rigor and *observational* evidence.

**Author's response**: As explained in the section 2.4, $NO_x$ and VOC emissions from biomass combustion sources were downloaded from the fourth generation of the Global Fire Emissions Database (GFED) at 1°×1°. GFED4 has a higher spatial resolution of 0.25° and was obtained using satellite information on fire activity and vegetation productivity from 1997. MEGAN is an inventory with a basic resolution of around 1 km$^2$ (kg m$^{-2}$ s$^{-1}$), the determining variables of which are derived from models and satellite and ground observations, enabling simulations to be carried out on a regional and global scale. MEGAN estimates the net emission rate of VOCs, other trace gases and aerosols from terrestrial ecosystems into the atmosphere above the canopy. It takes into account the emission factor, the emission activity factor and the factor that explains production and losses within the plant canopy.
These inventories emissions are widely used and Global Fire Emissions Database GFED have been recommended recently by Stauffer et al. (2024) to study potential shifts in the timing and spatial patterns of biomass burning and ozone precursor emissions in the tropics.
The correlations in table 5 give a precise idea, depending on the site, of the possible links between ozone, its precursors and meteorological parameters. These links are discussed in greater detail in the revised version. In the revised version, the section concerning the PCA method will be deleted (section 3.4). Only the correlation results will remain in the paper and be merged with the first objective, which integrates ozone climatology. Section 2.4 are more documented concerning the origin of the emission inventories used in the paper.

**Author's changes in manuscript:** Modified Line 167 to 171, 292 to 295, 301 to 303, 306 to 314 and 651 to 816.

**COMMENTS 3**

 **There are** a number of places where statements should be made more accurately. Many suitable references are given in the paper but sometimes in a misleading way. Examples from Section 1 follow: a. Line 61. Since Zhang et al. (2016) both modeling and observational studies have shown that ozone trends are not uniform regionally or seasonally, i.e. even in the tropics a number of sites with ozonesonde profiles exhibit no trend (Thompson et al., 2021). A study with sondes over equatorial southeast Asia, published in the TOAR II collection by Stauffer et al. (2024), shows no definite ozone trend annually but a 6-8%/decade increase limited to 3 months/year. Insert words to this effect on Line 62 after Adon et al., 2010. The references are here:

Thompson, A. M., R. M. Stauffer, K. Wargan, J. C. Witte, D. E. Kollonige, J. R. Ziemke, Regional and seasonal trends in tropical ozone from SHADOZ profiles: Reference for models and satellite products, J. Geophys. Res., https://agupubs.onlinelibrary.wiley.com/doi/10.1029/2021JD034691, 2021.

Stauffer, R. M., A. M. Thompson, D. E. Kollonige, N. Komala, H. Khirzin Al-Ghazali, D. Y. Risdianto, A. Dindang, A. F. bin Jamaluddin, M. Kumar Sammathuria, N. Binti Zakaria, B. J. Johnson, P. D. Cullis, Dynamical drivers of free-tropospheric ozone increases over equatorial Southeast Asia, Atmos. Chem. Phys., https://doi.org/10.5194/acp-24-5221-2024.

**Author's response**: The references mentioned by the reviewer have been integrated into the manuscript

**Author's changes in manuscript:** Modified Line 69 to 73.

b. Line 63-65. Laban et al. (2018), which describes patterns of South African ozone at a number of sites that are both rural and anthropogenically influenced, emphasizes that the highest ozone is in the dry season in the semi-arid areas in the Highveld, largely due to a maximum in domestic fuel usage for winter heating, not biosphere interactions, as the paper states.

**Author's response**: It is now included in the revised manuscript

**Author's changes in manuscript:** Modified Line 75.

c. Lines 70-75. The Reviewer is not sure if the continents of South America and Africa were compared that Africa (at least sub-Saharan Africa) is "the least studied." On line 73 it is more accurate to say "may be the least studied continent." Over the past ~30 years there have been many large African field campaigns conducted (AMMA, EXPRESSO, SAFARI/TRACE-A, ORACLES, SAFARI-2000) such that hundreds of articles have been published on African air quality and environment. Links to dynamical factors affecting ozone seasonality (Diab et al., 1996; 2003; 2004), interannual variability in ozone related to ENSO (Balashov et al., 2014) and widespread impact of biomass and domestic fires in southern Africa are well-established. On latter, refer to detailed analyses of ozone and related measurements in Special Issues of J. Geophysical Res. on SAFARI/TRACE-A, and SAFARI-2000. Southern Africa has been the major arena of these ozone studies and South Africa has had several high-quality monitoring programs. In Line 65 it is more accurate to say "With the exception of South Africa, ozone variability is poorly documented on the African continent." For example, in North American and European journals, North Africa probably has the smallest number of articles. However, having said that there are many African studies about ozone and air quality, the number of measurements publicly available may be very small. The authors can point out that the INDAAF is among the few datasets that are available to the scientific community.

Diab et al: Diab, R. D., A. M. Thompson, M. Zunckel, G. J. R. Coetzee, J. B. Combrink, G. E. Bodeker, J. Fishman, F. Sokolic, D. P. McNamara, C. B. Archer, and D. Nganga, Vertical ozone distribution over southern Africa and adjacent oceans during SAFARI-92, J. Geophys. Res., 101, 23,809-23,821, 1996; Diab, R. D., A. Raghunandran,

A. M. Thompson, V. Thouret, Classification of tropospheric ozone profiles over Johannesburg based on MOZAIC aircraft data, Atmos. Chem. Phys., 3, 713-723, 2003; Diab, R. D., A. M. Thompson, K. Mari, L. Ramsay, G. J. R. Coetzee, Tropospheric Ozone Climatology over Irene, South Africa from 1990 - 1994 and 1998-2002, J, Geophys. Res., 109, D20, D20301, doi: 10.1029/2004JD004293, 2004.

**Author's response**: Thanks for mentioning these studies. They are now included in the introduction of the revised manuscript.

**Author's changes in manuscript:** Modified Line 83, 104 to 113, 151 to 153.

d. Lines 100-105. A number of references are given about studies and campaigns but little information about the findings of each that are relevant to the authors' study. What the reader wants to see is "what do we know from the prior campaigns?" "Do they agree with one another?" "What are new INDAAF results that confirm, contradict or complement the earlier findings?" The list in these lines is not useful without connecting the background to the current paper.

**Author's response**: It is now included in the revised version.

**Author's changes in manuscript:** Modified Line 116 to 121, 124 to 127, 128 to 131, 134 to 145 and 148 to 149.

e. In like manner to comment (b) above about Line 65, LINE 106 is not correct. There have been a number of studies with South African ozone and related data. Line 106 should say "With the exception of South Africa very little information is available on the long-term evolution of O3 chemistry over Africa. The impact of meteorological parameters and atmospheric chemistry… and the analysis of long-term trends *is only partially explained.*" In other words it is not correct to say the trends are "unexplored." The authors cite Balashov et al. (2014) (line 537) which determines trends for 5 South African sites with high quality surface O3 data, over ~15 years; see also Martins et al, 2007. Trends over a longer period, ~1990 to 2011 in ACP, described the seasonality of free tropospheric O3 trends over Irene [Pretoria], South Africa with both IAGOS and ozonesonde data (Thompson et al., Atmos. Chem. Phys., 14, 9855-9869, 2014; the latter paper corrected a sampling error in Clain et al., 2009). In ACP, Gaudel et al. (egusphere-2023-3095) use IAGOS African aircraft data to estimate trends over a number of sub-Saharan African cities for the period ~1995-2019.

**Author's response**: The reviewer's observations have been included into the introduction of revised manuscript.

**Author's changes in manuscript:** Modified Line 152 to 153 and 159 to 160.

**COMMENTS 4**

**Section 2.2.** This section, fundamental to the quality of the paper, is inadequate. It is not enough to cite previous papers to establish the accuracy of the INDAAF ozone record. Uncertainties in ozone mixing ratios for typical samples need to be provided. Uncertainties are also needed in the graph of trends in Figure 15. To give further confidence in the ozone time series for the INDAAF sites, comparisons of sampler ozone with independent ozone measurements should be made for sites where the latter data are available within the 1995-2020 period. Examples: ozone from Irene, South Africa, sondes at the surface and Welgegund (continuous ozone monitor) can be compared to LT; if ozone from the WMO/GAW station at Cape Point is available from an independent analyzer, a comparison of trends from such an instrument during the INDAAF period should be made. Although Nairobi observations are not co-located with the Mbita INDAAF data, their ozone should display similar seasonal patterns. Nairobi ozonesonde data are available from SHADOZ (1998-2023; https://tropo.gsfc.nasa.gov/ shadoz); surface monitor ozone data may also be available.

**Author's response**: The quality assurance and control tests carried out on the INDAAF passive samplers have been described in section 2.2.

**Author's changes in manuscript:** Modified Line 215 to 218 and 230 to 255.

**COMMENTS 5**

**Sections 3.1 and 3.2.** The analyses corresponding to Figures 3-9 (with Table 3) are very good. In later Sections, e.g., 3.4.2.3, there are references to meteorological influences on South African ozone seasonality (humidity, temperature; Balashov et al., 2014; Laban et al., 2018; 2020). The Introduction and/or earlier sections would be strengthened by moving some of these references forward.

**Author's response**: It is now included in the introduction of revised version.

**Author's changes in manuscript:** Modified Line 123 to 127 and 128 to 131.

Figures 4, 6, 8. These Figures originate from averaging the values in Figures 3, 5, 7, respectively, no? Please explain in the captions.

**Author's response**: Figures 4, 6, 8 result from averaging the values in Figures 3, 5 and 7. It is now included in the captions.

**Author's changes in manuscript:** Modified Line 432, 493 and 558 to 559.

Figure 9. Likewise, although parts of the text define the periods of "wet", "dry" season, it would be useful to repeat or summarize those definitions in the Figure 9 caption.

**Author's response**: It is now included in the Figure 9 caption.

**Author's changes in manuscript:** Modified Line 577 to 581.

Page 12. Discussion. References to Nepal climatology are not relevant here. Remove.

**Author's response**: We removed the references about Nepal climatology in the manuscript.

**Author's changes in manuscript:** Modified Line 590 (Table 5).

Pages 16 and 17. These comparisons are not relevant to the paper. There is no value to Table 4 and it should be deleted along with Lines 380-391. We don't know that the same analytical methods are used as at INDAAF sites. The dates are not a match for the INDAAF period in many cases. What is the point of making comparisons with Arctic, Antarctic, midlatitude sites?

**Author's response**: As stated in the paragraph from lines 371 to 375 in the first version, we have compared the African ozone concentrations reported in this study with studies carried out in Africa and worldwide over the last 20 years. We have also stated in this paragraph that the bibliographic synthesis includes studies where the data measurement methodology was clearly described and that only sites where concentrations were measured by passive samplers are listed. The table 5 contains ozone concentration values in the 4th column ($O_3$ ppb) and we think that this table in the framework of this study and gives an overview of ozone concentrations in the world.

Figure 10. This Figure *is* relevant to the discussion of INDAAF ozone climatology. In particular, the INDAAF data for South Africa should be added to the Figure and those values should be compared to earlier publications with South African data. Reasons for similarities and differences should be discussed, including how sampling systems for the non-INDAAF data compare to INDAAF measurements in similar parts of South Africa.

**Author's response**: It is now included in the revised version.

**Author's changes in manuscript:** Modified Line 593 to 641.

**COMMENTS 6**

**Sections 3.3, 3.4.** As mentioned above, most of this material should be removed or saved for a separate manuscript because it is about precursor variability and trends for which necessary explanations, references and background are beyond the scope of this paper.

**Author's response**: We have included the reviewer's observations in the revised version and deleted the part concerns the PCA methods.

**Author's changes in manuscript:** Modified Line 167 to 171, 306 to 314, Line 400 to 409, 412 to 423, 436 to 440, 454 to 465, 477 to 482, 495 to 500, 503 to 507, 515 to 517, 519 to 534 and 560 to 566 and 651 to 815.

Section 3.5. The focus on $O_3$ trends is one of two major results of this paper but the discussion is incomplete (no South African data displayed, for example).

**Author's response**: The discussion of $O_3$ trends has been extended in the revised manuscript.

**Author's changes in manuscript:** Modified Line 835 to 876, 906 to 911, 922 to 924, 926 to 942.

Some points raised in Section 3.5.1 refer to BVOC trends (? From models; if so give short explanation).

**Author's response**: Based on the emission inventory data, VOC trends were calculated using the Mann Kendall statistical method. The origin of the emission inventories was discussed in above comments 2: Sections on NO, VOC, BVOC.

There have been trends papers with African ozone data that cover most of the INDAAF period analyzed here, ~mid-late 1990s through 2020: Gaudel et al., 2018; 2020; in review- egusphere-2023-3095; Thompson et al., 2021. How do the authors' INDAAF trends agree? Discuss in more depth. There should be references on seasonality of fire impacts from the Piketh and van Zyl-Beukes groups. Expand the literature search. It may be possible to add truly relevant articles to the bibliography and remove references on biogenic emissions or model "trends" that are not data.

**Author's response**: We have included in the manuscript an improved version of the trends section.

**Author's changes in manuscript:** Modified Line 835 to 876, 906 to 911, 922 to 924, 926 to 942.

**Comments 7: Miscellaneous Comments and Questions:**
**Abstract**. Place names. Location, Country; - separate with semi-colon. Banizoumbou, Niger; Katibougou and Agoufou, Mali; etc.

**Author's response**: It is now included in the revised manuscript.

**Author's changes in manuscript**: Modified Line 28 to 30.

**Line 53** Don't begin sentence with $O_3$, use "Ozone"

**Author's response**: It is now included in the revised manuscript.

**Author's changes in manuscript:** Modified Line 60.

**Line 94**. "Previous studies" – delete "existing"

**Author's response**: It is now included in the revised manuscript.

**Author's changes in manuscript:** Modified Line 104.

**Line 112**. Delete the 2nd occurrence of "first"

**Author's response**: It is now included in the revised manuscript.

**Author's changes in manuscript:** Modified Line 163.

**Lines 116-120**. Recommend omitting 3 sentences. A second objective is then: "In the 2$^{nd}$ objective, we use non-parametric statistical tests to assess seasonal and annual trends in O$_3$ in the context of other trend analyses of Africa ozone (Thompson et al., 2014; Gaudel et al., 2018; Gaudel et al., 2024)." Option to add: "In a companion/separate article INDAAF ozone trends are linked to potential changes in NO2, VOC, BVOC."

**Author's response**: From the arguments we developed above concerning the data on NOx and VOC emission inventories (comments 2) and including the reviewer's recommendations, we proposed two specific objectives in the revised version of the manuscript. The first concerns ozone climatology in addition, the correlations between O$_3$, its gaseous precursors and meteorological parameters. The second objective will take long-term trends into account. The study about the impact of meteorological parameters (temperature, humidity, precipitation, radiation) and atmospheric chemical precursors (NOx and VOC) on photochemical O$_3$ production, using principal component analysis, have be deleted from the revised version of the manuscript.

**Author's changes in manuscript:** Modified Line 167 to 171.

**Table 2.** For South Africa LT, Sk, Af – End point is 2015. Did the program end there or are there data taken since 2015 that are not publicly available?

**Author's response**: Passive samplers were deployed at Louis Trichardt, Amersfoort and Skukuza after 2015 up until 2017. However, none of these passive samplers has been analysed, since funding of measurements at these sites was discontinued at the end of 2015. Passive sampling continued at Cape Point after 2015 and is still ongoing. These samplers have also not yet been analysed due to lack of funding and capacity.

**COMMUNITY CC1**

**Comment 1**: Table 2: It would be more helpful if instead of the absolute numbers listed for data collection efficiency, a % was given, plus the n-value for total samples. This makes it more comparable across sites regardless of the n-value.

**Author's response**: It is now included in the manuscript a modified table 2.

**Author's changes in manuscript**: Modified Line 226 to 228 (Table 2).

**Comment 2**: section 2.2. while the method for passive ozone sampling is well established, it would still be important to include text on the number of blanks that were evaluated and if/how any blank correction was done. Also, while monthly is mentioned, it is not explicit if the samples were all monthly and did this correspond to calendar months or were there different start and end dates for sampling rather than the first of the month - last day of the month? Finally, were these sampling times coordinated across the sites or did they vary?

**Author's response**: Sampling periods at the measurement sites were coordinated and passive samplers are exposed on a monthly basis using the calendar months. One blank dedicated to ozone is included in the expedition of samplers each two months on sites. In this way, the delay between field

deployment and analysis are the same both blanks and exposed samples. All data presented in this paper are blank corrected.

**Author's changes in manuscript**: We added mentions in the revised text section 2.2.1 from Line 215 to 218.

**COMMUNITY CC2**

**Comment 1**: The expression "statistically significant" is used throughout the submitted manuscript, however this expression is now recognized as being problematic and it should be abandoned and replaced by the more useful method of reporting all trends (with uncertainty, e.g. 95% confidence intervals) and all p- values, followed by a discussion of the trends and the author's opinion regarding their confidence in the trend values. This advice comes from a highly influential paper by Wasserstein et al. (2019), published in the journal, The American Statistician, that has already been cited over 1800 times (according to Web of Science). This advice was adopted by the first phase of TOAR (Tarasick et al., 2019) and is also used by TOAR-II. Some other recent papers on ozone trends that have taken this advice are: Chang et al., 2020; Cooper et al., 2020; Gaudel et al., 2020; Chang et al., 2022; Wang et al., 2022. Because these papers report all trend values, uncertainties, and all p-values, and also discuss the trend results, there is no confusion regarding the findings, and one does not even notice that the term "statistically significant" is not used at all. Table 3 of the TOAR-II statistical guidelines provides calibrated language for describing trends and uncertainty, similar to the approach of IPCC.

**Author's response**: Considering Wassertein et al. (2019) and recent ozone trends publications, we have proposed in the revision process in the manuscript:

- Trends calculation per decade
- Modifications in the trend's discussion, in abstract and conclusion section taking into account the TOAR-II statistical guideline and presenting all calculated trends.

**Author's changes in manuscript**: Modified Line 44 to 50, 316 to 317, 319 to 323, 819 to 831, 835 to 837, 840 to 846, 849 to 855, 894 to 905 and 974 to 979.

**Comment 2**: In particular, Figure 16 only shows trend values that were deemed "significant" based on the p-value; this approach is inconsistent with the TOAR guidelines and the figure needs to report trend values from all sites, with 95% confidence intervals and p-values. Another TOAR guideline is that trends be reported in units of ppbv per decade so that trends can be compared between different studies. It's fine if you want to report your trend s in units of % yr$^{-1}$, but please also report the trends in ppbv decade$^{-1}$.

**Author's response**: It is now modified in the revised manuscript (in figures 15 and 16).

**Author's changes in manuscript**: Modified Line 881 to 885 and 943 to 945.

**Comment 3**: The "Data Availability" statement provides links for each of the monitoring stations, allowing the observations to be downloaded. However, it's not clear which link belongs to which station. It would be helpful if the links can be listed in a table, which also provides the station name and country

**Author's response**: We have proposed in the reviewed manuscript a new table in the "Data availability" section to present sites and associated DOIs (links) (with the complete citation reference that will be included in the reference list.

**Author's changes in manuscript**: Modified Line 990 to 997.

---

## Referee Report (RR1)

**18 Aug 2024 Suggested Changes to Donnou et al. ABSTRACT – The purpose is to have GREATER Impact for TOAR II and Beyond.**

**Original has more nearly 500 words**, ACP limit is 250?

In the framework of the International Network to study Deposition and Atmospheric chemistry in Africa (INDAAF) program, we present the seasonal variability of atmospheric ozone concentrations at the regional scale. The correlations of local atmospheric chemistry and meteorological parameters to ozone photochemistry are investigated, as are long-term trends in ozone concentrations. Fourteen measurement sites were identified for this study, representative of the main African ecosystems: dry savannas (Banizoumbou, Niger; Katibougou and Agoufou, Mali; Bambey and Dahra, Senegal), wet savannas (Lamto, Côte d'Ivoire; Djougou, Benin), forests (Zoétélé, Cameroon; Bomassa, Republic of Congo) and semi-agricultural/arid savanna (Mbita, Kenya; Louis Trichardt, Amersfoort, Skukuza and Cape Point, South Africa). As part of several study programmes, validation and intercomparison tests of passive samplers at remote sites have been carried out to ensure controlled-quality measurements and to provide reliable long-term gas concentrations. Over the period 1995-2020, monthly ozone concentrations were measured at these sites using passive samplers. Monthly averages of surface ozone range from 4.7±1.4 ppb (Bomassa) to 31.0±10.5 ppb (Louis Trichardt). Ozone concentrations in the wet season (in dry savanna) are higher and in the same order of magnitude comparable to concentrations in the dry season (in wet savanna and forest). In East Africa, ozone levels show no marked seasonality., We established a positive gradient of mean annual O3 concentrations from West Central Africa to South Africa. In the dry savanna, under the influence of temperature, ozone concentrations are closely linked to Biogenic Volatile Organic Carbon (BVOC) emissions (0.51 < r < 0.95). They are also sensitive to nitrogen monoxide (NO) emissions in the presence of high precipitation and humidity. Biogenic VOC emissions, anthropogenic NOx, temperature and radiation exhibit a good correlation (0.49 < r < 0.92) 40 with O3 formation in wet savannas and forests. At the southern African sites, the photochemistry of O3 is influenced most by humidity, rainfall, temperature, NOx emissions (anthropogenic and biogenic) and VOC. At the annual scale, from 2000 to 2020, Katibougou and Banizoumbou sites (dry savanna) experienced a significant decrease in ozone concentrations respectively around -0.2.4 ppb decade (with a very high certainty) and -0.8 -0.15 ppb /decade at a 95% confidence interval. Seasonal Kendall statistical tests revealed with a high certainty decreasing trends of -0.0.7 ppb decade in Banizoumbou and -0.2.4 ppb/decade in Katibougou. These decreasing trends are consistent with those observed for nitrogen dioxide (NO2) and biogenic VOCs. An significant increasing trend is observed in Zoétélé (2001-2020), estimated at 0.71 ppb decade-1 and at Skukuza (2000-2015; 0.3.4 ppb decade). The increasing trends are consistent with the increase in biogenic emissions at Zoétélé and NO2 levels at Skukuza. Very few surface O3 measurements exist in Africa, and long-term results presented in this study are the most extensive for the studied ecosystems.

**COMMENTS ON CHANGES – Because the paper now has less about links (hope that those correlations can go in a 2nd paper!) it is recommended to remove the shaded material to shorten the Abstract. Also the uncertainty information is too much for the Abstract. Below is 429 words.**

[revised manuscript text omitted]

---

## Author Response (AR2)

**Response to Reviewers**

Title: Measurement report: Long-term measurements of ozone concentrations in semi-natural African ecosystems

Author(s): Donnou et al.
MS No.: acp-2024-284

The authors would like to thank the editor and reviewers for the valuable time spent on the manuscript to further improve its quality. Their comments were constructive and helpful.

In general, we have modified the title, abstract and conclusion of the manuscript. Corrections have been made in the introduction by rewording some sentences. The axis ranges of some figures in sections 3.1 and 3.2 have been reduced. A recent reference has been added to the manuscript in the discussion of annual ozone trends. The analysis of these trends has been reformulated and summarized in Table 6.

Below, we provide a point-by-point response explaining how we have responded to each of the referees' comments. Our responses to the comments are in blue. We are confident that this stage of revision has further improved the manuscript. We look forward to the referees' feedback on the revised article, and to the editor's decision. With our best regards, and our deepest gratitude to the referees and editor.

On behalf of all the co-authors

Sincerely yours
Venance Donnou and Corinne Galy-Lacaux

**REFEREES' REPORTS:**

**REFEREE 1**

**Recommendations**: I recommend this be highlighted in a revised title: Long-term measurements of surface ozone and trends in semi-natural ecosystems in sub-Saharan African ecosystems." The latter change is recommended because the paper does not include north African data from nations along the Mediterranean.

**Author's response**: This comment is now included in the revised manuscript.

**Author's changes in manuscript**: Modified Title of Manuscript

Line 138 – Better form to say low surface ozone… "recorded at many sites…"

**Author's response**: This comment is now included in the revised manuscript.

**Author's changes in manuscript**: Modified Line 142

Line 142 – Do you mean " Lee et al. (2021) used models and measurements to estimate that 24% of boundary-layer ozone over Africa is estimated from biomass burning?" And what does that mean "over Africa" – the entire continent? Which regions? Important to clarify whether papers like Lee et al are using models or actual data

**Author's response**: In the works of Lee et al. (2021), the authors used aircraft-based measurements of $O_3$ and a range of its precursors in African wildfire outflow during 12 research flights spanning March 2017 to February 2020. The flights ranged from within a few kilometres of the fires over Senegalese and Ugandan savannah to several hundred kilometres away over the North Atlantic Ocean near Cape Verde. This part has been further clarified in the revised manuscript by integrating the study sites and data used.

**Author's changes in manuscript**: Modified Line 145 to 148

Lines 154-158. An awkward sentence – the meaning is not clear. Do you mean "Changing isoprene emissions, the temperature sensitivity of NOx and $O_3$ chemistry (Brown et al., 2022) as well as meteorological changes have all been implicated in the seasonality and spatial patterns of ozone trends in the tropics (Stauffer et al., 2024)" ?

**Author's response**: This paragraph has been rephrased in the revised version of the manuscript. We say that Brown et al, (2022) and Stauffer et al, (2024) recommended studies on isoprene emissions changes, on the temperature sensitivity of NOx and $O_3$ chemistry, as well as on meteorological changes involved in the seasonality and spatial patterns of ozone trends in the tropics.

**Author's changes in manuscript**: Modified Line 161 to 163

Line 219 - use a monthly…

**Author's response**: This comment is now included in the revised manuscript.
**Author's changes in manuscript**: Modified Line 221

Line 412 – In the dry savanna (insert 'the')

**Author's response**: This comment is now included in the revised manuscript.
**Author's changes in manuscript**: Modified Line 402

Line 609 – should read "during which" not during "what"

**Author's response**: This comment is now included in the revised manuscript.

**Author's changes in manuscript**: Modified Line 606

Line 612 – can delete "was evaluated and"

**Author's response**: This comment is now included in the revised manuscript.

**Author's changes in manuscript**: Modified Line 609

Line 640 – "densify" not a good word. Start sentence with Additional efforts must therefore be made through programs to enhance the density of monitoring networks…

**Author's response**: This comment is now included in the revised manuscript.

**Author's changes in manuscript**: Modified Line 636 to 637

Lines 835 to 840 – These sentences are confusing. In general you are using different criteria on uncertainties- (Mann-Kendall) and p-values. That is fine. Figures 14 and 15 provide excellent summaries of the results! In both cases it is seen that only 3-4 sites have annual trends with low confidence (p value >0.2). Figure 14 is very convincing that losses occur at a number of ecosystem types. Recommend that you revise text with less reference to uncertainty – the reader can see that in the Figures and also in a Table – recommend in the next paragraph that you add a Table. The discussion in Section 3.1 would be easier to follow if you make a Table with the 4 columns: station name // trend in ppbv/decade // p value // addl comment on confidence level/ comment on related VOC or other trend. Line 837 – no need to say "there is very little chance … will occur."

**Author's response**: This comment is now included in the revised manuscript.

**Author's changes in manuscript**: Modified Line 657 to 692

Line 856 – Start a new paragraph with "The absence…"

**Author's response**: This comment is now included in the revised manuscript.

**Author's changes in manuscript**: Modified Line 694

Lines 858-869. The Wang et al. (2022) is mostly a modeling study with satellite results; that is not a good comparison point for your observational study. Recommend you delete it in this part of the paper. Comparisons made with Gaudel et al (2020) are somewhat relevant and your points about the airport stations being more polluted and close to sources are excellent interpretations. Note, however, at Nairobi (Thompson et al., 2021) the ozonesonde changes are almost negligible, so even in urban areas trends can be modest.

**Author's response**: The reference of   Wang et al. (2022) has been deleted in the manuscript the revised manuscript.

**Author's changes in manuscript**: Modified Line 698

You should add a new TOAR II paper in press: Gaudel et al. (2024). It will be published very soon and is a successor to the Gaudel et al. (2020). The new paper uses a lot of IAGOS aircraft data to derive trends ~1995-2019. However, the Supplemental Material in the paper (look at egusphere-2023-3095) reveals a large "jump" or discontinuity over IAGOS stations not only in Africa but over South America between 1994-1997. That jump is generally NOT seen in African or South American ozonesonde records (Witte et al., 2017; 2018) although there are only 3 central and South American stations with 20 year trends. Trends after 2000 (a paper in preparation by Van Malderen et al. (for TOAR II on TOAR II /HEGIFTOM ground based data: 2000-2022 – tropospheric ozone from spectrometers as well as sonde and aircraft profiles), also show quite modest trends. In summary, one can assume that Gaudel et al. (2020; 2024) report overestimates of African trends. Gaudel et al. (2024) also contains NEW OMI/MLS satellite data (newer than Hou et al.) that cover 2005-2019/2020. The new data have relatively small trends over Africa, an excellent reference for your paper.

**Author's response**: The reference of Gaudel et al. (2024) has been cited in the revised manuscript.
**Author's changes in manuscript**: Modified Line 698 to 702, 708 and 732 to 739

Line 894 - Start the sentence with "The tests reveal…"

**Author's response**: It is now in the revised manuscript
**Author's changes in manuscript**: Modified Line 732

Lines 924 – Remember to use months not "spring" or "summer" for Irene because the seasons are opposite months in the southern hemisphere.

**Author's response**: It is now included in the revised manuscript

**Author's changes in manuscript**: Modified Line 750

Line 961 – "In the semi-arid"

**Author's response**: It is now included in the revised manuscript

**Author's changes in manuscript**: Modified Line 785

Suggested Text Changes in the Conclusion

**Author's response**: Suggestions for changes in the conclusion are included in the revised manuscript.

**Author's changes in manuscript**: Modified Line 801 to 818

Suggested Text Changes in the Abstract

**Author's response**: Suggestions for changes in the abstract are included in the revised manuscript.

**Author's changes in manuscript**: Modified Line 25 to 58

**REFEREE 2**

Technical Corrections:

It should be NO (subscript x) and not NOx, please correct throughout the manuscript.

**Author's response**: This comment is now included in the revised manuscript.

**Author's changes in manuscript**: Modified Line 67 to 1 194

Line 101, page 3:

ENSO is first used in line 101, page 3 but defined in line 119, page 3.

**Author's response**: This comment is now included in the revised manuscript.

**Author's changes in manuscript:** Modified Line 111

Line 125, page 3:

I would keep the phrase Further work has been carried out in different locations in Africa to characterise O3 levels.' without including the citation to avoid repetitions since the citations are mentioned just below.

**Author's response**: It is now in the revised manuscript

**Author's changes in manuscript:** Modified Line 134 to 138

Line 266, page 8:

NOX has been introduced in line 58 and VOCs in line 59.

**Author's response**: It is now in the revised manuscript

**Author's changes in manuscript:** Modified Line 282

Line 321, page 10:

ITCZ abbreviation has been already introduced in line 138.

Please check where the abbreviations are introduced throughout the manuscript.

**Author's response**: It is now included in the revised manuscript

**Author's changes in manuscript:** Modified Line 337

Figure 2, page 10:

Would it be possible to keep the same axis ranges for all the sites (e.g. 0-600 for left axis and 0-7 for the right axis)? I believe that having the same ranges would help to compare between the sites. I would suggest having as few different ranges as possible. Same for Figure 5, page 14 and Figure 11, page 20.

**Author's response**: The reviewer's comments are pertinent and have been taken into account for the most part. Indeed, to prevent certain parameters represented on the graphs from being overwritten and illegible to the reader, we have not been able to set all the graphs to the same range. However, we tried as far as possible to having as few different ranges.

**Author's changes in manuscript:** Modified Line 350 to 365 (Figure 2), Line 423 to 433 (Figure 5), Line 488 to 499 (Figure 8) and Line 553 to 564 (Figure 11).